# When Prompt Engineering Meets Software Engineering: CNL-P as Natural and Robust "APIs" for Human-AI Interaction

**Zhenchang Xing**
CSIRO's Data61
`zhenchang.xing`
`@data61.csiro.au`

**Yang Liu,**[*] **Zhuo Cheng,**[†] **Qing Huang**[‡]
Jiangxi Provincial Key Laboratory for High Performance
Computing, State International Science & Technology
Cooperation Base of Networked Supporting Software,
Language Intelligence Research Center, Jiangxi Normal University
`{bgly, zcheng, qh}@jxnu.edu.cn`

**Dehai Zhao**
CSIRO's Data61
`dehai.zhao@data61.csiro.au`

**Daniel SUN**
Institute of AI, Newcastle University, UK and UGAiForge
`Daniel.sun@newcastle.ac.uk`

**Chenhua Liu**
Jiangxi Normal University
`programmerliu@jxnu.edu.cn`

## Abstract

With the growing capabilities of large language models (LLMs), they are increasingly applied in areas like intelligent customer service, code generation, and knowledge management. Natural language (NL) prompts act as the "APIs" for human-LLM interaction. To improve prompt quality, best practices for prompt engineering (PE) have been developed, including writing guidelines and templates. Building on this, we propose Controlled NL for Prompt (CNL-P), which not only incorporates PE best practices but also draws on key principles from software engineering (SE). CNL-P introduces precise grammar structures and strict semantic norms, further eliminating NL's ambiguity, allowing for a declarative but structured and accurate expression of user intent. This helps LLMs better interpret and execute the prompts, leading to more consistent and higher-quality outputs. We also introduce an NL2CNL-P conversion tool based on LLMs, enabling users to write prompts in NL, which are then transformed into CNL-P format, thus lowering the learning curve of CNL-P. In particular, we develop a linting tool that checks CNL-P prompts for syntactic and semantic accuracy, applying static analysis techniques to NL for the first time. Extensive experiments demonstrate that CNL-P enhances the quality of LLM responses through the novel and organic synergy of PE and SE. We believe that CNL-P can bridge the gap between emerging PE and traditional SE, laying the foundation for a new programming paradigm centered around NL. To further demonstrate the effectiveness of CNL-P, we develop the AI-native CNL-P IDE, UGAiForge that leverages CNL-P to enable users to Think, Describe, and Build with AI and for AI. For detailed information, please refer https://ugaiforge.ai.

## 1 Introduction

In the context of AI, particularly with models like GPT-4 (OpenAI, 2024), a "prompt" refers to the input that directs the model's output. Prompt engineering is the process of designing these inputs

---

[*]Zhenchang Xing and Yang Liu are co-first authors.
[†]Zhuo Cheng is a corresponding author.
[‡]Qing Huang is a corresponding author.

to guide the model toward generating the desired output. This can involve specifying parameters, providing examples, or framing instructions in a particular way. Prompts enable humans to interact with AI systems. This interaction is akin to how we use software to interface with an operating system via APIs (Application Programming Interfaces). If we consider models like GPT-4 as an "AI operating system", prompts can be viewed as a new form of API for interacting with this system. The rationale is straightforward: (i) Interfacing with AI: Prompts act as the mechanism through which users interact with and instruct the AI, just as APIs facilitate communication and command execution between software applications. (ii) Guiding Behavior: Prompts shape the output and behavior of the AI model, in the same way that APIs define the possible interactions with software. (iii) Accessibility: Similar to how APIs allow developers to access the pre-built functionality of a software application, prompts allow users to leverage the capabilities of AI models without needing to understand or manipulate their underlying workings.

However, we need to be mindful of certain nuances. (i) Expressive Limit: While traditional APIs can handle complex, chained requests, prompts may struggle to express such complexity or manage varied tasks in a single interaction. (ii) State Management: Traditional APIs often have mechanisms to manage state and persistent data across interactions. In contrast, prompts, especially in stateless models like GPT-4, do not retain state or memory across interactions. (iii) Response Predictability: APIs typically provide defined and predictable responses based on documentation, whereas AI models like GPT-4 may produce responses with some degree of variability and unpredictability.

To address these nuances, the research community has introduced several effective approaches. A technical direction focuses on using programming languages (PL) to drive AI models. Methods such as LangChain (LangChain, 2024), Semantic Kernel (Microsoft, 2024a), and DSPy (Khattab et al., 2023) attach abstraction and templates to PL, while others, like Guidance (AI, 2024), LMQL (SRI, 2024), SGLang (Zheng et al., 2024), and APPL (Team, 2024), extend PL natively. These approaches harness the expressive power, robust state management, and precise control capabilities of PL to tackle the aforementioned challenges.

However, they also bring new challenges, including: (i) Complex NL-PL Conversion and Wrapping: By "forcing" models to output in specific formats (e.g., JSON), these methods require complex parsers to capture and fix many edge cases of the output. (ii) Prompt-Code Tight Coupling: Prompts are often tightly integrated with code, requiring extensive debugging and optimization to achieve desired outcomes. However, the code implementing the prompt is often simple, merely acting as a logical connector, making it difficult to separate the roles of prompt engineers and developers, leading to inefficiency. (iii) Unfriendliness to Non-Technical Personnel: Prompt debugging typically requires language and domain experts to iteratively optimize semantics. However, traditional development methods encapsulate prompts within code, preventing language experts and non-technical experts from participating in the development and optimization process, and limiting their ability to contribute. This development approach makes prompt engineering difficult to implement effectively. To better utilize the expertise of language experts and non-technical experts, we need to decouple prompts from code, enabling prompt engineering to adopt role division and collaborative development like other software engineering processes.

Programming languages (PL) have solved many challenges but also introduced new ones. We should consider using natural language (NL) to take on some of the functions traditionally handled by PL. This leads to an alternative technical path, which involves using NL templates or style guides, e.g., RISEN (Balmer, 2024), RODES (Sebo, 2024), to drive AI models. While this approach can improve response predictability, it still struggles with issues like expressive limits and state management. This is because current methods only provide micro-level incremental upgrades to NL. If we treat prompts as a new form of software, we should integrate software engineering (SE) principles at a macro level into NL. To tackle these challenges, we can apply first principles thinking to seek solutions. **Identifying and Defining the First Principles:** (i) Core of Traditional Coding: Writing instructions in a programming language is the fundamental method for creating software. (ii) Key SE Principles: Modularity, abstraction, encapsulation, and maintainability form the basis of good software design and development. **Deconstructing the Problem:** (i) Complexity of Traditional Coding: Traditional coding can be difficult and inaccessible for those without specialized training, limiting wider participation in software development. (ii) AI's Impact on Coding: AI, particularly generative AI, is changing the landscape of coding by automating or simplifying certain aspects of the process. (iii) Need for Intuitive Programming: There is a growing demand for more intuitive and accessible programming methods, which could democratize software development and increase

efficiency. **Reconstructing the Problem:** (i) Introducing Controlled NL (CNL): Propose CNL as an alternative to traditional coding, allowing users to write instructions in a format closer to natural language, making programming more accessible. (ii) Adhering to SE First Principles: Design CNL to support core SE principles like modularity and abstraction, while ensuring it remains easy to understand and use. (iii) AI as an Interpreter: Use AI to interpret CNL and translate it into executable code, positioning AI not just as an automation tool, but as a bridge between human language and machine-readable code. These insights culminate in our vision of advancing a more natural and robust SE paradigm as an AI infrastructure to enhance human-AI Interaction.

## 2 DESIGN PHILOSOPHY FOR CNL FOR HUMAN-AI INTERACTION

### 2.1 PRINCIPLES OF SOFTWARE ENGINEERING

The first principles of SE are fundamental concepts and truths that form the backbone of the SE discipline. These principles are not specific to any language, technology, or framework but are universal guidelines that shape good software development practices. They also inform the macro-level design of Controlled Natural Language (CNL). Here are some key SE first principles:

**Modularity** Divide software into distinct, independent components. This improves understanding, development, maintenance, and debugging. Each module should serve a specific function and interact with others in a well-defined manner.

**Abstraction** Simplify complexity by exposing only the necessary aspects. It represents essential features without including extraneous details, allowing developers to focus on high-level concepts without being bogged down by minutiae.

**Encapsulation** Encapsulation bundles data (variables) and the methods (functions) that operate on that data into a single unit (like a class in object-oriented programming). It also restricts direct access to some of the object's components to prevent misuse or unintended interference with the methods and data.

**Separation of Concerns (SoC)** Manage different aspects of the software separately, organizing code so that each part addresses a distinct concern. This makes the system more scalable, manageable, and easier to understand.

### 2.2 INSPIRATIONS FROM PROMPT ENGINEERING

CNL-P introduces several best practices from PE that enhance interaction with AI models. Some typical practices are outlined below:

**Persona** A "persona" refers to the specific role or identity assigned to the model in the prompt. Defining a persona helps the model adopt a certain tone, style, or perspective, improving the relevance and quality of its responses. For example, specifying that the model should respond as a subject matter expert (e.g., "As a medical expert, explain...") can make the response more authoritative. This helps the model better understand the context and adjust its answers accordingly, leading to more engaging and appropriate outputs.

**Constraints** "Constraints" are guidelines or limitations imposed on the model's output to shape the response more effectively. Constraints focus on the model's creativity, ensuring the results meet specific expectations. For instance, a format constraint might specify how the response should be structured (e.g., "List five benefits in bullet points"), while a content constraint might dictate what to include or exclude (e.g., "Discuss renewable energy without mentioning solar power").

**Chain of thought (CoT)** CoT refers to a technique where the model is encouraged to articulate its reasoning process step-by-step before arriving at a final answer. This method helps break down complex tasks into manageable parts, improving clarity and accuracy.

```
CNLP_AGENT := "[DEFINE_AGENT:" AGENT_NAME ["\"" STATIC_DESCRIPTION "\""] "]"
CNLP_PROMPT "[END_AGENT]"
CNLP_PROMPT := PERSONA [CONSTRAINTS] [TYPES] [VARIABLES] [WORKER]
AGENT_NAME := <word>

OPTIONAL_ASPECT := OPTIONAL_ASPECT_NAME ":" DESCRIPTION_WITH_REFERENCES
OPTIONAL_ASPECT_NAME := <word>
ASPECT_NAME := ROLE_ASPECT_NAME | OPTIONAL_ASPECT_NAME

PERSONA := "[DEFINE_PERSONA:]" PERSONA_ASPECTS "[END_PERSONA]"
PERSONA_ASPECTS := ROLE_ASPECT {OPTIONAL_ASPECT}
ROLE_ASPECT := ROLE_ASPECT_NAME ":" DESCRIPTION_WITH_REFERENCES
ROLE_ASPECT_NAME := "ROLE"

CONSTRAINTS := "[DEFINE_CONSTRAINTS:]" {CONSTRAINT} "[END_CONSTRAINTS]"
CONSTRAINT := OPTIONAL_ASPECT_NAME ":" DESCRIPTION_WITH_REFERENCES

TYPES := "[DEFINE_TYPES:]" {ENUM_TYPE_DECLARATION |
STRUCTURED_DATA_TYPE_DECLARATION} "[END_TYPES]"
ENUM_TYPE_DECLARATION := DECLARED_TYPE_NAME "=" ENUM_TYPE
STRUCTURED_DATA_TYPE_DECLARATION := DECLARED_TYPE_NAME "="
STRUCTURED_DATA_TYPE
DECLARED_TYPE_NAME := <word>

DATA_TYPE := ARRAY_DATA_TYPE | STRUCTURED_DATA_TYPE | ENUM_TYPE |
TYPE_NAME
TYPE_NAME := SIMPLE_TYPE_NAME | DECLARED_TYPE_NAME
SIMPLE_TYPE_NAME := "text" | "number" | "boolean"
ENUM_TYPE := "[" <word> {, <word>} "]"
ARRAY_DATA_TYPE := "List [" DATA_TYPE "]"
STRUCTURED_DATA_TYPE := "{" STRUCTURED_TYPE_BODY "}"  |  "{ }"
STRUCTURED_TYPE_BODY := TYPE_ELEMENT | TYPE_ELEMENT ","
STRUCTURED_TYPE_BODY
TYPE_ELEMENT := ELEMENT_NAME ":" DATA_TYPE
ELEMENT_NAME := <word>

VARIABLES := "[DEFINE_VARIABLES:]" {VARIABLE_DECLARATION} "[END_VARIABLES]"
VARIABLE_DECLARATION := ["\"" DESCRIPTION_WITH_REFERENCES "\""] VAR_NAME ":"
DATA_TYPE
VAR_NAME := <word>

WORKER := "[DEFINE_WORKER:" ["\"" STATIC_DESCRIPTION "\""] WORKER_NAME "]"
[INPUTS] [OUTPUTS] MAIN_FLOW "[END_WORKER]"
WORKER_NAME := <word>

INPUTS := "[INPUTS]" {REFERENCE_DATA} "[END_INPUTS]"
OUTPUTS := "[OUTPUTS]" {REFERENCE_DATA} "[END_OUTPUTS]"
REFERENCE_DATA := "<REF>" VAR_NAME "</REF>"

MAIN_FLOW := "[MAIN_FLOW]" {BLOCK} "[END_MAIN_FLOW]"
CONDITION := DESCRIPTION_WITH_REFERENCES

BLOCK := SEQUENTIAL_BLOCK | IF_BLOCK
SEQUENTIAL_BLOCK := "[SEQUENTIAL_BLOCK]" {COMMAND}
"[END_SEQUENTIAL_BLOCK]"
IF_BLOCK := "[IF" CONDITION "]" {COMMAND} {"[ELSEIF" CONDITION "]" {COMMAND}}
["[ELSE]" {COMMAND}] "[END_IF]"

COMMAND := COMMAND_INDEX COMMAND_BODY
COMMAND_INDEX := "COMMAND-" <number>
COMMAND_BODY := GENERAL_COMMAND | CALL_API | REQUEST_INPUT |
DISPLAY_MESSAGE
GENERAL_COMMAND := "[COMMAND" DESCRIPTION_WITH_REFERENCES ["RESULT"
COMMAND_RESULT ["SET" | "APPEND"]] "]"
DISPLAY_MESSAGE := "[DISPLAY" DESCRIPTION_WITH_REFERENCES "]"
REQUEST_INPUT := "[INPUT" ["DISPLAY" DESCRIPTION_WITH_REFERENCES "VALUE"
COMMAND_RESULT ["SET" | "APPEND"]] "]"
CALL_API := "[CALL" API_NAME {"," API_NAME} ["WITH" ARGUMENT_LIST {","
ARGUMENT_LIST}] ["RESPONSE" COMMAND_RESULT ["SET" | "APPEND"]] "]"
API_NAME := <word>
ARGUMENT_LIST := STRUCTURED_TEXT
COMMAND_RESULT := VAR_NAME ":" DATA_TYPE | REFERENCE

DESCRIPTION_WITH_REFERENCES := DESCRIPTION_ELEMENT
{DESCRIPTION_ELEMENT}
DESCRIPTION_ELEMENT := STATIC_DESCRIPTION | REFERENCE
STATIC_DESCRIPTION := <word> | <word> <space> STATIC_DESCRIPTION
REFERENCE := "<REF>" NAME "</REF>"
NAME := SIMPLE_NAME | QUALIFIED_NAME | ARRAY_ACCESS | DICT_ACCESS
SIMPLE_NAME := <word>
QUALIFIED_NAME := NAME "." SIMPLE_NAME | NAME "." ARRAY_ACCESS | NAME "."
DICT_ACCESS
ARRAY_ACCESS := NAME "[" <number> "]"
DICT_ACCESS := NAME "[" SIMPLE_NAME "]"

STRUCTURED_TEXT := "{" STRUCTURED_TEXT_BODY "}"
STRUCTURED_TEXT_BODY := FORMAT_ELEMENT | FORMAT_ELEMENT ","
STRUCTURED_TEXT_BODY
FORMAT_ELEMENT := KEY : VALUE | VALUE
KEY := <word>
VALUE := DESCRIPTION_WITH_REFERENCES | ARRAY | STRUCTURED_TEXT

ARRAY := "[" ARRAY_ELEMENTS "]" | "[ ]"
ARRAY_ELEMENTS := VALUE | VALUE "," ARRAY_ELEMENTS

<space> ::= " " | "\t"
<number> ::= <digit> {<digit>} ["." <digit> {<digit>}]
<digit> ::= "0" | "1" | "2" | "3" | "4" | "5" | "6" | "7" | "8" | "9"
<word> ::= <character> {<character>}
<character> ::= <letter> | <digit> | <symbol>
<letter> ::= "a" | "b" | "c" | "d" | "e" | "f" | "g" | "h" | "i" | "j" | "k" | "l" | "m" | "n" | "o" | "p" | "q" | "r" | "s"
| "t" | "u" | "v" | "w" | "x" | "y" | "z"
      | "A" | "B" | "C" | "D" | "E" | "F" | "G" | "H" | "I" | "J" | "K" | "L" | "M" | "N" | "O" | "P" | "Q" | "R"
| "S" | "T" | "U" | "V" | "W" | "X" | "Y" | "Z"
<symbol> ::= "!" | "@" | "#" | "$" | "%" | "^" | "&" | "*" | "(" | ")" | "-" | "_" | "=" | "+" | "[" | "]" | "{" | "}"
      | ":" | ";" | "'" | "\"" | "<" | ">" | "," | "." | "/" | "?" | "|" | "\\" | "~" | "`"
```

Figure 1: Syntax of CNL-P Expressed in BNF (Backus-Naur Form)

# 3  CONTROLLED NATURAL LANGUAGE FOR PROMPT (CNL-P)

## 3.1  CNL-P SYNTAX

The syntax of CNL-P is shown in Figure 1, with its execution facilitated by the `CNLP_AGENT`. Within this framework, the `CNLP_PROMPT` conveys the complete prompt, defining the prompt information and configuration for the intelligent agent. This serves as a comprehensive descriptor that specifies the agent's behavior, constraints, data types, variables, workflows, and more, ensuring its actions are well-defined, consistent, and predictable, which aids in smooth integration and interaction with various systems and components. Here is a detailed explanation of its role:

**Defining Agent's Prompt Information** `PERSONA`: Specifies the agent's role and characteristics. `CONSTRAINTS`: Defines limitations or rules the agent must follow. `DATA_TYPE`: Describes the types of data the agent will handle. `VARIABLES`: Lists the variables that the agent will use or manipulate. `WORKER`: Outlines the main flow of tasks and operations the agent will perform. `CALL_API`: calling APIs that the agent can interact with. `INPUTS`, `OUTPUTS`: Defines inputs and outputs for the worker. `REFERENCE_DATA`: References a variable. `COMMAND`: Defines a command, which may include general commands, API calls, input requests, or display messages.

**Ensuring Consistency and Clarity** By defining these components within `CNLP_PROMPT`, developers ensure agent's behavior is consistent and clear. This enhances **predictability**, allowing agent's actions to be anticipated based on defined prompt. It also simplifies **debugging**, as well-defined configurations facilitate easier identification and resolution of issues. Additionally, it provides clear **documentation** of agent's capabilities and limitations.

As mentioned in section 2, the design of CNL-P is inspired by SE and PE. More explanation for the alignment analysis can be found in Appendix A.2.

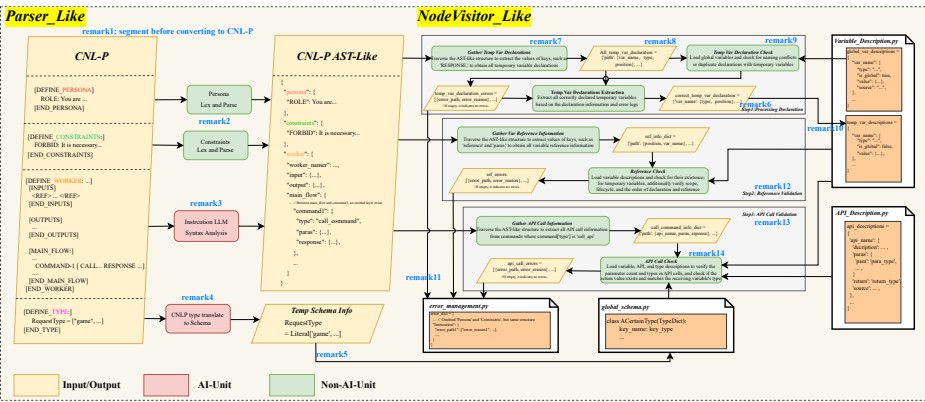

Figure 2: Overview of CNL-P Linting Tool

## 3.2 TRANSFORMER AGENTS

We have developed three transformer agents, RISEN, RODES, and CNL-P transformers, for the automatic conversion of NL to corresponding templates or CNL-P. These frameworks share a systematic approach to prompt generation, focusing on the meticulous extraction and structuring of user input. Specifically, the NL to CNL-P transformer agent simplifies CNL-P syntax interactions for non-experts, benefiting end-users by balancing user-friendliness and cognitive load. For more detailed prompts of these agents, please refer to the section A.3 of the Appendix.

## 3.3 CNL-P LINTING

Based on the precise grammar structures and strict semantic norms outlined in Section 3.1, and inspired by TypeChat (Microsoft, 2024b) and Pydantic (Pydantic, 2024), we developed a linting tool for CNL-P that includes both syntactic and semantic checks, making static analysis techniques applicable to NL. The methodology is illustrated in Figure 2.

We draw inspiration from the code compilation process and divide the static analysis of CNL-P into two stages, akin to compilation: syntactic analysis and semantic analysis. The syntactic analysis abstracts CNL-P into an *AST_Like* structure, physically representing it in JSON format. This tool to perform this series of steps is hereinafter referred to as *Parser_Like*. The semantic analysis, based on *CNL-P AST_Like* structure, gathers information by traversing the JSON and introduces background data. Precise checking of certain semantic information (particularly variable types) in CNL-P is entirely implemented through programming. The tool to perform this series of steps is hereinafter referred to as *NodeVisitor_Like*. For *Parser_Like*, we use text identifiers to distinguish structures in CNL-P. Unlike keywords, these identifiers can appear in the NL of CNL-P and are hereinafter referred to as *keyword_Like*. Some *keyword_Like* frequently occur in NL, (e.g., CALL), while others are less common (e.g., DEFINE_PERSONA).

In the static analysis process of CNL-P, relationship between LLM and Program is not either-or; instead, they complement each other. In the future, we will further organically combine LLM and the Program to conduct more comprehensive and detailed semantic checks on CNL-P. In addition, note that the entire process continues uninterrupted even if errors are encountered, ensuring that all semantic checks are fully executed. A more detailed description of the linting tool can be found in the Appendix A.4

In order to better demonstrate full workflow: (i) automatic conversion from NL to CNL-P conducted by NL to CNL-P transformer agent described in section 3.2, and (ii) applying linting tool described in this section for static checking, a running example is illustrated in section A.5 of the Appendix.

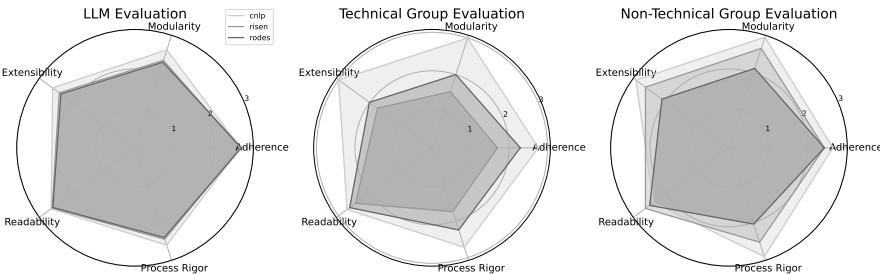

Figure 3: Overall Evaluation Radar

# 4 EXPERIMENTS

## 4.1 RESEARCH QUESTION 1: ASSESSING THE QUALITY OF CONVERSION

When converting NL prompts to conform to CNL-P or NL style guides using transformer agents like RISEN and RODES, evaluating the quality of the transformation is essential. Thus, we propose our first research question:

> *RQ1: How to evaluate the quality of the conversion from natural language prompts to CNL-P or NL style guides.*

To assess this, we outline five evaluation dimensions based on the prompting effect of LLMs:

**Adherence to Original Intent (D1)** Evaluates whether the structured prompt accurately reflects the original natural language intent, retaining core details without necessitating structural similarity. Structured keywords should enhance clarity and module boundary definition without distorting or negatively impacting the original meaning.

**Modularity (D2)** Measures how well the structured prompt extracts and organizes implicit structured knowledge—like data structures, variable types, and logical relationships—into independent modules with minimal coupling.

**Extensibility and Maintainability (D3)** Evaluates how easily the structured prompt can be modified, extended, or updated, ensuring stability and adaptability of the overall structure with clear rationale and guidance for changes.

**Readability and Structural Clarity (D4)** Assesses the logical organization and clarity of the prompt's structure, ensuring that structured keywords highlight module boundaries and aid comprehension. Structured keywords should make the prompt more readable to an LLM by clearly delineating different sections without causing confusion or redundancy.

**Process Rigor (D5)** Assesses the structured prompt's workflow integrity, including the clarity of iterative steps, variable passing, and input-output management, ensuring a precise and logical flow that an LLM can follow.

The specific criteria of these five evaluation dimensions can be found in appendix A.6.1. We discovered a highly popular project (Akin, 2024), on GitHub, which has garnered 113K stars. This project is dedicated to collecting various prompts used with ChatGPT. To better structure these prompts, we selected 93 prompt instants with sufficient content and rich steps from the project. The results were then independently evaluated by LLM (GPT-4o) and human evaluators (4 technical personnel with over four years of programming experience and 2 non-technical evaluators from Education and Business School), using fundamentally distinct yet individually justified approaches aligned with the five evaluation dimensions. [1].

Before the evaluation, we did not provide any explanations related to CNL-P or Few-Shot Learning to the LLM, nor did we provide any additional training to the evaluators, allowing them to assess

---

[1]All the data and codes used in the experiment section can be found in the repository through link https://github.com/Irasoo/CNL-P.

based on their own understanding. For LLM evaluation, we asked the LLM to evaluate all 93 * 3 (each initial NL prompt was converted into CNL-P, RISEN, and RODES) converted prompts according to the scoring criteria on a scale of 100, the detailed results are displayed in Figure 3. For human evaluation, because assessing "Adherence to Original Intent" and "Process Rigor" requires a significant amount of time for comparison with the original text, and considering that performing a detailed evaluation on a 100-point scale is difficult (Hamel, 2024), we randomly selected 30 * 3 converted prompts for evaluation. Evaluators assessed them on a "low, medium, high" scale according to the scoring criteria, and the detailed results were also displayed in Figure 3.

Based on the results, CNL-P showed the most significant advantage in the technical group's evaluation and also had certain advantages in the non-technical group and LLM evaluations. The non-technical group's results aligned more closely with the LLM-based evaluation, which suggests that the LLM's understanding of the effects of CNL-P is similar to the perception of the target user group for CNL-P.

On the dimensions of "Modularity" and "Extensibility and Maintainability," all three groups consistently showed CNL-P's clear advantage. Combined with feedback from the technical group: "When evaluating these two dimensions, the scores for CNL-P, RISEN, and RODES generally do not change due to specific content differences in the Prompt but are determined by their internal structure," it suggests that this is an inherent advantage of CNL-P, a grammar-driven language, over simple NL Templates in terms of underlying design.

On the "Process Rigor" dimension, CNL-P also demonstrated advantages in all three groups. Non-technical evaluators noted that after a brief adaptation, CNL-P's use of fixed characters to distinguish modules and different line indentations to define hierarchy made the entire process description much more rigorous. It was very similar to the structured documents they were familiar with but with clearer semantic definitions than those structures.

However, on the "Readability and Structural Clarity" dimension, non-technical evaluators, due to a lack of relevant technical knowledge, rated CNL-P lower in readability, even though they found its structure visually clearer. They needed more time to understand the application of Types and Variables in the Prompt, which led to lower readability scores in the non-technical group. (This threshold could be reduced by an NL-to-CNL-P Transformer Agent, but it is beyond the scope of this experiment.)

On evaluating "Adherence to Original Intent," we observed that RISEN and RODES (especially RISEN) received significantly lower scores in the technical group compared to CNL-P. After discussions with the technical group, we found that their perspective on NL was different from that of the non-technical group. The non-technical group viewed "Adherence to Original Intent" literally, while the technical group considered the NL text as an "Agent definition for a type of task" and "a specific task that the Agent is supposed to solve." They believed that these two aspects needed to be clearly distinguished and that narrowing the Agent's scope would violate "Adherence to Original Intent." RISEN, in particular, narrows the scope of the Agent due to the presence of the "Expectation" module, which limits the Agent's ability to only address a specific task. For a more detailed discussion of RISEN, please refer to Section A.6.3.

From the results of all three groups, we observe that CNL-P was more favorably recognized compared to RISEN and RODES, especially with a strong preference shown by the technical group for CNL-P.

## 4.2 RESEARCH QUESTION 2: UNDERSTANDING OF CNL-P BY LLMS

Since CNL-P introduces precise syntax and semantics, it is more complex in form compared to simple template-based methods. We want to know whether this complexity will affect the understanding and execution of CNL-P by LLMs. Thus, we set up RQ2.

> *RQ2: Can LLMs understand and execute CNL-P without additional explanations or Few-Shot Learning?*

A recent paper (msclar, 2024) explored the performance fluctuations in LLMs due to small changes in prompt formats, such as changing "Passage: Answer: " to "PASSAGE: ANSWER:". The responses from LLMs reflected performance fluctuations in classification tasks within their train-

Figure 4: Comparison of NL, CNL-P, RISEN, and RODES Scores Across Models

ing datasets, caused by slight format variations. CNL-P differ significantly in "appearance" from other natural language prompts (NL Prompts). We aim to investigate whether significant changes in prompt structure and content at the macro level result in sharp fluctuations in model performance without providing any explanations or using Few-Shot Learning.

In this experiment, we selected six classification tasks from dataset (AllenAI, 2024) presented at Sclar et al. (2024). These tasks are with standard answers in a broad sense and different levels of difficulty. We hope to obtain an objective and fair conclusion for the RQ2's main objective through such tasks in this first work. There are some relatively complex tasks among these classification tasks.

For example, Task 1162 asks the model to assess whether the title of a given paper paragraph is appropriate. Some instants of this task are with many technical terms and the language is rigorous and precise, testing the model's reasoning and summarization abilities. Tasks 1424 and 1678, on the other hand, are related to mathematical reasoning. Task 1424 involves answering questions related to probability, while task 1678 is a computational problem. Tasks' details are shown in Figure 16 of the Appendix. There are some relatively complex tasks among these classification tasks. Detailed example can refer Figure 17.

For the underline LLMs, GPT-4o, Gemini-1.5-Pro-002, GPT-4o-Mini, Llama3-70B-8192, and Claude-3-Haiku are chosen.

The experimental process was as follows: (i) Instance Sampling: We set a random number seed and selected 50 instances from each task, ensuring that the same batch was used across different models during testing. (ii) Complete task instants: For each task, we generated three prompt types: CNL-P, RISEN, and RODES. Along with the NL prompt, these four prompts were used as system prompts respectively, to drive the LLMs to complete the tasks. (iii) Task Evaluation: Finally, we assessed whether each task was completed correctly by comparing the LLMs' responses to the corresponding standard answers. We employed five LLMs across six tasks, resulting in 30 groups of accuracy rate results for NL, CNL-P, RISEN, and RODES, respectively. We selected the prompt(s) with the highest accuracy rate from each group as the best prompt(s), assigning it a score of 100. The performance values of other prompts were then expressed relative to this best prompt. The results are detailed in Figure 18 and Figure 4. The experimental results show that although CNL-P introduces precise syntax and semantics, its simple and intuitive design allows LLMs to understand its syntax and semantics without the need for providing CNL-P language additional explanations or Few-Shot Learning. Overall, this allows LLMs to achieve performance comparable to natural language prompts in output generation.

## 4.3  RESEARCH QUESTION 3: STATIC CHECKING IN CNL-P

The proposed linting tool enables static checking in CNL for the first time. In this experiment, we explore the third research question.

> *RQ3: Can the designed linting tool effectively conduct static checking in natural language prompt that conform to CNL-P syntax?*

**Dataset Construction and Description**  We compared compilation errors in code against the syntactic features of CNL-P, identifying 14 distinct task types. Initially, we manually created three error templates and applied them to these tasks using GPT-4o. By leveraging GPT-4o, we generated a

dataset of 47 task instants, each with at least three instances, ensuring comprehensive coverage and accuracy for evaluating CNL-P's effectiveness in handling various coding errors.

**Summary of Results** Among the 47 task instants verified with the linting tool, the accuracy rate was 100% with a redundancy rate of 0%. In contrast, verification with GPT-4o revealed 6 cases of incorrect location and reason, 4 cases with correct location but incorrect reason, and 1 case with incorrect location but correct reason, resulting in an overall accuracy of 76.60% and a redundancy rate of 19%.

**Results Analysis** CNL-P linting demonstrated a significant advantage in meticulousness during checks. For instance, in task 007, when the global variable `_user_account_fitness` was used as a parameter for the API `get_workout_plan`, the key `region` with the value 'JP' did not conform to the optional fields ['US', 'CA', 'AU', 'UK', 'IN']. Despite multiple tests, the LLM consistently overlooked this error, whereas the linting tool identified it. Regarding redundant error content generation, the LLM's inherent flaw of "making things up" due to its NTP (Neural Text Processing) mechanism is evident. In multiple experiments, whether across different tasks with the same template or within the same task with different templates, redundant content generation randomly appeared in such clusters of instances. Nevertheless, CNL-P maintains a 76.60% accuracy and a 0% redundancy rate, indicating that the GPT-4o can indeed understand this structured natural language. This suggests GPT4-o remains capable of interpreting and processing structured prompts effectively. Therefore, we conclude that the designed linting tool can effectively conduct static checking for NL prompts conforming to CNL-P syntax. In Appendix A.8, we provide an example of an actual instance examined by the Linting tool. More analysis of the experiments can be found in Appendix A.9.

## 5  DISCUSSION

CNL-P as an executable requirement can enhance clarity, testing, and education. In the upcoming version's development, CNL-P will be treated as an executable requirement, integrating `ALTERNATIVE FLOWS` and `EXCEPTIONS` inspired by use case modeling, along with `SCENARIOS` inspired by Gherkin user stories. Additionally, the `API` section can include a construct `FUNCTION`, which clearly defines intended use and potential misuse directly within its definition, serving as inline documentation. This feature is particularly beneficial in educational contexts, emphasizing correct function usage and common pitfalls to avoid. Furthermore, by incorporating Python's doctest and Hypothesis, CNL-P as executable requirements simplifies the specification and testing process, ensuring reliability and clarity, making it an invaluable tool for developers and educators.

CNL-P has the potential to support the CNL-P to code paradigm through a full-stack compiler, ensuring a clear distinction between "What" (high-level intent) and "How" (low-level implementation). This CNL-P compiler will handle all compilation stages, ensuring efficient and error-free code generation. CNL-P's structured syntax enables developers to specify desired behavior without concerning themselves with implementation details, which the compiler will seamlessly translate into optimized machine code. Unlike NL-PL systems, CNL-P maintains a clear separation between specification and implementation, eliminating ambiguity and potential errors. This thorough approach ensures precise, efficient, and maintainable code, setting CNL-P apart from the tight coupling often found in NL-PL conversions.

The CNL-P agent represents a novel software integration form that extends beyond traditional programming paradigms. By seamlessly integrating with broader SE activities, such as generative user interface (UI) development and technical documentation (e.g., reStructuredText), the CNL-P agent enhances the efficiency and coherence of software development processes. This integration streamlines the creation of dynamic and responsive UIs and ensures that technical documentation is both comprehensive and easily maintainable.

However, it is important to note that CNL-P alone cannot address all the challenges associated with LLM-based agent development. Just as programming languages depend on compilers and a suite of supporting tools such as linters and testing frameworks, CNL-P requires a comprehensive ecosystem of software engineering tools to facilitate the development, testing, maintenance, and operation of CNL-P agents. We envision a broader suite of LLM-assisted software engineering tools to support the entire CNL-P development lifecycle, including: A CNL-P tutorial agent that teaches CNL-P, provides relevant examples, and answers syntax and semantic questions. A documentation agent

that converts CNL-P requirements into technical documentation (e.g., reStructuredText), integrating CNL-P into the broader software DevOps context. Such a software engineering ecosystem around CNL-P aligns with core software engineering principles and best practices, such as early error detection, which reduces the cost of fixing issues at later stages of development. By identifying errors or inconsistencies at the requirements level, CNL-P can prevent costly implementation mistakes. Together, these tools will form a CNL-P-centric SE4AI infrastructure, leveraging LLMs to empower more people to integrate AI into their lives and work.

## 6  RELATED WORK

Recent research in SE for AI (SE4AI) focuses on applying SE principles, such as modularity, abstraction, encapsulation, and maintainability, to PE, transforming it into a new form of SE. This shift enhances the predictability and quality of AI outputs while integrating AI capabilities into traditional SE workflows. The proposed CNL-P framework embodies this concept by incorporating SE principles to reduce NL ambiguity through precise grammar structures and semantic norms, enabling more accurate and consistent interpretation by LLMs. In line with SE4AI, CNL-P bridges the gap between PE and SE, laying the groundwork for a new NL-centered programming paradigm.

Another closely related area is AI4SE, which leverages AI to enhance SE tasks such as API design, code generation and automated testing. In this context, prompts serve as interfaces between humans and AI systems, akin to APIs in SE. By optimizing prompt design with CNL-P, which eliminates NL ambiguity through structured grammar, CNL-P enhances the clarity and effectiveness of these interactions, functioning as APIs for AI systems like GPT-4.

Compared to programming language (PL) extensions such as Guidance (AI, 2024), LMQL (SRI, 2024), SGLang (Zheng et al., 2024), APPL (Team, 2024), and LangChain (LangChain, 2024), Semantic Kernel (Microsoft, 2024a), and DSPy (Khattab et al., 2023) which attaching abstraction and templates to PL, use PLs to manage state and structure in LLM prompts, CNL-P offers a more accessible and natural solution. While PL extensions provide robust control and expressiveness, they encounter challenges like complex NL-PL conversion and tight prompt-code coupling, making them less suitable for non-technical users and collaborative development. In contrast, CNL-P decouples prompts from code, providing clear role separation and enabling collaborative PE by both developers and language experts, thus addressing key limitations of PL-based methods. [2]

When comparing NL-based methods, such as RISEN (Balmer, 2024) and RODES (Sebo, 2024), which aim to enhance AI model performance through NL templates and style guides, CNL-P stands out by offering a macro-level transformation. While RISEN and RODES provide micro-level improvements to NL, they fall short in terms of expressive power and state management. In contrast, CNL-P introduces a macro-level transformation by integrating SE principles to manage complexity and reduce NL ambiguity, resulting in more consistent and high-quality outputs from LLMs. By merging the strengths of NL and SE, CNL-P represents a significant advancement in PE, overcoming the limitations of both PL and NL-based approaches.

In summary, CNL-P aligns with SE4AI and AI4SE by introducing a controlled NL framework that enhances the precision, accessibility, and collaboration of PE. It effectively bridges emerging PE methods with traditional SE practices, offering a novel, NL-centered programming paradigm that propels both fields forward.

## 7  CONCLUSIONS

As LLMs progress, we introduce CNL-P to address the inherent ambiguity in NL prompts. By merging best practices from PE and SE, CNL-P utilizes precise grammar and semantic rules. Together with the developed NL to CNL-P transformer agent and a CNL-P linting tool, CNL-P opens the door to achieving the ambitious vision that is advancing a more natural and robust SE paradigm as an AI infrastructure to enhance human-AI Interaction.

---

[2] Comprehensive Comparative Evaluation of CNL-P versus DSPy, LangChain, and Semantic Kernel can be found in Appendix A.10

ACKNOWLEDGEMENTS

This work is supported by National Natural Science Foundation of China (Grant No. 62102171, 62262031), the Distinguished Youth Fund Project of the Natural Science Foundation of Jiangxi Province (Grant No. 20242BAB23011), Jiangxi Provincial Department of Education Science and Technology Youth Project (Grant No. GJJ210340).

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

# A APPENDIX

## A.1 PENS ANALYSIS

To establish a principled classification scheme for CNLs, Kuhn (2014) introduces a four-dimensional framework called PENS, representing Precision, Expressiveness, Naturalness, and Simplicity. Precision assesses the ambiguity, predictability, and formality of CNL definitions. Expressiveness indicates how well a CNL can represent concepts compared to natural language. Naturalness combines grammar modifications, understandability, and a natural aesthetic (look-and-feel). The fourth dimension can be called Complexity or to have a dimension of the type "more is better" — Simplicity. PENS creates a conceptual space for CNLs with a five-level classification (1 to 5) across these dimensions, blending natural and formal languages such as English and propositional logic. Based on this classification, CNL-P is denoted as $P^3E^5N^4S^4$. Its specific meanings include:

**Reliably interpretable languages** ($P^3$)**.** The syntax of these languages is heavily restricted, though not necessarily formally defined. The restrictions are strong enough to ensure reliable automatic interpretation.

**Languages with maximal expressiveness** ($E^5$)**.** Capable of conveying any message exchanged between two humans, covering all logical statements.

**Languages with natural sentences** ($N^4$)**.** Consisting of valid natural sentences recognized and understood by native speakers without need for instructions or training.

**Languages with short descriptions** ($S^4$)**.** Require a comprehensive description spanning more than one but fewer than ten pages.

Under this classification, CNL-P is characterized by concise grammar and semantics ($S^4$), predominantly expressed in NL ($N^4$), matching the expressiveness of NL ($E^5$), and offering moderate interpretability ($P^3$).

## A.2 ALIGNMENT ANALYSIS OF CNL-P WITH INSPIRATION OF SE & PE

The design of CNL-P embodies a comprehensive approach to SE principles, ensuring a flexible, maintainable, and extensible framework for defining and configuring intelligent agents. By adhering to principles such as modularity, abstraction, separation of concerns, and encapsulation, CNL-P offers developers a robust "API" platform.

**Modularity** CNL-P is organized into independent components, each responsible for specific functionalities. This modular design allows for independent development, testing, and maintenance, promoting reusability and scalability. Clear interfaces between components prevent unintended side effects, simplifying system maintenance and extension.

**Abstraction** High-level concepts in CNL-P abstract complex implementation details, enabling developers to focus on essential features. Non-terminal symbols and component abstractions provide a concise way to describe agent behavior and attributes. Well-defined interfaces and the abstraction of complex operations further streamline the development process, making the language easier to use and extend.

**Separation of Concerns** Adhering to the single responsibility principle, each component in CNL-P handles a specific function. Clear interfaces and boundaries ensure that changes in one component do not affect others, enhancing extensibility and ease of maintenance. This design keeps the system flexible and maintainable, allowing developers to manage different functionalities independently.

**Encapsulation**  CNL-P employs a component-based design where each component encapsulates its functionalities and data. Data hiding prevents direct modification of a component's internal state, maintaining integrity. Well-defined interfaces serve as contracts between components, promoting modularity and reusability. This encapsulation simplifies maintenance, as bugs and issues can be isolated and fixed without impacting the entire system.

CNL-P's design is also inspired from emergent practices in PE. The CNL-P syntax itself includes constructs for the `persona` and `constraints`, which inherently support best practices in PE. Moreover, its sequential workflow definition, conditional execution, and command execution resonate with the step-by-step reasoning typical of the CoT approach. These elements enable developers to define and configure agents that can perform complex tasks by breaking them down into smaller, manageable steps and executing them in a logical sequence. This also conforms to the idea of divide and conquer. The specific explanations are as follows.

**Sequential Definition**  The `WORKER` component facilitates the definition of sequential workflows, ensuring that tasks and operations are executed in a specified order. This sequential execution reflects the step-by-step reasoning characteristic of the CoT approach.

**Conditional Execution**  The `IF_BLOCK` component enables conditional execution of commands, allowing the agent to evaluate specific conditions and execute different commands accordingly. This aspect of conditional reasoning is crucial in the CoT approach, as it involves assessing intermediate states and making decisions based on those evaluations.

**Command Execution**  The `COMMAND` component defines individual commands that the agent can execute. These commands represent distinct reasoning steps within the CoT process, enhancing the agent's ability to perform complex tasks systematically.

## A.3   TRANSFORMER AGENTS

We have developed three transformer agents, RISEN, RODES, and CNL-P transformers, for the automatic conversion of NL to corresponding templates or CNL-P shown in Figure 5, Figure 6, Figure 7, and Figure 8.

These frameworks share a systematic approach to prompt generation, focusing on the meticulous extraction and structuring of user input. This common methodology can be broken down into several key stages.

**Role and Context Identification**  Each framework starts by defining the AI's role or persona, ensuring that it operates within the intended context. This step is essential for establishing the tone and scope of the AI's interaction with the user.

**Objective Extraction**  The frameworks then extract the primary goal or intent from the user's input, creating a clear foundation for the prompt and ensuring that the AI understands the main task or goal.

**Detailed Instructions and Steps**  Each framework provides a set of detailed, actionable steps to guide the AI through the execution process. This stage ensures the AI follows a structured path to achieve the desired outcome, improving the accuracy and effectiveness of its responses.

**Expectations and Examples**  To align the AI's responses with user intent, the frameworks specify expected outcomes or offer examples. This step clarifies the desired output, ensuring that the AI's response meets the user's needs.

**Constraints and Limitations**  Finally, the frameworks account for constraints and limitations, tailoring the AI's output to specific user requirements. This ensures the response stays focused and relevant, avoiding unnecessary deviations.

**Conversion Process**  The frameworks follow a step-by-step conversion process that transforms user input into a structured format, prioritizing accuracy and clarity. This process guarantees that the AI's response is precisely aligned with user input.

This systematic approach ensures that user input is converted into structured and coherent prompts, guiding the AI to perform tasks efficiently and accurately. Specifically, the NL to CNL-P transformer

```
### **Task Overview**
Transform the user's input into a RISEN
framework prompt, ensuring each section is
clearly defined and accurately reflects the
user's intent.

### **Understanding RISEN**
- Role: [Define the AI's role. E.g., Advisor,
Creator]
- Input: [Provide detailed input. E.g.,
Specific question or topic]
- Steps: [Outline clear steps. E.g., First,
provide an overview, then delve into details]
- Expectation: [State your desired outcome.
E.g., A comprehensive guide, a brief summary]
- Narrowing: [State any limitations,
restrictions, or what to focus on. E.g., word
limit, focus on cost-effective options]

### **Conversion Process**
1. **Role**: Identify and specify the role
based on the user's task (e.g., "Advisor,"
"Content Creator").
2. **Input**: Extract the key topics, questions,
or data provided by the user.
```

```
3. **Steps**: Outline clear, actionable steps
the AI should follow, in sequence.
4. **Expectation**: Determine the type of
output the user wants.
5. **Narrowing**: Include any constraints,
limitations, or specific focus areas.

### **Key Considerations**
- Keep it concise and accurate.
- Ensure each section directly reflects the
user's input.
- Maintain clarity and logical flow.

### **Prompt Architecture**

Role: [AI's role, e.g., "Analyst"]
Input: [User's detailed question or topic]

Steps:
1. [First action]
2. [Next action]

Expectation: [Desired outcome]
Narrowing: [Any restrictions or focus areas]
```

Figure 5: Transformer Agent for NL to RISEN Template

```
### **Task Overview**
You are tasked with generating a RODES framework prompt
based on user input, ensuring that all sections are
clearly defined and accurately tailored to the user's
intent.

### **Understanding RODES**
The RODES framework consists of five main components:
```
1. **Role (R)**: Defines the persona or identity the AI
should adopt.
2. **Objective (O)**: Specifies the main goal or task.
3. **Details (D)**: Outlines specific requirements,
constraints, or steps for the task.
4. **Examples (E)**: Provides examples of the desired
input and output.
5. **Sense Check (S)**: Confirms understanding and checks
for additional clarifications needed.
```
The final RODES prompt should adhere to this structure,
enabling the AI to perform the task effectively.

### **Conversion Process for Each RODES Component**
1. **Role (R)**:
   - Identify the persona or role needed for the task
based on user input.
   - Example: If the user asks for marketing advice, the
role could be "Digital Marketing Expert."
2. **Objective (O)**:
   - Extract the main goal or task from the user's input.
   - Example: If the user wants insights on social media
trends, the objective could be "Provide an analysis of
current social media trends."
3. **Details (D)**:
   - Include any specific instructions, requirements, or
steps provided by the user.
   - Example: If the user specifies, "Focus on trends
related to TikTok and Instagram," include this in the
details.
```

```
4. **Examples (E)**:
   - Identify and provide any examples given by the user
to clarify expectations.
   - If no examples are provided, mention this section
should be completed based on the context.

5. **Sense Check (S)**:
   - Ensure that a final step asks if the AI understands
the task and if additional details are needed.

### **Key Considerations**
- Keep the prompt clear and concise.
- Ensure each section directly reflects the user's intent.
- Maintain logical flow and avoid unnecessary complexity.

### **Prompt Architecture**
```
1. Role (R):
   [Define the AI's role based on the task. E.g., "SEO
Specialist"]

2. Objective (O):
   [State the goal. E.g., "Analyze and provide SEO
strategies for a new website."]

3. Details (D):
   [Include any requirements, constraints, or steps. E.g.,
"Focus on keyword research and link-building techniques."]

4. Examples (E):
   [Provide input-output examples if available, or
specify that the user should provide them.]

5. Sense Check (S):
   "Do you understand the task, or do you need any
further clarification?"
```

Figure 6: Transformer Agent for NL to RODES Template

agent simplifies CNL-P syntax interactions for non-experts, benefiting end-users by balancing user-friendliness and cognitive load.

```
#You are tasked with converting natural language user input
into a Domain-Specific Language (DSL) based on the DSL BNF
structure provided. Follow these guidelines carefully:

### 1. Understanding DSL BNF:
    Learn the syntax of DSL BNF as it will guide the
transformation process.
    Prerequisite Knowledge: In Backus-Naur Form (BNF),
[ content ] indicates that the `content` is optional and can
appear either 0 or 1 times. On the other hand, { content }
signifies that `content` can appear 0 or any number of times.
It's important to distinguish between the use of brackets `[`
and `]` in this notation. Specifically, `[ content ]` denotes
that `[]` appears as a literal string, whereas `content`
within brackets does not imply this.

<REF>CNL-P_Syantax_Expressed_in_BNF<REF>

### 2. **Conversion Process:**
    1. **Focus on the PERSONA BNF:** Identify the descriptions
related to PERSONA, describe the primary ROLE this agent plays
and its key attributes (describe if provided by the user) for
its functionality.

    2. **Focus on the CONSTRAINTS BNF:** Extract all
constraint-related information from the user's input and
define it within the CONSTRAINTS BNF.
```

```
3. **Define Variables and Data Types:**
    - **3.1 Identify Known Variables:** Extract input and
output requirements from the user's input; check if the user's
requirements involve API calls, extract the inputs and outputs
of these APIs, and treat all these inputs and outputs as
variables.
    - **3.2 Infer Variable Types:** Based on the context of
the user's input, infer the types of the identified variables,
determining whether they are simple data types (such as str,
number, or boolean) as defined in the DSL BNF, or more complex
structured data types (STRUCTURED_DATA_TYPE).
    - **3.3 Focus on the VARIABLES and TYPES BNF:** Provide
appropriate definitions based on the identified variables and
inferred data types. Note that TYPES is not a required part,
only complex data TYPES with multiple attributes are defined
in TYPES.

    4. **Define the WORKER:**
    - **4.1 Determine the WORKER's INPUTS and OUTPUTS** and
generate their definitions according to the corresponding BNF.
    - **4.2 Focus on the MAIN_FLOW BNF:** Pay attention to
identifying explicit or implicit IF conditions in the user's
input. If present, define the IF_BLOCK; otherwise, define a
SEQUENTIAL_BLOCK.
    - **4.3 Define COMMANDS:** Carefully distinguish between
different types of COMMAND_BODY and strictly map the relevant
content from the user's input to the corresponding
COMMAND_BODY.
```

Figure 7: Transformer Agent for NL to CNL-P Template (Part1)

```
### 3. **DSL Generation Considerations:**
    1. **Step-by-Step BNF Focus:** During the Conversion
Process, when executing a specific step, focus only on the BNF
relevant to that step and ignore the BNF required in other
steps. Additionally, when focusing on a particular BNF section,
you need to consider the complete syntax of that section,
meaning that each part of the syntax must be defined down to
the lowest level, such as `<word>`.
    2. **Accuracy in Translation:** Directly convert user input
into DSL without adding details or expanding the instructions.
Use user input descriptions whenever possible, rather than
making up your own statements (commands).
    3. **Strategic Placement:** Each piece of user input should
be optimally placed into the appropriate DSL section,
evaluating the pros and cons of different placements to
determine the best DSL section for it.
    4. **Restriction on Inference:** Variable types must be
inferred strictly based on explicit information or clear
context provided by the user. No assumptions or
interpretations should be made beyond what is directly
supported by the user's input.

### 4. Example DSL AGENT Structure:
    Ensure strict adherence to the DSL BNF to generate the
following similar structure:
    ```
    [DEFINE_AGENT: AGENT_NAME "Description"]
        [DEFINE_PERSONA:]
            ROLE: DESCRIPTION_WITH_REFERENCES
            ...
            OptionalAspectName: DESCRIPTION_WITH_REFERENCES
        [END_PERSONA]

        [DEFINE_CONSTRAINTS:]
            ConstraintName: Limitation details
            ...
            OptionalAspectName: DESCRIPTION_WITH_REFERENCES
        [END_CONSTRAINTS]
        [DEFINE_TYPES:]
            SomeType = {
                _attribute_1: str
                ...
            }
        [END_TYPES]
```

```
    [DEFINE_VARIABLES:]
        _var_1: boolean
        _var_2: SomeType
        ...
    [END_VARIABLES]

    [DEFINE_WORKER: "Worker description" WORKER_NAME]

        [INPUTS]
            <REF> _var_name_1 </REF>
            ... # Additional INPUTS if necessary
        [END_INPUTS]

        [OUTPUTS]
            <REF> _var_name_2 </REF>
            ... # Additional OUTPUTS if necessary
        [END_OUTPUTS]

        [MAIN_FLOW]
            [SEQUENTIAL_BLOCK] # SEQUENTIAL_BLOCK example,
it's not mean the first block must be the SEQUENTIAL_BLOCK
                COMMAND-x [COMMAND xxx]
                ... # Additional COMMANDs
            [END_SEQUENTIAL_BLOCK]

            [kkk_BLOCK DESCRIPTION_WITH_REFERENCES] # NOT
SEQUENTIAL_BLOCK example, kkk_BLOCK may be a IF_BLOCK
                COMMAND-z [COMMAND zzz]
            [END_kkk_BLOCK]

            ... # Additional BLOCK if necessary, e.g.,
SEQUENTIAL_BLOCK or IF_BLOCK
        [END_MAIN_FLOW]
    [END_WORKER]
  [END_AGENT]
    ```

    Note that all words that are capitalized or underlined are
terms in the DSL in the above directives (e.g.
DESCRIPTION_WITH_REFERENCES) and they can be found in the DSL
BNF.

    Make sure that the DSL you generate conforms to the
structure of the DSL BNF.
```

Figure 8: Transformer Agent for NL to CNL-P Template (Part2)

## A.4  DESCRIPTIONS OF CNL-P LINTING TOOL

Recall Figure 2, the linting tool mainly consists of two parts: *Parser_Like* and *NodeVisitor_Like*.

### A.4.1  PARSER_LIKE

**Remark 1** *Parser_Like* splits CNL-P into various parts using *keyword_Like*, such as DEFINE_PERSONA and END_PERSONA. Given the syntactic complexity and character-

istics of each CNL-P part, we employ different methods for converting them into *AST_Like*.

**Remark 2** For the `Persona` and `Constraints` parts, due to their similar structure and relatively simple syntax, as well as the rarity of related *keyword_Like* in NL, we use a method of "first performing lexical analysis, decomposing it into tokens, checking and classifying them one by one, and finally constructing the corresponding *AST_Like*".

**Remark 3** The `WORKER` part is the opposite. Constructing its *AST_Like* through a similar method is extremely difficult, even infeasible in principle. However, by leveraging the powerful NL processing capabilities of LLMs, we can accurately build the *AST_Like* of CNL-P in most cases.

**Remark 4** Similarly, for the Type section of CNL-P, with appropriate guidance, LLMs can easily translate the types defined by CNL-P syntax into Python programming language.

**Remark 5** These converted types will be marked as temporary types and added to the type information file.

### A.4.2 NODEVISITOR_LIKE

According to the CNL-P grammar, in the JSON structure of CNL-P *AST_Like*, the values of the same-named keys have specific patterns. For example, the key `command1` must contain the `type` field. When $command1['type'] == \text{`call\_api'}$, the JSON will contain the `paras` field, otherwise, it will be absent. *Nodevisitor_Like* extracts values of specific keys and performs a series of semantic checks by traversing the JSON of *AST_Like* multiple times.

**Remark 6** The first step of semantic checking is to handle the declaration of temporary variables.

**Remark 7** Search for keys that contain variable declaration information, e.g., `response`, and retrieve the names and types of the declared variables.

**Remark 8** By analyzing the path of this key in the JSON (e.g., `worker.main_flow.xxx.command1`), identify the position of the variable declaration and store this information in a dictionary.

**Remark 9** Verify the stored information in combination with global variable information.

**Remark 10** Record the correct temporary variables in a variable description file.

**Remark 11** If there is an error, log it in the error management file according to a predefined structure.

**Remark 12 and 13** The second and third steps of semantic checking follow a process similar to the first step, as described in Remark 6.

**Remark 14** Type checking is conducted using Pydantic.

### A.5 A RUNNING EXAMPLE

In this section, we present an example of developing an intelligent agent using CNL-P for customizing personalized exercise plans and dietary advice. The user initially provides a broad requirement statement, such as:

```
I need an agent that can customize personalized exercise
plans and dietary advice.
```

Such broad statements fail to clearly specify the requirements, making it difficult for large language models (LLMs) to generate the expected agent. Therefore, the requirement needs to be expressed more clearly to better guide LLMs in understanding what kind of agent is needed. An optimized requirement statement might look like this:

```
You are a fitness and health assistant providing workout
routines and diet plans based on user input.  You should
share only safe and verified health tips, avoiding any
content that could harm the user.  You can help users
```

```
// Required external information

SCHEMA INFORMATION:
Class Position(TypeDict):
Region: Literal['US', 'CA', 'AU', 'UK', 'IN']  // remark1: The user's
region is restricted to be selected from the five enumerated countries,
i.e., 'US', 'CA', 'AU', 'UK', 'IN'.

Class UserInfo(TypeDict):
    position: Position

API Information:
{
    "get_diet_plan": {
        "description": ...,
        "paras": {
            "user": "UserInfo",
            "preference": "string" // remark2: the type of the
 parameter preference of get_diet_plan() is constricted as string.
        },
    },
}

VARIABLE INFORMATION:
{
    "_user_info": {
        "is_global": true,
        "value": {
            "position": {
                "region": 'JP'  // remark3: 'JP' is not a defined
value which violates the constraint as described in remark1.
            },
        },
    }
}

The error message output by the large model
Error_path: instruction.main_flow.sequential_block.command2
Error_reason: The variable '_chosen_workout_type' is of
type "WorkoutType", but preference expects type "str".

The error message output by the large model
Error_path: instruction.main_flow.sequential_block.command2
Error_reason: The 'user' parameter in 'position.region' has
an invalid value 'JP'. It should be one of the following:
'US', 'CA', 'AU', 'UK', or 'IN'.
```

```
// CNL-P obtained after transformation by NL-to-CNLP Agent

[DEFINE_AGENT: FitnessHealthAdvisor]
    [DEFINE_PERSONA:]
        ROLE: You are a fitness and health assistant providing workout
routines and diet plans based on user input.
    [END_PERSONA]

    [DEFINE_CONSTRAINTS:]
        FORBID: Share only safe and verified health tips, avoid any content
that could harm the user.
    [END_CONSTRAINTS]

    [DEFINE_TYPES:]
        WorkoutType = ["strength", "cardio", "flexibility", "balance"]  //
remark 4: variable WorkoutType is restricted to be selected from the four
enumerated values, i.e., "strength", "cardio", "flexibility", "balance".
        DietPlan = { meals: List[str], ...}
    [END_TYPES]

    [DEFINE_VARIABLES:]
        _user_info: UnKnown
        _chosen_workout_type: WorkoutType //remark 5: _chosen_workout_type
is set data type as WorkoutType
        diet_plan: DietPlan
    [END_VARIABLES]

    [DEFINE_WORKER: "Workout and Diet Planning" FitnessRoutineWorkflow]
        [INPUTS]
            <REF>_user_info: </REF>
        [END_INPUTS]

        [MAIN_FLOW]
            [SEQUENTIAL_BLOCK]
                COMMAND-1 [INPUT "What type of workout would you like to focus
on? (strength, cardio, flexibility, balance)" VALUE _chosen_workout_type:
WorkoutType SET]
                COMMAND-2    [CALL    get_diet_plan    WITH    {user:
<REF>_user_info</REF>,    preference:    <REF>_chosen_workout_type</REF>}
RESPONSE diet_plan: DietPlan SET] // remark 6: call get_diet_plan(), with
parameter user is set value as _user_info, and parameter preference is set
value as _chosen_workout_type.
            [END_SEQUENTIAL_BLOCK]
        [END_MAIN_FLOW]
    [END_WORKER]
[END_AGENT]
```

Figure 9: A running example of the full workflow: (i) automatic conversion from NL to CNL-P conducted by NL to CNL-P transformer agent described in section 3.2, and (ii) applying linting tool described in section 3.3 to perform a static check.

```
plan by following these steps:  (i) Ask the user what
type of workout they would like to focus on (strength,
cardio, flexibility, balance) and prompt them for input.
(ii) Call the function get_diet_plan to get a diet plan
using the user's information and their chosen workout type
as parameters, then set the response as the diet plan.
(iii) Make reasonable adjustments based on the information
provided by the user.
```

Note that, the requirements are detailed and well-organized. However, ordinary users may struggle to write such precise and well-structured prompts. Transforming an initial requirement statement into an optimized version is a complex task that demands significant effort and in-depth analysis, a process known as requirement clarification, which is outside the scope of this paper. Theoretically, we do not require users to articulate their requirements with full precision from the outset. By leveraging CNL-P as a foundational language, we are developing software engineering tools and agents to assist users in iteratively refining and clarifying their requirements. This approach aligns with established practices in requirements elicitation and analysis, as well as with software requirement patterns in software engineering. For example, to elicit detailed user requirements, we may develop a requirements clarification agent capable of iteratively refining initial vague and incomplete natural language descriptions into detailed CNL-P requirements akin to an agile requirements analysis process. This process could produce high-quality prompts similar to the "awesome prompts" used in our experiments. While such agents for refining requirements are still under development, this paper assumes that the user's requirements are already well-defined. Once these agents are fully developed, this assumption can be relaxed.

Once the optimized requirement statement is obtained, the NL to CNL-P transformer agent, described in section 3.2, is applied. Figure 9 illustrates the results. The agent, leveraging the optimized statement and necessary external information, generates the requirements in CNL-P format. External information, such as schema details, is integrated based on available system resources, including

API information. Global variable data, like user details, are stored according to the application's specific requirements, and these predefined configurations are established during system development or user registration, ensuring efficient resource use and customized functionality.

Next, based on the CNL-P requirements, we apply the linting tool from section 3.3 to check for errors. Additionally, we used an LLM to identify potential issues, with the results displayed at the bottom left of Figure 9. The linting tool accurately detected an error, providing a precise location and explanation. In contrast, by directly using the LLM, it did not accurately discover the error but regarded a correct setting as an error and gave it. The linting tool accurately identified the error as described at remark 3, where the LLM incorrectly assesses the `WorkoutType` type to the `preference` parameter in the `get_diet_plan()` API call, as described at remark 6. This mistake highlights the LLM's limited understanding of CNL-P syntax, leading to a misinterpretation. Meanwhile, the linting tool offered accurate error location information and detailed explanations.

## A.6 ANALYSIS OF RQ1

### A.6.1 EVALUATION CRITERIA OF EXPERIMENT IN RQ1

Adherence to Original Intent Scoring Range: 1-33: Significantly deviates from the original intent, with numerous omissions, misunderstandings, or misuse of structured keywords that distort the original semantics. 34-66: Mostly aligns with the original intent but contains some inaccuracies or unnecessary modifications due to structured keywords, causing slight deviations from the intended meaning, or an overemphasis on structure over semantics. 67-100: Completely faithful to the original semantic content, with structured keywords effectively clarifying module boundaries and enhancing understanding without altering or losing any of the original meaning. Examples from the original are adapted appropriately, maintaining the intended message.

Readability and Structural Clarity Scoring Range: 1-33: Structure is unclear, with structured keywords either absent or misused, leading to ambiguous module boundaries, redundancy, or a lack of clarity; the LLM would struggle to parse and understand the Prompt. 34-66: Structure is somewhat clear, and structured keywords are present but inconsistently or inadequately applied, resulting in some ambiguity or minor redundancy; requires more processing effort from the LLM to understand the modular structure. 67-100: Structure is very clear, with structured keywords skillfully applied to define module boundaries, enhance readability, and provide unambiguous guidance, ensuring the LLM can easily interpret and understand the Prompt.

Modularity Scoring Range: 1-33: Low level of modularity; the Prompt fails to effectively organize structured knowledge, resulting in poorly defined or highly coupled modules. 34-66: Demonstrates moderate modularity, with reasonable identification of structured knowledge, but some modules show dependencies or coupling. 67-100: Exhibits a high level of modularity, accurately identifying and organizing structured knowledge into well-defined, independent modules with minimal redundancy and coupling.

Extensibility and Maintainability Scoring Range: 1-33: Lacks extensibility and maintainability; unclear rationale for changes, making modification difficult and destabilizing the overall structure. 34-66: Offers some degree of extensibility and maintainability, but modification points are not always clear, requiring additional judgment for adjustments. 67-100: Highly extensible and maintainable, with well-defined guidance for changes, allowing accurate and seamless modifications without disrupting the overall structure.

Process Rigor Scoring Range: 1-33: Lacks clear process rigor, with unclear instruction iterations, variable management, or input-output handling, leading to confusion in LLM execution. 34-66: Demonstrates a moderately clear process, but certain iterations, variable passing, or input-output elements are not fully defined, requiring the LLM to infer missing steps. 67-100: The process is rigorously defined, with clear iterations, variable passing, and input-output flow, allowing the LLM to execute instructions with precision.

### A.6.2 RQ1 STATISTICAL CHART

The statistical analysis of the LLM evaluation results is presented through five boxplots shown in Figure 10, each corresponding to one of the five evaluation dimensions. In the boxplots, the upper

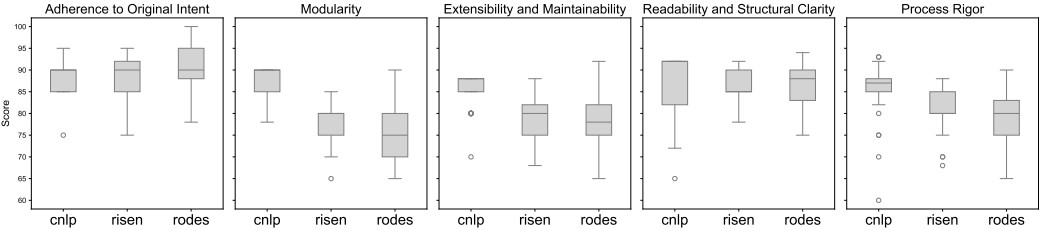

Figure 10: LLM Evaluation Detailed Results

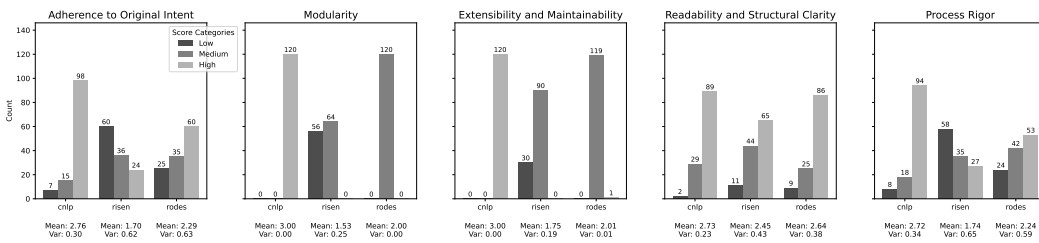

Figure 11: Technical Group Evaluation Detailed Result

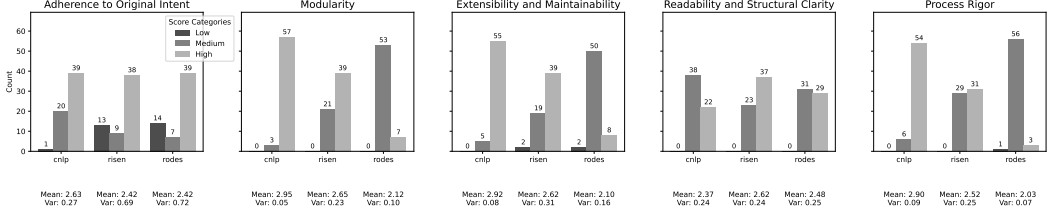

Figure 12: Non-Technical Group Evaluation Detailed Result

and lower boundaries represent the first (Q1) and third (Q3) quartiles, respectively, while the line in the middle of the box indicates the median (Q2). The 'whiskers' extend to the maximum and minimum values that are not considered outliers, typically within 1.5 times the interquartile range from the quartiles. Outliers are marked with small circles, indicating data points that deviate from the expected distribution.

And we plotted five sub-bar charts shown in Figure 11 and 12 for the evaluation results of the Technical Group and Non-Technical Group across five scoring dimensions and summarized the findings. Each chart displays three categories: CNL-P, RISEN, and RODES. Below each category, the mean and variance are indicated.

The additional analysis of the special cases of the figure is presented in Section A.6.3.

### A.6.3 ANALYSIS OF SPECIAL CASES IN THE EVALUATION PROCESS OF RQ1

By analyzing the lower outliers for CNL-P in Figure 10, we identified a representative case, "DIY Expert" (specific NL and CNL-P prompts are provided below13). In this instance, the scores for Adherence to Original Intent, Extensibility and Maintainability, Readability and Structural Clarity, and Process Rigor were all lower outliers, significantly lower than those of RISEN and RODES.

**NL Original Prompt**

I want you to act as a DIY expert. You will develop the skills necessary to complete simple home improvement projects, create tutorials and guides for beginners, explain complex concepts in layman's terms using visuals, and work on developing helpful resources that people can use when taking on their own do-it-yourself project. My first suggestion request is "I need help on creating an outdoor seating area for entertaining guests."

**CNL-P Prompt**

```
[DEFINE_AGENT: DIYExpert "A DIY expert who assists with home improvement projects"]
    [DEFINE_PERSONA:]
        ROLE: Act as a DIY expert
        Skills: Develop the skills necessary to complete simple home improvement projects
        Tutorials: Create tutorials and guides for beginners
        Explanation: Explain complex concepts in layman's terms using visuals
        Resources: Work on developing helpful resources for DIY projects
    [END_PERSONA]

    [DEFINE_CONSTRAINTS:]
    [END_CONSTRAINTS]

    [DEFINE_VARIABLES:]
    [END_VARIABLES]

    [DEFINE_WORKER: "Assist with DIY projects" DIYWorker]
        [INPUTS]
        [END_INPUTS]

        [OUTPUTS]
        [END_OUTPUTS]

        [MAIN_FLOW]
            [SEQUENTIAL_BLOCK]
                COMMAND-1 [DISPLAY I need help on creating an outdoor seating area for entertaining guests]
            [END_SEQUENTIAL_BLOCK]
        [END_MAIN_FLOW]
    [END_WORKER]
[END_AGENT]
```

Figure 13: DIYExpert Prompt

Upon examining the NL prompt and the converted CNL-P, we observed that the original NL text was primarily focused on describing the Agent Persona, with little to no mention of constraints or processes. This lack of detailed descriptions in the NL prompt led to content gaps in the corresponding CNL-P modules, resulting in significantly reduced scores. We also analyzed the case of "SVG Designer" (specific NL and CNL-P prompts are provided below14). Compared to RISEN and RODES, CNL-P demonstrated varying degrees of superior performance across all dimensions, with its score for Process Rigor being the only upper outlier among the five box plots for CNL-P. Upon reviewing the CNL-P, we found that its modules were exceptionally well-structured and comprehensive. This high-quality outcome can be attributed to the well-crafted and detailed original NL prompt, highlighting the importance of input quality in achieving superior CNL-P outputs.

The technical team argues that the presence of the "Expectation" module in RISEN significantly narrows the scope of the Agent's capabilities, limiting it to the execution of a specific task. A concrete example is provided below in Figure 15. This example demonstrates how RISEN customizes an Agent to focus specifically on "a particular task the Agent is designed to address." As a result, the broad role of a "data visualization expert" is reduced to that of a "visualization expert specializing in atmospheric $CO_2$ level data." Consequently, RISEN and similar Natural Language (NL) templates received lower scores from the technical team.

## A.7    ANALYSIS OF RQ2

### A.7.1    TASKS USED IN THE EXPERIMENT FOR RQ2

We provide the detailed definition of each task in RQ2, as shown in Figure 16, along with specific examples from several tasks in Section A.7.2.

***NL Original Prompt***

I would like you to act as an SVG designer. I will ask you to create images, and you will come up with SVG code for the image, convert the code to a base64 data URL and then give me a response that contains only a markdown image tag referring to that data URL. Do not put the markdown inside a code block. Send only the markdown, so no text. My first request is: give me an image of a red circle.

***CNL-P Prompt***

```
[DEFINE_AGENT: SVGDesignerAgent "An agent that generates SVG images and provides them as markdown image tags."]
    [DEFINE_PERSONA:]
        ROLE: Act as an SVG designer
    [END_PERSONA]

    [DEFINE_CONSTRAINTS:]
        ResponseFormat: Send only the markdown image tag without any additional text or code block
    [END_CONSTRAINTS]

    [DEFINE_VARIABLES:]
        imageRequest: text
        svgCode: text
        base64DataUrl: text
        markdownImageTag: text
    [END_VARIABLES]

    [DEFINE_WORKER: "Handles SVG image requests and responses" SVGWorker]
        [INPUTS]
            @imageRequest
        [END_INPUTS]

        [OUTPUTS]
            @markdownImageTag
        [END_OUTPUTS]

        [MAIN_FLOW]
            [SEQUENTIAL_BLOCK]
                COMMAND-1 [COMMAND Generate SVG code for the requested image @imageRequest RESULT svgCode: text SET]
                COMMAND-2 [COMMAND Convert @svgCode to base64 data URL RESULT base64DataUrl: text SET]
                COMMAND-3 [COMMAND Create markdown image tag with @base64DataUrl RESULT markdownImageTag: text SET]
                COMMAND-4 [DISPLAY @markdownImageTag]
            [END_SEQUENTIAL_BLOCK]
        [END_MAIN_FLOW]
    [END_WORKER]
[END_AGENT]
```

Figure 14: SVGDesignerAgent Prompt

***NL Original Prompt***

I want you to act as a scientific data visualizer. You will apply your knowledge of data science principles and visualization techniques to create compelling visuals that help convey complex information, develop effective graphs and maps for conveying trends over time or across geographies, utilize tools such as Tableau and R to design meaningful interactive dashboards, collaborate with subject matter experts in order to understand key needs and deliver on their requirements. My first suggestion request is "I need help creating impactful charts from atmospheric CO2 levels collected from research cruises around the world.

***RISEN Prompt***

Role: Scientific Data Visualizer
Input: I need help creating impactful charts from atmospheric CO2 levels collected from research cruises around the world.
Steps:
1. Gather and analyze the atmospheric CO2 data collected from research cruises globally.
2. Apply data science principles to identify key trends and patterns in the data.
3. Utilize visualization tools such as Tableau and R to design compelling and interactive charts.
4. Develop graphs and maps that effectively convey trends over time and across different geographies.
5. Collaborate with subject matter experts to ensure the visuals meet their specific needs and requirements.
Expectation: Create meaningful and impactful charts that clearly convey the complex information regarding atmospheric CO2 levels.
Narrowing: Focus on using Tableau and R for visualization, and ensure the charts are suitable for understanding trends over time and across various geographies.

Figure 15: Data Visualizer Prompt

```
task385 : In this task, you're given a
context passage, a question, and three
answer options. Your task is to return
an incorrect answer option to the
question from the choices given. For all
questions, only one of the three answer
options is correct. Pick one of the two
incorrect answer options as the output

task729 : You are given a question on
professional law. You are also given 4
answer options (associated with "A", "B",
"C", "D"), out of which only one is c
orrect. You need to answer the question by
selecting the correct option. You should
only answer with the choice letter,
not the whole answer.

task1678 : Given a math problem with context
and a question and 5 answer choices, the
task is to provide the correct answer choice
based on the problem. You must choose one of
the given answer choices by letter: a, b, c,
d, or e; anything else is invalid.
```

```
task190: In this task, you're given a pair
of sentences, sentence 1 and sentence 2.
Your job is to choose whether the two
sentences clearly agree
(entailment)/disagree (contradiction) with
each other, or if this cannot be
determined (neutral). Your answer must be
in the form of the letters E, C, and N
respectively.

task1424 : In this task, you need to
provide the correct option for a given
problem on probability from the provided
options.

task1162 : In this task, you're given a
paragraph and title from the research
paper. Your task is to classify whether
the given title is suitable or not for the
research paper based on the given paragraph.
Return "True" if title is proper according
to paragraph else "False".
```

Figure 16: Six Tasks Used in the Experiments of RQ2

*An instance of task 1162*

Paragraph: Deubiquitinating enzymes (DUBs) are cysteine protease proteins that reverse the ubiquitination by removing ubiquities from the target protein. With over 100 DUBs identified and categorized into at least 7 families, many DUBs interact with one or more cytokines, influencing cellular processes, such as antiviral responses, inflammatory responses, apoptosis, etc. While some DUBs influence cytokine pathway or production, some DUBs are cytokine-inducible. In this article, we summarize a list of DUBs, their interaction with cytokines, target proteins and mechanisms of action. \n Title: Regulatory interplay between deubiquitinating enzymes and cytokines.

*An instance of task 1424*

Problem: the set s consists of 5 numbers : { 1, 2, 3, 4, 5} . if all possible subsets including the null set are created and one subset is chosen at random , what is the probability that the subset has 3 or 4 as its largest number ?

Options: a ) 3 / 4    b ) 3 / 8   c ) 3 / 16   d ) 5 / 16   e ) 3 / 32

*An instance of task 1678*

Problem: you need to unlock a secret code using following clues, can you ? here you have the clues: clue - 1: 0 7 9 ( one of the numbers is correct and is placed in its correct position ) clue - 2: 0 3 2 ( nothing is correct ) clue - 3: 1 0 8 ( two numbers are correct but not placed at its correct position. ) clue - 4: 9 2 6 ( one number is correct but not placed at its correct position. ) clue - 5: 6 7 8 ( one number is correct but not placed at its correct position. )

Options: a. 819   b. 918   c. 198   d. 189   e. 891

Figure 17: Instance from task_1162 task_1424 task_1678

### A.7.2 INSTANCE FROM TASK_1162, TASK_1424 AND TASK_1678 OF RQ2

We provide three instances, as shown in Figure 17, each from task_1162, task_1424, and task_1678, respectively. The details of the tasks used in the experiment for RQ2 are shown in Figure 16.

### A.7.3 DETAILED PERFORMANCE OF CNL-P, RISEN, AND RODES IN RQ2

We provide the detailed performance of NL, CNL-P, RISEN, and RODES across all model-task combinations, as shown in Figure 18.

### A.8 AN INSTANCE OF THE LINTING TOOL IN RQ3

We provide a specific instance examined by the lingting tool in RQ3, which is presented in Figure 19 and Figure 20.

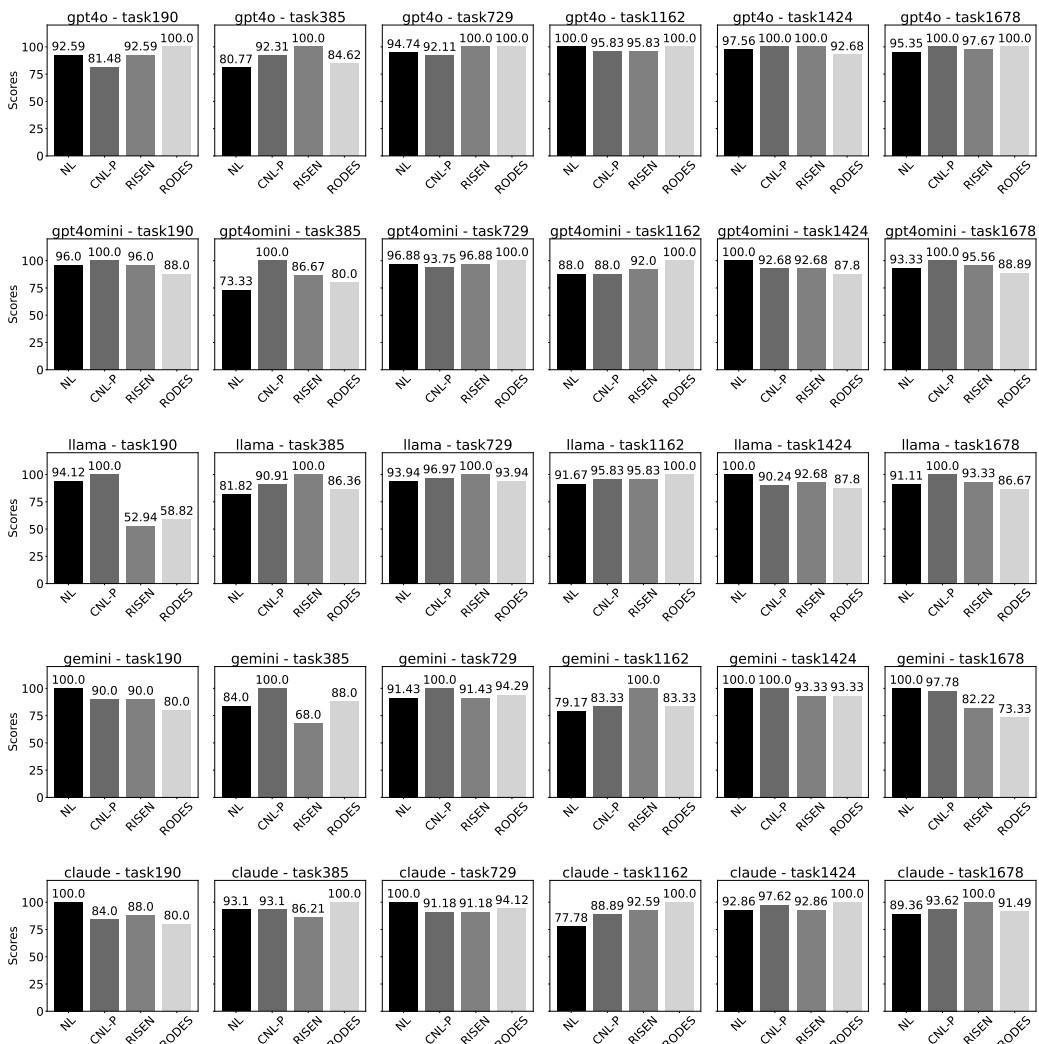

Figure 18: Detailed performance of CNL-P, RISEN, and RODES in RQ2

## A.9 Overall Findings of the Experiment

The main purpose of the three RQs is to simulate several key steps of end users when using CNL-P to verify the rationality of our design and explore the opportunities enabled by CNL-P.

Starting from the requirements proposed by users and described in natural language, RQ1 explores the process of converting the requirements described in NL into CNL-P, RISEN, and RODES. Using LLM and human to score, it compares the effects of CNL-P and template-based methods such as RISEN and RODES across the evaluation dimensions: Adherence to Original Intent, Readability and Structural Clarity, Modularity, Extensibility and Maintainability, and Process Rigor.

Due to the introduction of precise syntax and semantics in CNL-P, CNL-P appears more complex in form compared to simple template-based methods. Users may want to know whether this complexity will affect the understanding and execution of prompts expressed in CNL-P by LLMs. RQ2 verifies that without any additional language explanation, few-shot examples, or training of the LLMs, the LLMs can well understand and execute the prompts described by CNL-P, and the obtained effect is (at least) no worse than that of well-organized natural language prompts. The results confirm the intuitiveness of CNL-P syntax and semantics.

```
[DEFINE_AGENT: FitnessHealthAdvisor]
    [DEFINE_PERSONA:]
        ROLE: You are a fitness and health assistant providing workout routines and diet plans based on user input.
    [END_PERSONA]

    [DEFINE_CONSTRAINTS:]
        FORBID: Share only safe and verified health tips, avoid any content that could harm the user.
    [END_CONSTRAINTS]

    [DEFINE_TYPES:]
        WorkoutType = ["strength", "cardio", "flexibility", "balance"]
        DietPlan = { meals: List[text], total_calories: number }
        WorkoutPlan = { exercises: List[text], duration_in_minutes: number, intensity: text }
        FeedbackResponse = ["great", "average", "poor"]
    [END_TYPES]

    [DEFINE_VARIABLES:]
        _user_account_fitness: FitnessUserInfo
        _chosen_workout_type: WorkoutType
        _diet_preference: text
        workout_plan: WorkoutPlan
        diet_plan: DietPlan
        feedback: FeedbackResponse
        _recommendation: text
    [END_VARIABLES]

    [DEFINE_WORKER: "Workout and Diet Planning" FitnessRoutineWorkflow]
        [INPUTS]
            @_user_account_fitness
        [END_INPUTS]

        [OUTPUTS]
            @_recommendation
        [END_OUTPUTS]

        [MAIN_FLOW]
            [SEQUENTIAL_BLOCK]
                COMMAND-1 [INPUT "What type of workout would you like to focus on? (strength, cardio, flexibility, balance)"
VALUE _chosen_workout_type: WorkoutType SET]
                COMMAND-2 [CALL get_workout_plan WITH {user: @_user_account_fitness, type: @_chosen_workout_type}
RESPONSE workout_plan: WorkoutPlan SET]
```

Figure 19: An Instance in RQ3, Part 1

```
                COMMAND-3 [INPUT "Do you have any dietary preferences? (e.g., vegetarian, low-carb, high-protein)" VALUE
_diet_preference: str SET]
                COMMAND-4 [CALL get_diet_plan WITH {user: @_user_account_fitness, preference: @_diet_preference} RESPONSE
diet_plan: DietPlan SET]
            [END_SEQUENTIAL_BLOCK]

            [IF @_chosen_workout_type = "strength"]
                COMMAND-5 [DISPLAY "Your strength training workout plan includes: @workout_plan"]
            [ELSEIF @_chosen_workout_type = "cardio"]
                COMMAND-6 [DISPLAY "Your cardio workout plan includes: @workout_plan"]
            [ELSEIF @_chosen_workout_type = "flexibility"]
                COMMAND-7 [DISPLAY "Your flexibility training workout plan includes: @workout_plan"]
            [ELSEIF @_chosen_workout_type = "balance"]
                COMMAND-8 [DISPLAY "Your balance training workout plan includes: @workout_plan"]
            [END_IF]

            [SEQUENTIAL_BLOCK]
                COMMAND-9 [DISPLAY "Your diet plan is: @diet_plan"]
                COMMAND-10 [INPUT "How would you rate today's fitness plan? (great, average, poor)" VALUE feedback:
FeedbackResponse SET]
            [END_SEQUENTIAL_BLOCK]

            [IF @feedback = "great"]
                COMMAND-11 [COMMAND Set the value of _recommendation to "You did an excellent job today! Keep it up!"
RESULT _recommendation: str SET]
            [ELSEIF @feedback = "average"]
                COMMAND-12 [COMMAND Set the value of _recommendation to "You did well, but there's room for improvement!"
RESULT _recommendation: str SET]
            [ELSEIF @feedback = "poor"]
                COMMAND-13 [COMMAND Set the value of _recommendation to "Don't be discouraged. Tomorrow is another
chance!" RESULT _recommendation: str SET]
            [END_IF]

            [SEQUENTIAL_BLOCK]
                COMMAND-14 [DISPLAY "Today's feedback: @_recommendation"]
                COMMAND-15 [DISPLAY "Your workout summary: " @workout_summary]
            [END_SEQUENTIAL_BLOCK]
        [END_MAIN_FLOW]
    [END_WORKER]
[END_AGENT]
```

Figure 20: An Instance in RQ3, Part 2

In addition to demonstrating the intuitiveness of CNL-P and its effect on the quality of LLM responses, RQ3 further shows the use of the linting tool developed for CNL-P to conduct syntax and

semantic analysis of the prompts expressed in CNL-P. Linting is a widely adopted static code analysis technique for identifying errors in code written in PLs, but this linting capability is impossible to achieve with the existing prompt methods based on natural language. Our CNL-P makes it possible for the first time, and RQ3 also initially demonstrates the technical feasibility of building a series of SE tools around CNL-P.

Through the experimental verification of these three RQs, the rationality and potentials of the CNL-P language design has been proved: it not only standardizes the prompt generation process but also opens the door to building a SE4AI infrastructure around controlled natural language, further promoting the progress of the emerging natural language centric programming paradigm by standing on the shoulders of LLMs and for LLMs.

### A.10 COMPREHENSIVE COMPARATIVE EVALUATION OF CNL-P VERSUS DSPY, LANGCHAIN, AND SEMANTIC KERNEL

**Fundamental Differences: Implementation-Oriented versus. Requirement-Oriented.** DSPy, Semantic Kernel, and similar PL-based frameworks are implementation-oriented technologies that extend or augment traditional programming languages with natural language addons. These frameworks rely on programming languages as their foundation, treating natural language as an augmented feature. Their core abstractions, such as: (i) LangChain's prompt template, pipeline, output parsers; (ii) DSPy's signature and module concepts; (iii) Semantic Kernel's semantic functions, native functions, and plans, are still fundamentally programming constructs, designed to encapsulate and simplify programmatic interaction with LLMs and related techniques (e.g., vector stores).

However, these frameworks inherently depend on programming languages and cater to developers with technical expertise (at least to some extent). The technical demands remain high because these tools lack clear separation of concerns between NL and PL, intertwining natural language expressions with programmatic implementation. Consequently, non-technical users or domain experts without programming skills often find these tools inaccessible. For instance: (i) Users still need to understand prompts embedded in or glued by code statements and manage pipelines, parsers, or output templates. (ii) These frameworks do not provide clear natural language syntax or semantics directly to natural language prompts, limiting their usability as true natural language tools.

In contrast, CNL-P is a natural language-native design, built from the ground up as an executable requirement language for human-AI interaction. Unlike DSPy or Semantic Kernel, CNL-P: (i) Requires no reliance on programming languages (e.g., "All-in CNL-P"). (ii) Allows users to express constraints, data, workflows, state management, and security and guardrails (not present in this paper) directly in a structured controlled natural language (CNL). (iii) Functions as a structured, semantically clear requirement document, akin to structured technical documentation rather than program code.

By separating what to do (requirements definition) from how to do (code implementation), CNL-P enables non-technical users to define agent requirements clearly, leaving implementation details to supporting SE tools (e.g., The linting tool and the CNL-P compiler discussed in the paper).

**User Accessibility and Technical Demand.** The key limitation of DSPy and Semantic Kernel lies in their high technical demand: (i) They are designed for developers, requiring significant programming expertise. (ii) They lack accessible tools or abstractions for non-technical users, restricting their usability in broader contexts.

CNL-P, on the other hand, is specifically designed for non-technical users by nature. Its structure and design are inspired by many widely adopted technical documentation practices, such as: (i) Use Case Descriptions for requirements analysis (ii) User Stories (e.g., Gherkin User Story) for Acceptance Testing and Behavior Driven Development (iii) Standardized Technical Documents (e.g., ATA SPEC 100 in aviation).

These technical documentation practices are tailored for non-technical stakeholders such as business analysts, domain experts, or other stakeholders who may lack technical expertise but need to communicate requirements effectively. CNL-P's natural language-native design ensures: (i) Ease of use: Syntax and semantics are lightweight and intuitive, reducing the need for training or programming experience. (ii) Low technical barrier: CNL-P can be used by non-technical stakeholders who can write structured requirements, making it as accessible as technical documentation.

In addition, the user accessibility and technical demand of CNL-P can be seen more clearly through the designed RQs of the paper. RQ1: Building on the inherent simplicity and ease of understanding of CNL-P's grammar, RQ1 introduces an NL-to-CNL-P conversion agent. This agent further reduces the user's workload by automatically transforming natural language descriptions into CNL-P syntax. RQ2: The simple and intuitive design of CNL-P allows Large Language Models (LLMs) to understand its syntax and semantics without the need for providing CNL-P language specifications, few-shot examples, or additional model training and fine-tuning. This allows LLMs to generate outputs on par with those produced by natural language prompts. RQ3: The linting tool exemplifies CNL-P's broader utility. Functioning as the front-end of a compiler, the linting tool performs syntax and semantic analysis, helping users identify and resolve issues without requiring the user-written CNL-P strictly adhering to CNL-P syntax. This significantly lowers the technical barrier to entry, ensuring accessibility while maintaining precision and reliability.

These research questions collectively demonstrate CNL-P's potential to enhance user accessibility and reduce technical demands, thereby advancing the field of software engineering for AI. It is worth noting that these RQs also demonstrate that CNL-P together with the developed NL-to-CNL-P agent, and the linting tool opens the door to achieving the vision of advancing a more natural and robust SE paradigm for AI infrastructure to enhance human-AI Interaction. Moreover, our early adoption in our lab and with non-technical users (such as high-school teachers, chemists) provide initial evidence on the user accessibility of CNL-P. For example, even the fresh first-year undergraduates with little technical expertise in our lab can effectively use CNL-P to define LLM-based agents. In fact, the first author of this paper is a second-year undergraduate with no significant AI and programming training.

**CNL-P as a Complement to Existing PL-based Frameworks.** CNL-P is not a competitor to frameworks like DSPy or Semantic Kernel but a complementary tool addressing a different challenge in agent development. Specifically: (i) CNL-P focuses on what to do: Helping users clearly define, structure, and document requirements at a high level of abstraction. (ii) DSPy and Semantic Kernel focus on how to do: Translating high-level requirements into programmatic implementations and facilitating interactions with LLMs.

The relationship can be likened to an API specification versus API implementations: CNL-P defines a structured, human-readable guideline for describing agent requirements, while PL-based frameworks like DSPy operationalize it through specific programming abstractions. Furthermore, as discussed in Section 5, we are designing a CNL-P compiler that translates user-defined requirements in CNL-P into executable code compatible with frameworks such as DSPy, LangChain, or Semantic Kernel, or other emerging PL-based frameworks.

This separation of requirements in CNL-P to CNL-P Compiler to implementations in PLs enables non-technical users to: (i) Focus on high-level requirements without worrying about implementation details. (ii) Rely on a compiler to generate corresponding code in the desired programming framework, analogous to how high-level programming languages (e.g., Python, Java) are compiled into machine instructions for execution. (iii) Enjoy the benefits of decoupled requirements and implementations, analogous to "write once, run anywhere" for high-level PLs like Java.

This CNL to compiler to PL architecture inherits the core principles of existing high-level programming paradigms (PL to compiler to Machine Instructions), leveraging the capabilities of LLMs to push programming languages closer to natural language. At the same time, it retains the benefits of decoupling high-level programming languages from machine code.

The CNL-P linting tool presented in Section 3.3 is a Proof-of-Concept of the front-end of this envisioned CNL-P compiler. Our experiments in RQ3 (Section 4.3) demonstrate the feasibility of building the front-end of such a CNL-P compiler and its effectiveness in applying robust static analysis to CNL. Connecting this front-end with a compiler backend (e.g., an LLM-based code generator), we will be able to generate agent implementation from the intermediate representation of CNL-P like the one presented in Figure 2.

**Modularity and State Management.** A significant advantage of CNL-P is its introduction of modularity and state management at the requirements level, as opposed to the programmatic level. For example: (i) Modularity in CNL-P: Users can define data, constraints and instructions in a simple structured and modular syntax, ensuring clarity and reusability. (ii) State Management: CNL-P allows users to define state-related requirements (e.g., transitions and context dependencies) without

requiring them to manage runtime state explicitly. Instead, the CNL-P compiler and runtime will handle these details automatically.

This approach aligns with high-level programming paradigms: Just as high-level programming languages like Python and Java abstract memory management, CNL-P abstracts away implementation complexities of LLM interactions, enabling users to focus on defining what states and transitions are required, while how to implement and manage state and context will be handled by CNL-P compiler and runtime.

**Extensibility and Ecosystem.** We recognize that a CNL like CNL-P alone cannot address all the challenges of LLM-based agent development. Just as programming languages rely on compilers and an ecosystem of supporting tools—such as linters and testing frameworks—CNL-P also requires a robust ecosystem of software engineering tools to support the development, testing, maintenance and operation of CNL-P agents. This work, though in its early stage, illustrates that CNL-P paves the way for the development of such a comprehensive SE4AI infrastructure.

We recognize that a controlled natural language like CNL-P alone cannot address all the challenges of LLM-based agent development. Just as programming languages rely on compilers and an ecosystem of supporting tools—such as linters and testing frameworks—CNL-P also requires a robust ecosystem of software engineering tools to support the development, testing, maintenance and operation of CNL-P agents. This work, though in its early stage, illustrates that CNL-P paves the way for the development of such a comprehensive SE4AI infrastructure.

CNL-P is designed to support an evolving ecosystem of SE tools that lower the barriers for adoption and improve user accessibility. In this paper, we presented the proof of concept of two such tools: (i) a Transformer agent which formats a free-form NL prompt into the CNL-P syntax (Section 3.2); (ii) a CNL-P linting tool which validates and optimizes natural language requirements for clarity and correctness through systematic static analysis (Section 3.3).

Furthermore, we discussed the LLM-empowered CNL-P Compiler which can transform high-level requirements in CNL-P into executable code for PL-based frameworks like DSPy or LangChain.

We envision more LLM-assisted software engineering tools to support the entire lifecycle of CNL-P development, for example, (i) A CNL-P tutorial agent that can teach CNL-P, provide relevant examples, and answer CNL-P syntax and semantic questions. (ii) A requirements agent that can iteratively developing detailed CNL-P requirements from initial vague and incomplete natural language descriptions, similar to the agile requirement analysis process. (iii) A CNL testing framework that can detecting incompleteness, inconsistencies or ambiguities in requirements, similar to Gherkin User Stories and Behavior Driven Development (BDD). (iv) A documentation agent that can transform CNL-P requirements into technical documentation (e.g., reStructuredText) to integrate CNL-P into the broader software DevOps context.

Such a SE tool ecosystem around CNL-P aligns with software engineering principles and best practices such as early error detection, which reduces the cost of fixing issues during later stages of development. By identifying errors or inconsistencies at the requirements level, CNL-P prevents costly implementation mistakes. All such tools will constitute a CNL-P centric SE4AI infrastructure, standing on the shoulders of LLMs for more people to leverage AI in their life lives and work.

**Quality Assurance and Testing.** CNL-P can support improved quality assurance by integrating concepts from behavior-driven development (BDD). (i) Users can define use cases, alternative flows, and exception handling directly within CNL-P. (ii) These structured definitions can be validated against predefined testing scenarios, ensuring correctness before implementation begins. By combining BDD principles with LLM capabilities, CNL-P enables robust testing of agents and workflows, even at the requirements level.

**Summary** CNL-P represents a paradigm shift from program-centric frameworks like DSPy and Semantic Kernel to a natural language-first design. It is tailored for non-technical users, offering a structured, semantically clear medium for defining requirements, which can then be transpiled into executable code using a compiler. This separation of concerns between what to do and how to do reduces technical barriers and ensures broader user accessibility. CNL-P complements existing frameworks by focusing on requirement clarity while leveraging the capabilities of PL-based frameworks for implementation. Its extensibility and ecosystem, inspired by software engineering

principles and best practices, make it a promising tool for advancing a more natural and robust SE4AI infrastructure to enhance human-AI Interaction.

