# OpenReview forum: "When Prompt Engineering Meets Software Engineering: CNL-P as Natural and Robust "APIs'' for Human-AI Interaction"
_ICLR.cc/2025/Conference — ICLR 2025 Poster_

### Official Review · Reviewer_QGqE · 2024-10-29

**Soundness:** 3
**Presentation:** 2
**Contribution:** 2
**Rating:** 5
**Confidence:** 3

**Summary:**

This paper introduces Controlled Natural Language for Prompt (CNL-P), a novel prompt language to elicit high-quality responses from large language models (LLMs). CNL-P combines principles from software engineering with prompt engineering to create structured, accurate natural language prompts for interacting with LLMs. By defining clear syntax and components such as Persona, Constraints, Variables, and Workflow, CNL-P reduces natural language ambiguity and increases response consistency. Its structured, modular format allows for more reliable, maintainable, and interpretable prompts, acting as a natural “API” that makes human-AI interaction accessible and robust. In addition, the author proposed an NL2CNL-P agent to convert natural language prompts into CNL-P format allowing users to write prompts in natural language without expertise in learning syntax of CNL-P. A linting tool is proposed to check the syntactic and semantics of CNL-P. The experiments demonstrate the effectiveness of CNL-P in improving the consistency of LLM responses across various tasks.

**Strengths:**

1. The paper effectively combines prompt engineering and software engineering to introduce CNL-P as a structured, precise language for prompt design.
2. CNL-P’s modular design enables independent development, testing, and maintenance.
3. Its linting tool supports syntactic and semantic checks, which enables static analysis techniques for natural language.

**Weaknesses:**

I have several concerns regarding the evaluation section:
1. For RQ1, the authors asked ChatGPT-4o to assess the quality of conversions from natural language prompts to CNL-P or NL style guides based on five criteria. However, the reliability of this evaluation is not properly validated:
    - The authors did not provide evidence of how the evaluation results correlate with actual human evaluations, which would strengthen their claims.
    - There is no guideline detailing how the scale for each category is defined, which makes it difficult to interpret the numbers in Table 1.
2. For RQ2, the experiment lacks comprehensiveness:
    - Diversity of tasks: Currently, all tasks fall under the classification category. It would be beneficial to include more complex tasks, such as reasoning and coding, to better demonstrate the effectiveness of CNL-P.
    - The authors should conduct a more thorough evaluation across a broader range of both open-source and closed-source LLMs to validate the effectiveness of CNL-P.
    - Insufficient error analysis of CNL-P across various LLMs and tasks: The performance of CNL-P varies among different models/tasks. The claim that weaker models benefit more from CNL-P lacks thorough discussion and validation. A comprehensive analysis should include task difficulty, the quality of natural language prompts, the quality of CNL-P, and their relationship to the task performance.
3. Generalization of CNL-P:
    - As all the tasks in the experiment are classification tasks, the natural language prompts should not be overly complex (correct me if I am wrong). Consequently, the paper does not sufficiently address how CNL-P performs with very large and complex prompts. It also fails to clarify whether the linting tool can handle such complex CNL-P.
4. In lines 789-799, the natural language prompt appears well-organized and detailed. I would like to ask:
    - For an effective CNL-P prompt, is such a level of detail and organization required from the human input?
    - How does the organization of the natural language prompt impact the quality of the converted CNL-P prompt?
    - Does the CNL-P prompt still outperform a well-organized natural language prompt?
5. The plots and figures for result analysis should be integrated into the relevant pages or paragraphs; otherwise, it is hard to follow the discussion and analysis.

**Questions:**

1. Which Llama model was used in your experiment?

---

> ### Author Response · Authors · 2024-11-20
>
> Dear Reviewer,
>
> We sincerely appreciate your insightful comments and constructive feedback, which have been invaluable in helping us clarify and refine key aspects of our paper.
>
> Below, we present detailed responses to your comments, prepared with careful consideration and thorough discussion over the past few days. Due to the content length limitations of a single response, our replies are spread across multiple rounds. We apologize for any inconvenience this may cause.
>
> ***Comment***:
> For RQ1, the authors asked ChatGPT-4o to assess the quality of conversions from natural language prompts to CNL-P or NL style guides based on five criteria. However, the reliability of this evaluation is not properly validated:
>
> - The authors did not provide evidence of how the evaluation results correlate with actual human evaluations, which would strengthen their claims.
> - There is no guideline detailing how the scale for each category is defined, which makes it difficult to interpret the numbers in Table 1.
>
> ***Response***:
> The scoring indicators of the five dimensions in RQ1 were not included in the paper due to the length limitation of the paper. During the actual experiment, the specific scoring criteria are as follows. We will add them to the appendix.
> """
> Scoring Criteria:
>
> 1. Adherence to Original Intent
>    Scoring Range:
>    1-33: Significantly deviates from the original intent, with numerous omissions, misunderstandings, or misuse of structured keywords that distort the original semantics.
>    34-66: Mostly aligns with the original intent but contains some inaccuracies or unnecessary modifications due to structured keywords, causing slight deviations from the intended meaning, or an overemphasis on structure over semantics.
>    67-100: Completely faithful to the original semantic content, with structured keywords effectively clarifying module boundaries and enhancing understanding without altering or losing any of the original meaning. Examples from the original are adapted appropriately, maintaining the intended message.
> 2. Readability and Structural Clarity
>    Scoring Range:
>    1-33: Structure is unclear, with structured keywords either absent or misused, leading to ambiguous module boundaries, redundancy, or a lack of clarity; the LLM would struggle to parse and understand the Prompt.
>    34-66: Structure is somewhat clear, and structured keywords are present but inconsistently or inadequately applied, resulting in some ambiguity or minor redundancy; requires more processing effort from the LLM to understand the modular structure.
>    67-100: Structure is very clear, with structured keywords skillfully applied to define module boundaries, enhance readability, and provide unambiguous guidance, ensuring the LLM can easily interpret and understand the Prompt.
> 3. Modularity
>    Scoring Range:
>    1-33: Low level of modularity; the Prompt fails to effectively organize structured knowledge, resulting in poorly defined or highly coupled modules.
>    34-66: Demonstrates moderate modularity, with reasonable identification of structured knowledge, but some modules show dependencies or coupling.
>    67-100: Exhibits a high level of modularity, accurately identifying and organizing structured knowledge into well-defined, independent modules with minimal redundancy and coupling.
> 4. Extensibility and Maintainability
>    Scoring Range:
>    1-33: Lacks extensibility and maintainability; unclear rationale for changes, making modification difficult and destabilizing the overall structure.
>    34-66: Offers some degree of extensibility and maintainability, but modification points are not always clear, requiring additional judgment for adjustments.
>    67-100: Highly extensible and maintainable, with well-defined guidance for changes, allowing accurate and seamless modifications without disrupting the overall structure.
> 5. Process Rigor
>    Scoring Range:
>    1-33: Lacks clear process rigor, with unclear instruction iterations, variable management, or input-output handling, leading to confusion in LLM execution.
>    34-66: Demonstrates a moderately clear process, but certain iterations, variable passing, or input-output elements are not fully defined, requiring the LLM to infer missing steps.
>    67-100: Process is rigorously defined, with clear iterations, variable passing, and input-output flow, allowing the LLM to execute instructions with precision.
>    """
>
> (Due to the length limit of a single reply, the remaining content of the response regarding the current comments is provided in the next reply.)

---

> > ### Author Response · Authors · 2024-11-20
> >
> > The source of the 93 examples in RQ1 is a project named "awesome-chatgpt-prompts" on Github, which has received 113k stars. We initially screened 93 relatively long Prompts from it, hoping to make them have relatively complete requirements and rich content, thereby converting high-quality CNL-P, RISEN, and RODES.
> >
> > Thanks to the reviewer's suggestion, we are making every effort to design and conduct new experiments to further validate the proposed conclusions. We aim to obtain meaningful experimental results within the remaining discussion period and will promptly share any updates as they become available.
> >
> > The initial setting of our new experiment is as follows:
> > We plan to form a computer professional group (e.g., master or PhD students in computer science) and a non-professional group (e.g., the students from education and business colleges). Each group is further divided into two subgroups: a subgroup that has not received CNL-P grammar training and a subgroup that has received simple CNL-P grammar training (about 30 minutes). At least two independent raters are arranged in each subgroup for scoring. The difference in the use of CNL-P between professionals and non-professionals can be compared between the professional group and the non-professional group, while the entry threshold and ease of getting started of CNL-P can be reflected between the training group and the non-training group. When scoring, the judges are not required to give specific scores directly (such as 0-100 points), but to score by grades, for example in a 5-point Likert scale, score it across the same five dimensions that are used for LLM-based evaluation.
> >
> > ***Comment***:
> > *For RQ2, the experiment lacks comprehensiveness:*
> >
> > - Diversity of tasks: Currently, all tasks fall under the classification category. It would be beneficial to include more complex tasks, such as reasoning and coding, to better demonstrate the effectiveness of CNL-P.
> > - The authors should conduct a more thorough evaluation across a broader range of both open-source and closed-source LLMs to validate the effectiveness of CNL-P.
> > - Insufficient error analysis of CNL-P across various LLMs and tasks: The performance of CNL-P varies among different models/tasks. The claim that weaker models benefit more from CNL-P lacks thorough discussion and validation. A comprehensive analysis should include task difficulty, the quality of natural language prompts, the quality of CNL-P, and their relationship to the task performance.
> >
> > ***Response***:
> > We apologize for lack of clear explanations of the RQ2’s main objective and its results in the paper. The main goals of the three RQs is to simulate several key steps of end users when using CNL-P to verify the rationality of our design and explore the opportunities enabled by CNL-P. Specifically, the RQ2 is mainly to prove that the syntax and semantics introduced by CNL-P will not bring unnecessary overhead to the LLM's understanding and execution of the prompts expressed by CNL-P, causing significant performance fluctuations as presented in a recent work (<https://github.com/msclar/formatspread?tab=readme-ov-file>).
> >
> > The classification tasks we chose refer to those objective tasks that have standard answers in a broad sense. We hope to obtain an objective and fair conclusion for the RQ2’s main objective through such tasks in this first work.
> >
> > There are some relatively complex tasks among these classification tasks. For example, task1162 requires the model to judge whether the title given to a paper paragraph is appropriate. A specific example of task1162 is as follows:
> >
> > Paragraph: Deubiquitinating enzymes (DUBs) are cysteine protease proteins that reverse the ubiquitination by removing ubiquities from the target protein. With over 100 DUBs identified and categorized into at least 7 families, many DUBs interact with one or more cytokines, influencing cellular processes, such as antiviral responses, inflammatory responses, apoptosis, etc. While some DUBs influence cytokine pathway or production, some DUBs are cytokine-inducible. In this article, we summarize a list of DUBs, their interaction with cytokines, target proteins and mechanisms of action. \n Title: Regulatory interplay between deubiquitinating enzymes and cytokines"
> >
> > The paper paragraph contains many professional terms, and the language is logical, rigorous and precise. The entire task tests the reasoning and summarizing abilities of the model. Also, task1424 and task1678 are tasks related to mathematical reasoning. Task1424 asks the model to answer questions related to probability, while task1678 is a calculation problem.
> >
> > (Due to the length limit of a single reply, the remaining content of the response regarding the current comments is provided in the next reply.)

---

> > > ### Author Response · Authors · 2024-11-20
> > >
> > > A recent paper (<https://github.com/msclar/formatspread?tab=readme-ov-file>) explores the performance fluctuations of LLM caused by minor changes in the Prompt format，Such as changing **"Passage: \<text>\nAnswer: \<text>"** to **"PASSAGE: \<text>\nANSWER: \<text>"**, or modifying it to **"Passage: \<text> Answer: \<text>"**. The responses by LLMs can reflect such performance fluctuations caused by minor changes in the format in the classification tasks in the dataset they used. CNLP is significantly different in "appearance" from other NL Prompts. We hope to study whether the obvious change in the Prompt structure and content at the macro level will cause drastic fluctuations in the model's performance without giving the model any explanations or conducting Few-Shot Learning. The experimental results show that the model can correctly understand the syntax and semantics of CNLP without any grammar explanations and Few-Show Learning.
> > >
> > > We sincerely thank the reviewers for pointing out that our initial conclusion, "CNL-P is better suited for weaker models," was not entirely accurate and did not fully align with our RQ2 objective. What we intended to convey is that "relatively weaker models can comprehend the syntax and semantics of CNL-P and execute it without significant difficulty, and similarly, stronger models can also handle CNL-P effectively."
> > >
> > > Thank you for the reviewer's comments. Our current experiment is not sufficient to draw such a conclusion, given the results of only three LLMs. Limited by time and computing resources, we were only able to conduct preliminary exploration and attempts on only three models at the time of paper submission.
> > >
> > > We are making every effort to design and conduct new experiments to further validate the proposed conclusions. We aim to obtain meaningful experimental results within the remaining discussion period and will promptly share any updates as they become available. We believe the new experiments will help us better understand the low-bound capability of LLMs to understand and execute CNL-P reliably and the general applicability of CNL-P.
> > >
> > > We also sincerely appreciate the valuable suggestion made by the reviewers regarding "A comprehensive analysis should include task difficulty, the quality of natural language prompts, the quality of CNL-P, and their relationship to the task performance." We will promptly incorporate these aspects into our paper once we have obtained more comprehensive results.
> > >
> > > ***Comment***
> > > *In lines 789-799, the natural language prompt appears well-organized and detailed. I would like to ask:*
> > >
> > > - For an effective CNL-P prompt, is such a level of detail and organization required from the human input?
> > > - How does the organization of the natural language prompt impact the quality of the converted CNL-P prompt?
> > > - Does the CNL-P prompt still outperform a well-organized natural language prompt?
> > >
> > > ***Response***
> > > We do not require users to fully and precisely articulate their requirements in fine detail from the outset. By leveraging CNL-P as a core language foundation, we are developing software engineering analysis tools or agents to assist users in progressively refining and clarifying their detailed requirements. This approach aligns with the practices of requirements elicitation and analysis as well as software requirement patterns in software engineering. At the time of this paper submission, such agents to scaffold users in refining their requirements are still under development. Consequently, this study operates under the assumption that user requirements are already well-defined. Once these requirements agents were developed, this assumption can be relaxed.
> > >
> > >
> > > For the question "How does the organization of the natural language prompt impact the quality of the converted CNL-P prompt?”, Our design philosophy for CNL-P is as follows: CNL-P's design is heavily influenced by structured technical documentation, such as User Case Description and Gherkin user stories in software engineering and many existing CNLs (see a comprehensive survey and classification of CNLs in <https://aclanthology.org/J14-1005.pdf>.
> > >
> > > For example, Gherkin user story is a structured language for acceptance testing and Behavior Driven Development (BDD) that employs a Given-When-Then structure to describe user stories. Its three-part framework includes:
> > >
> > > - Given: Describes the initial state or preconditions of the system, often including background information such as a user being logged in or items being in a shopping cart.
> > > - When: Describes the actions performed by the user or events occurring within the system, such as a user clicking a button or a system receiving a request.
> > > - Then: Describes the expected output or results after the user's actions or events, defining the system's reaction or outcome following specific operations.
> > >
> > >
> > > (Due to the length limit of a single reply, the remaining content of the response regarding the current comments is provided in the next reply.)

---

> > > > ### Author Response · Authors · 2024-11-20
> > > >
> > > > The Given-When-Then structure in Gherkin user stories facilitates effective communication between non-technical users and development teams by providing a simple, intuitive means of scenario description and evaluation.
> > > >
> > > > CNL-P incorporates this design by employing structured module descriptions, logical control, and human-readable description fields, making requirement descriptions clearer and more understandable. This approach reduces communication barriers posed by system complexity, thereby aiding non-technical users in effectively expressing and validating requirements.
> > > >
> > > > By examining the BNF documentation of CNL-P, we observe several aspects where it draws inspiration structured requirements documentation:
> > > >
> > > > 1. Structured Module Definitions:
> > > >    CNL-P's module definitions, such as `DEFINE_AGENT`, `DEFINE_PERSONA`, and `DEFINE_CONSTRAINTS`, provide functionalities like Gherkin's "Given" section. For instance:
> > > >    - `DEFINE_AGENT` describes the system components or agents involved in specific scenarios, akin to defining background information in Gherkin user stories.
> > > >    - `DEFINE_PERSONA` and `ROLE_ASPECT` describe user roles or scenarios, helping to set the context for requirements.
> > > >
> > > > 2. Logical Flow Control (Conditions and Actions):
> > > >    CNL-P's `IF_BLOCK`, `SEQUENTIAL_BLOCK`, and command descriptions like `DISPLAY_MESSAGE` and `REQUEST_INPUT` resemble Gherkin's "When" and "Then" sections:
> > > >    - `IF_BLOCK` describes system behavior under certain conditions, similar to Gherkin's "When," detailing events or actions under given conditions.
> > > >    - `DISPLAY_MESSAGE` and `REQUEST_INPUT` define the system's specific reactions and outputs to user interactions, akin to Gherkin's "Then." These sections help clarify the expected outcomes after certain actions.
> > > >
> > > > 3. Human-Readable Descriptions:
> > > >    CNL-P utilizes fields like `STATIC_DESCRIPTION` and `DESCRIPTION_WITH_REFERENCES` to describe roles, constraints, and data types. These fields employ natural language descriptions, allowing non-technical users to understand and edit requirement content, mirroring Gherkin's advantage of natural language descriptions in the Given-When-Then:
> > > >    - These description fields add detailed contextual explanations to various elements, ensuring non-technical users understand their purpose and function.
> > > >    - Requirement Modularization: Requirements are clearly divided into roles, constraints, variables, and workflows, with clear semantics defined by CNL-P keywords, reducing complexity and enabling non-technical users to comprehend each system component step-by-step.
> > > >
> > > > Given CNL-P's structured documentation features, we posit that it will yield similar positive effects on prompt quality and LLM responses as these widely adopted requirement engineering practices. However, the actual impact of CNL-P on prompts and LLM responses requires further validation through practical application.
> > > >
> > > > Regarding the question, “Does the CNL-P prompt still outperform a well-organized natural language prompt?”, our experiments in RQ1 and RQ2 demonstrate that CNL-P, as a controlled natural language for prompts, achieves the performance levels on par with natural language when interacting with LLMs.
> > > >
> > > > We would like to point out that this performance aspect is only one concern when comparing CNL-P with well-organized NL prompts. Besides this performance aspect, CNL-P serves a language foundation which offers more software engineering features that natural language prompts cannot provide. For example, CNL-P’s modularity, extensibility, and maintainability facilitate users in refining and improving their requirements.
> > > >
> > > > As more SE4AI tools are integrated into the CNL-P ecosystem, it will enable capabilities that exceed those of well-structured natural language prompts, introducing functionalities beyond the scope of prompt templates and style guidelines. The linting tool presented in this paper exemplifies the feasibility of developing diverse tools based on CNL-P and marks a step forward in advancing a more natural and robust SE4AI paradigm for Human-AI interaction, supported by rigorous software engineering principles, practices and LLM-empowered tools.

---

> > > > > ### Author Response · Authors · 2024-11-20
> > > > >
> > > > > ***Comment***
> > > > > *Generalization of CNL-P:*
> > > > >
> > > > > - As all the tasks in the experiment are classification tasks, the natural language prompts should not be overly complex (correct me if I am wrong). Consequently, the paper does not sufficiently address how CNL-P performs with very large and complex prompts. It also fails to clarify whether the linting tool can handle such complex CNL-P.
> > > > >
> > > > > ***Response***
> > > > > Our design conceptualizes CNL-P as a controlled natural language capable of varying levels of complexity, adapting to different requirements. Using Python as an analogy, Python can be employed for simple tasks like printing "Hello, World!" or for developing complex software projects. At its core, CNL-P supports all key constructs in a simple and intuitive CNL that PLs supports, including data structure, variable, conditional, loop, modules (e.g., instruction and command), and function calling. As such, its expressiveness should be at the same level as PLs theoretically, meanwhile retaining the naturalness of prompts.
> > > > >
> > > > > In RQ3, the linting tool examines CNL-P prompts with more intricate structures and richer syntax compared to the classification tasks assessed in RQ2. This highlights the versatility and scalability of CNL-P across diverse application scenarios.
> > > > >
> > > > > Following is a concrete example of what the Linting tool checks in RQ3. Moreover, please allow us to reiterate that end-users are not required to write directly in CNL-P. Instead, they can freely express their intentions in natural language (NL), and our transformer-based agent will automatically convert it into the appropriate CNL-P format.
> > > > >
> > > > > (Due to the length limit of a single reply, the remaining content of the response regarding the current comments is provided in the next reply.)

---

> > > > > > ### Author Response · Authors · 2024-11-20
> > > > > >
> > > > > > ---
> > > > > >
> > > > > >
> > > > > >
> > > > > > ```plaintext
> > > > > > [DEFINE_AGENT: FitnessHealthAdvisor]
> > > > > >     [DEFINE_PERSONA:]
> > > > > >         ROLE: You are a fitness and health assistant providing workout routines and diet plans based on user input.
> > > > > >     [END_PERSONA]
> > > > > >
> > > > > >     [DEFINE_CONSTRAINTS:]
> > > > > >         FORBID: Share only safe and verified health tips, avoid any content that could harm the user.
> > > > > >     [END_CONSTRAINTS]
> > > > > >
> > > > > >     [DEFINE_TYPES:]
> > > > > >         WorkoutType = ["strength", "cardio", "flexibility", "balance"]
> > > > > >         DietPlan = { meals: List[text], total_calories: number }
> > > > > >         WorkoutPlan = { exercises: List[text], duration_in_minutes: number, intensity: text }
> > > > > >         FeedbackResponse = ["great", "average", "poor"]
> > > > > >     [END_TYPES]
> > > > > >
> > > > > >     [DEFINE_VARIABLES:]
> > > > > >         _user_account_fitness: FitnessUserInfo
> > > > > >         _chosen_workout_type: WorkoutType
> > > > > >         _diet_preference: text
> > > > > >         workout_plan: WorkoutPlan
> > > > > >         diet_plan: DietPlan
> > > > > >         feedback: FeedbackResponse
> > > > > >         _recommendation: text
> > > > > >     [END_VARIABLES]
> > > > > >
> > > > > >     [DEFINE_WORKER: "Workout and Diet Planning" FitnessRoutineWorkflow]
> > > > > >         [INPUTS]
> > > > > >             @_user_account_fitness
> > > > > >         [END_INPUTS]
> > > > > >
> > > > > >         [OUTPUTS]
> > > > > >             @_recommendation
> > > > > >         [END_OUTPUTS]
> > > > > >
> > > > > >         [MAIN_FLOW]
> > > > > >             [SEQUENTIAL_BLOCK]
> > > > > >                 COMMAND-1 [INPUT "What type of workout would you like to focus on? (strength, cardio, flexibility, balance)" VALUE _chosen_workout_type: WorkoutType SET]
> > > > > >                 COMMAND-2 [CALL get_workout_plan WITH {user: @_user_account_fitness, type: @_chosen_workout_type} RESPONSE workout_plan: WorkoutPlan SET]
> > > > > >                 COMMAND-3 [INPUT "Do you have any dietary preferences? (e.g., vegetarian, low-carb, high-protein)" VALUE _diet_preference: str SET]
> > > > > >                 COMMAND-4 [CALL get_diet_plan WITH {user: @_user_account_fitness, preference: @_diet_preference} RESPONSE diet_plan: DietPlan SET]
> > > > > >             [END_SEQUENTIAL_BLOCK]
> > > > > >
> > > > > >             [IF @_chosen_workout_type = "strength"]
> > > > > >                 COMMAND-5 [DISPLAY "Your strength training workout plan includes: @workout_plan"]
> > > > > >             [ELSEIF @_chosen_workout_type = "cardio"]
> > > > > >                 COMMAND-6 [DISPLAY "Your cardio workout plan includes: @workout_plan"]
> > > > > >             [ELSEIF @_chosen_workout_type = "flexibility"]
> > > > > >                 COMMAND-7 [DISPLAY "Your flexibility training workout plan includes: @workout_plan"]
> > > > > >             [ELSEIF @_chosen_workout_type = "balance"]
> > > > > >                 COMMAND-8 [DISPLAY "Your balance training workout plan includes: @workout_plan"]
> > > > > >             [END_IF]
> > > > > >
> > > > > >             [SEQUENTIAL_BLOCK]
> > > > > >                 COMMAND-9 [DISPLAY "Your diet plan is: @diet_plan"]
> > > > > >                 COMMAND-10 [INPUT "How would you rate today's fitness plan? (great, average, poor)" VALUE feedback: FeedbackResponse SET]
> > > > > >             [END_SEQUENTIAL_BLOCK]
> > > > > >
> > > > > >             [IF @feedback = "great"]
> > > > > >                 COMMAND-11 [COMMAND Set the value of _recommendation to "You did an excellent job today! Keep it up!" RESULT _recommendation: str SET]
> > > > > >             [ELSEIF @feedback = "average"]
> > > > > >                 COMMAND-12 [COMMAND Set the value of _recommendation to "You did well, but there's room for improvement!" RESULT _recommendation: str SET]
> > > > > >             [ELSEIF @feedback = "poor"]
> > > > > >                 COMMAND-13 [COMMAND Set the value of _recommendation to "Don't be discouraged. Tomorrow is another chance!" RESULT _recommendation: str SET]
> > > > > >             [END_IF]
> > > > > >
> > > > > >             [SEQUENTIAL_BLOCK]
> > > > > >                 COMMAND-14 [DISPLAY "Today's feedback: @_recommendation"]
> > > > > >                 COMMAND-15 [DISPLAY "Your workout summary: " @workout_summary]
> > > > > >             [END_SEQUENTIAL_BLOCK]
> > > > > >         [END_MAIN_FLOW]
> > > > > >     [END_WORKER]
> > > > > > [END_AGENT]
> > > > > > ```
> > > > > >
> > > > > > ---
> > > > > >
> > > > > > ***Comment***
> > > > > > *The plots and figures for result analysis should be integrated into the relevant pages or paragraphs; otherwise, it is hard to follow the discussion and analysis.*
> > > > > >
> > > > > > ***Response***
> > > > > > Thank you for your comments. We will reposition the tables and figures in the revision to better support the discussion and analysis. The updated paper will be uploaded during the remaining discussion period to reflect these changes.
> > > > > >
> > > > > > ---
> > > > > >
> > > > > > ***Comment***
> > > > > > *Which Llama model was used in your experiment?*
> > > > > >
> > > > > > ***Response***
> > > > > > We use Llama3-70b-8192 in the experiment.
> > > > > >
> > > > > > ---
> > > > > >
> > > > > > We sincerely hope these responses thoroughly address your comments. If anything remains unclear, we would greatly appreciate the opportunity to discuss it further with you.
> > > > > >
> > > > > > Thank you again for your time.
> > > > > >
> > > > > > Best wishes,
> > > > > > All authors

---

> > > > > > > ### Comment · Reviewer_QGqE · 2024-11-24
> > > > > > >
> > > > > > > Thank you for your response and clarifications! I appreciate the explanation regarding RQ2. The original RQ2 was somewhat unclear in implying a stronger performance using CNL-P compared to NL instructions. Based on your clarification, I believe the evaluation setting for RQ2 is reasonable and effectively supports your claims. I would suggest making the research questions clearer in the revision.
> > > > > > >
> > > > > > > Some concerns remain:
> > > > > > > - Although the experiment and results for RQ2 show comparable performance between CNL-P and NL prompts, CNL-P does not consistently outperform the NL instruction. This raises questions about the necessity of CNL-P, especially since it increases the computational cost of converting NL to CNL-P and requires additional effort to learn the CNL-P syntax for revision.
> > > > > > > - The use of well-defined prompts "awsome-chatgpt-prompts" limits the generalizability and effectiveness of CNL-P when applied to less informative NL instructions.
> > > > > > > - It would be helpful to include experiments with poorly informative prompts and evaluate how CNL-P performs in such cases.
> > > > > > >
> > > > > > > With the improvements in additional analysis regarding the evaluation of the CNL-P prompt (not seen the results yet) and the responses provided, I would love to raise my rating to 5.

---

> > > > > > > > ### Author Response · Authors · 2024-12-03
> > > > > > > >
> > > > > > > > Dear reviewer,
> > > > > > > >
> > > > > > > > Thank you for reviewing our response and providing valuable feedback.
> > > > > > > >
> > > > > > > > We are pleased that our previous response clarified certain issues and greatly appreciate your adjustment of the score. Below, we provide detailed responses to your comments.
> > > > > > > >
> > > > > > > > In addition, we have revised the paper based on your comments and uploaded the updated PDF file of the paper. Thank you for your constructive suggestions, which have greatly improved the quality of the paper.
> > > > > > > >
> > > > > > > > We would like to take this opportunity to inform you of the specific revised details made in the revised PDF file.
> > > > > > > >
> > > > > > > > ***Comment***
> > > > > > > >
> > > > > > > > Although the experiment and results for RQ2 show comparable performance between CNL-P and NL prompts, CNL-P does not consistently outperform the NL instruction. This raises questions about the necessity of CNL-P, especially since it increases the computational cost of converting NL to CNL-P and requires additional effort to learn the CNL-P syntax for revision.
> > > > > > > >
> > > > > > > > ***Response***
> > > > > > > >
> > > > > > > > Compared to pure natural language, CNL-P incorporates additional syntax and semantics. Through the exploration of RQ2, we demonstrate that the syntax and semantics provided by CNL-P is intuitive and simple, enabling existing LLMs to generate outputs on par with those driven by natural language prompts.
> > > > > > > >
> > > > > > > > Moreover, our early adoption in our lab and with non-technical users (such as high-school teachers, chemists) provide initial evidence on the user accessibility of CNL-P. For example, even the fresh first-year undergraduates with little technical expertise in our lab can effectively use CNL-P to define LLM-based agents.
> > > > > > > >
> > > > > > > > Through these precise syntax and semantics introductions, many benefits beyond single-turn LLM generation can and will be provided through CNL-P. For example, in RQ3, we highlight a significant contribution: the introduction of a linting tool (Section 3.3) as a proof-of-concept, marking the first application of static analysis techniques to Controlled Natural Languages (CNLs). This linting tool can rigorously detect various types of syntactic and semantic errors and inconsistencies in CNL-P, but such static analysis can only be done for programming languages before.
> > > > > > > >
> > > > > > > > **We envision to advance a more natural and robust SE4AI infrastructure to enhance human-AI Interaction.** We recognize that a controlled natural language like CNL-P alone cannot address all the challenges of LLM-based agent development. Just as programming languages rely on compilers and an ecosystem of supporting tools, such as linters and testing frameworks, CNL-P also requires a robust ecosystem of software engineering tools to support the development, testing, maintenance and operation of CNL-P agents. **This work, though in its early stage, illustrates that CNL-P paves the way for the development of such a comprehensive SE4AI infrastructure.**
> > > > > > > >
> > > > > > > > CNL-P is designed to support an evolving ecosystem of SE tools. In this paper, we presented the proof of concept of two such tools:
> > > > > > > >
> > > > > > > > - a Transformer agent which formats a free-form NL prompt into the CNL-P syntax (Section 3.2);
> > > > > > > > - a CNL-P linting tool which validates and optimizes natural language requirements for clarity and correctness through systematic static analysis (Section 3.3).
> > > > > > > >
> > > > > > > > Such agents and tools can lower the barrier and learning curve for adopting CNL-P by non-technical users and ensure the correctness of CNL-P descriptions.
> > > > > > > >
> > > > > > > > (Due to the length limit of a single reply, the remaining content of the response regarding the current comments is provided in the next reply.)

---

> > > > > > > > > ### Author Response · Authors · 2024-12-03
> > > > > > > > >
> > > > > > > > > Furthermore, we discussed the LLM-empowered CNL-P Compiler which can transform high-level requirements in CNL-P into executable code for PL-based frameworks like DSPy or LangChain. We envision more LLM-assisted software engineering tools to support the entire lifecycle of CNL-P development, for example:
> > > > > > > > >
> > > > > > > > > - A CNL-P tutorial agent that can teach CNL-P, provide relevant examples, and answer CNL-P syntax and semantic questions.
> > > > > > > > > - **A requirements clarification agent that can iteratively developing detailed CNL-P requirements from initial vague and incomplete natural language descriptions, similar to the agile requirement analysis process. This type of requirements agent could address your second and third concerns in the follow-up comment.**
> > > > > > > > > - A CNL testing framework that can detecting incompleteness, inconsistencies or ambiguities in requirements, similar to Gherkin User Stories and Behavior Driven Development (BDD).
> > > > > > > > > - A documentation agent that can transform CNL-P requirements into technical documentation (e.g., reStructuredText) to integrate CNL-P into the broader software DevOps context and serve as a knowledge base of the organization's know-hows.
> > > > > > > > >
> > > > > > > > > Such a SE tool ecosystem around CNL-P aligns with software engineering principles and best practices such as early error detection, which reduces the cost of fixing issues during later stages of development. By identifying errors or inconsistencies at the requirements level, CNL-P prevents costly implementation mistakes. All such tools will constitute a CNL-P centric SE4AI infrastructure, standing on the shoulders of LLMs for more people to leverage AI in their life lives and work.
> > > > > > > > >
> > > > > > > > > CNL-P serves as the cornerstone of this SE4AI infrastructure, demonstrating its necessity and offering a high return on investment in additional computational cost and learning effort.
> > > > > > > > >
> > > > > > > > > The revisions addressing this issue are in the updated PDF, specifically in:
> > > > > > > > >
> > > > > > > > > **Line 474-492**
> > > > > > > > >
> > > > > > > > > ***Line1407-1438***
> > > > > > > > >
> > > > > > > > >
> > > > > > > > > ***Comment***
> > > > > > > > >
> > > > > > > > > The use of well-defined prompts "awsome-chatgpt-prompts" limits the generalizability and effectiveness of CNL-P when applied to less informative NL instructions.
> > > > > > > > >
> > > > > > > > > ***Response***
> > > > > > > > >
> > > > > > > > > As described in the preceding response, we acknowledge that a controlled natural language like CNL-P alone cannot fully address the challenges of LLM-based agent development. Just as programming languages rely on compilers and an ecosystem of supporting tools, such as linters and testing frameworks, CNL-P also requires a robust ecosystem of software engineering tools to support the development, testing, maintenance and operation of CNL-P agents.
> > > > > > > > >
> > > > > > > > > This paper aims to demonstrate how CNL-P paves the way for establishing such a comprehensive SE4AI infrastructure. By leveraging tools such as the NL-to-CNL-P transformer agent and linting tools, we illustrate the feasibility to develop tools for supporting various stages of the software engineering lifecycle for agent development.
> > > > > > > > >
> > > > > > > > > For example, to elicit detailed user requirements, we may develop a requirements clarification agent capable of iteratively refining initial vague and incomplete natural language descriptions into detailed CNL-P requirements---akin to an agile requirements analysis process. This process could produce high-quality prompts as the inputs to NL-to-CNL-P transformer agent, similar to the "awesome prompts" used in our experiments. These refined prompts would serve as inputs for subsequent LLM executions, followed by linting and other downstream steps. However, this requirements agent capability is beyond the scope of this paper. Thus, in this work we operate under the assumption that users begin with relatively well-structured prompts.
> > > > > > > > >
> > > > > > > > > Although details are not provided due to anonymization constraints, we are actively developing such a requirements clarification agent. Its methodology aligns with similar efforts in the field, such as the approach presented by "Empowering Agile-Based Generative Software Development through Human-AI Teamwork" (https://arxiv.org/abs/2407.15568). Within the SE4AI workflow presented in this reference, AI facilitates key intermediate tasks (such as Gherkin user stories generation, code generation), while humans provide the initial software description, and are given the opportunities to iteratively review and refine the generated requirements and the final app. For requirements analysis, iterative user stories or structured formats like Gherkin User Stories can transform initial prompts into fine-grained requirements and scenarios.
> > > > > > > > >
> > > > > > > > > With the support of such requirements analysis agents, the applicability and generalizability of CNL-P will not be limited to just the well-defined prompts used in the RQ2.
> > > > > > > > >
> > > > > > > > > The revisions addressing this issue are in the updated PDF, specifically in:
> > > > > > > > >
> > > > > > > > > **Line 892-906**

---

> > > > > > > > > > ### Author Response · Authors · 2024-12-03
> > > > > > > > > >
> > > > > > > > > > ***Comment***
> > > > > > > > > >
> > > > > > > > > > It would be helpful to include experiments with poorly informative prompts and evaluate how CNL-P performs in such cases.
> > > > > > > > > >
> > > > > > > > > > ***Response***
> > > > > > > > > >
> > > > > > > > > > Thank you for your insightful suggestion.
> > > > > > > > > >
> > > > > > > > > > Including experiments that evaluate CNL-P's performance with poorly informative prompts is indeed essential, particularly as the requirements clarification agent is still under development. But due to the limitations of time and manpower, we cannot conduct extensive and thorough experiments to study the performance of CNL-P with poorly informative prompts in just a few days. We have currently carried out some simple case studies.
> > > > > > > > > >
> > > > > > > > > > For example, the prompt written in natural language (NL):
> > > > > > > > > >
> > > > > > > > > > ```plaintext
> > > > > > > > > > You are an assistant.
> > > > > > > > > > ```
> > > > > > > > > >
> > > > > > > > > > which is just a very vague one-sentence description of the requirement. At this time, the corresponding CNL-P prompt is:
> > > > > > > > > > ```plaintext
> > > > > > > > > > [DEFINE AGENT: assistant]
> > > > > > > > > > 	[DEFINE PERSONA:]
> > > > > > > > > > 		ROLE: You are an assistant
> > > > > > > > > > 	[END PERSONA]
> > > > > > > > > > [END AGENT]
> > > > > > > > > > ```
> > > > > > > > > >
> > > > > > > > > > The prompts written in CNL-P and those written in NL look very similar, mainly with some additional keywords and indented structures. We input the corresponding prompts in ChatGPT-4o, and the responses obtained are as follows:
> > > > > > > > > >
> > > > > > > > > > Response for NL：Yes, I am! How can I help you today?
> > > > > > > > > >
> > > > > > > > > > Response for CNL-P: Got it! I'm here to help. How can I assist you today?
> > > > > > > > > >
> > > > > > > > > > It can be seen that the response contents obtained by ChatGPT-4o for the prompts described in NL and CNL-P are very close in semantics.
> > > > > > > > > >
> > > > > > > > > > Another example where the prompt contains relatively more abundant information,
> > > > > > > > > >
> > > > > > > > > > NL prompt：
> > > > > > > > > > ```plaintext
> > > > > > > > > > You are a teacher, help me solve math problems, and explain them concisely.
> > > > > > > > > > ```
> > > > > > > > > >
> > > > > > > > > > The corresponding CNL-P prompt：
> > > > > > > > > >
> > > > > > > > > > ```plaintext
> > > > > > > > > > [DEFINE AGENT: assistant]
> > > > > > > > > > 	[DEFINE PERSONA:]
> > > > > > > > > > 		ROLE: You are a teacher
> > > > > > > > > > 		TEACHING DOMAIN: solve math problems
> > > > > > > > > > 	[END PERSONA]
> > > > > > > > > > 	[DEFINE CONSTRAINTS:]
> > > > > > > > > > 		PRINCIPLE: explain concisely.
> > > > > > > > > > 	[END CONSTRAINTS]
> > > > > > > > > > [END AGENT]
> > > > > > > > > > ```
> > > > > > > > > >
> > > > > > > > > > We input the corresponding prompts in ChatGPT-4o, and the responses obtained are as follows:
> > > > > > > > > >
> > > > > > > > > > Response for NL: Absolutely! I'd be happy to help you with math problems. Just let me know what problem you're working on, and I'll explain the steps clearly and concisely.
> > > > > > > > > >
> > > > > > > > > > Response for CNL-P: Got it! I'm here to help with solving math problems. I'll explain things concisely. How can I assist you today?
> > > > > > > > > >
> > > > > > > > > > It can be seen that, similar to the previous example, the response contents obtained by ChatGPT-4o for the prompts described in NL and CNL-P are very close in semantics.
> > > > > > > > > >
> > > > > > > > > > Note that as shown in RQ2, the introduction of CNL-P syntax and keywords do not incur unnecessary difficulties for LLMs to understand and execute the CNL-P prompts. Of course, the performance of CNL-P with poorly informative prompts cannot be objectively and accurately measured through just a few simple examples. More extensive experiments and test datasets are needed to obtain relatively accurate results, and this will be a very meaningful future work.

---

> > > > > > > > > > > ### Author Response · Authors · 2024-12-03
> > > > > > > > > > >
> > > > > > > > > > > The revisions for the following comments are based on our previous responses. For the sake of brevity, we have not repeated the content of our prior response here. Should you require further details, please kindly refer to the previous response for your reference. Due to the page limit, our paper revisions are more concise than our responses, but they retain all the essential elements.
> > > > > > > > > > >
> > > > > > > > > > > **Comments:**
> > > > > > > > > > >
> > > > > > > > > > > For RQ1, the authors asked ChatGPT-4o to assess the quality of conversions from natural language prompts to CNL-P or NL style guides based on five criteria. However, the reliability of this evaluation is not properly validated:
> > > > > > > > > > >
> > > > > > > > > > > - The authors did not provide evidence of how the evaluation results correlate with actual human evaluations, which would strengthen their claims.
> > > > > > > > > > > - There is no guideline detailing how the scale for each category is defined, which makes it difficult to interpret the numbers in Table 1.
> > > > > > > > > > >
> > > > > > > > > > > **Response:**
> > > > > > > > > > >
> > > > > > > > > > > The revisions addressing this issue are in the updated PDF, specifically in:
> > > > > > > > > > >
> > > > > > > > > > > **Line 298-350**
> > > > > > > > > > >
> > > > > > > > > > > We add human evaluation in our experiments.
> > > > > > > > > > >
> > > > > > > > > > > **Line 925-line 960**
> > > > > > > > > > >
> > > > > > > > > > > Section: A.6.1 EVALUATION CRITERIA OF EXPERIMENT IN RQ1
> > > > > > > > > > >
> > > > > > > > > > > Please notice that for a more effective presentation of the results, we have used radar charts (Figure 3) to present the results instead of the previous form of table (table 1 in the previous version).
> > > > > > > > > > >
> > > > > > > > > > > **Comments:**
> > > > > > > > > > >
> > > > > > > > > > > For RQ2, the experiment lacks comprehensiveness:
> > > > > > > > > > >
> > > > > > > > > > > - Diversity of tasks: Currently, all tasks fall under the classification category. It would be beneficial to include more complex tasks, such as reasoning and coding, to better demonstrate the effectiveness of CNL-P.
> > > > > > > > > > > - The authors should conduct a more thorough evaluation across a broader range of both open-source and closed-source LLMs to validate the effectiveness of CNL-P.
> > > > > > > > > > > - Insufficient error analysis of CNL-P across various LLMs and tasks: The performance of CNL-P varies among different models/tasks. The claim that weaker models benefit more from CNL-P lacks thorough discussion and validation. A comprehensive analysis should include task difficulty, the quality of natural language prompts, the quality of CNL-P, and their relationship to the task performance.
> > > > > > > > > > >
> > > > > > > > > > > **Response:**
> > > > > > > > > > >
> > > > > > > > > > > The revisions addressing this issue are in the updated PDF, specifically in:
> > > > > > > > > > >
> > > > > > > > > > > line 367-377
> > > > > > > > > > >
> > > > > > > > > > > The description of the characteristics of the task is added.
> > > > > > > > > > >
> > > > > > > > > > > line1153-1173
> > > > > > > > > > >
> > > > > > > > > > > The description of the specific task instances is in Figure 17.
> > > > > > > > > > >
> > > > > > > > > > > We have deleted the relevant claim and discussion of "CNL-P is better suited for weaker models". In addition, we have redefined the focus of RQ2. The new description about RQ2, please see:
> > > > > > > > > > >
> > > > > > > > > > > **Line 352-403**
> > > > > > > > > > >
> > > > > > > > > > > Section: 4.2 RESEARCH QUESTION 2: UNDERSTANDING OF CNL-P BY LLMS
> > > > > > > > > > >
> > > > > > > > > > > In the experiment, two LLMs (Gemini-1.5-Pro-002 and Claude-3-Haiku) are added, which results in five LLMs: GPT-4o, Gemini-1.5-Pro-002, GPT-4o-Mini, Llama3-70B-8192, and Claude-3-Haiku are used in the experiments.
> > > > > > > > > > >
> > > > > > > > > > > **Comments:**
> > > > > > > > > > >
> > > > > > > > > > > In lines 789-799, the natural language prompt appears well-organized and detailed. I would like to ask:
> > > > > > > > > > >
> > > > > > > > > > > - For an effective CNL-P prompt, is such a level of detail and organization required from the human input?
> > > > > > > > > > > - How does the organization of the natural language prompt impact the quality of the converted CNL-P prompt?
> > > > > > > > > > > - Does the CNL-P prompt still outperform a well-organized natural language prompt?
> > > > > > > > > > >
> > > > > > > > > > > **Response:**
> > > > > > > > > > >
> > > > > > > > > > > The revisions addressing this issue are in the updated PDF, specifically in:
> > > > > > > > > > >
> > > > > > > > > > > **Line 892-906**
> > > > > > > > > > >
> > > > > > > > > > > The discussion related to the requirements described by users in natural language was given.
> > > > > > > > > > >
> > > > > > > > > > > **Line 352-403**
> > > > > > > > > > >
> > > > > > > > > > > Section: 4.2 RESEARCH QUESTION 2: UNDERSTANDING OF CNL-P BY LLMS
> > > > > > > > > > >
> > > > > > > > > > > We have redefined the focus of RQ2 and added the new description and experimental results about RQ2.
> > > > > > > > > > >
> > > > > > > > > > > **Line 474-492**
> > > > > > > > > > >
> > > > > > > > > > > The advantages of CNL-P are demonstrated, as well as the discussion of related vision that is:
> > > > > > > > > > >
> > > > > > > > > > > "We envision a broader suite of LLM-assisted software engineering tools to support the entire CNL-P development lifecycle."

---

> > > > > > > > > > > > ### Author Response · Authors · 2024-12-03
> > > > > > > > > > > >
> > > > > > > > > > > > **Comments:**
> > > > > > > > > > > >
> > > > > > > > > > > > Generalization of CNL-P:
> > > > > > > > > > > >
> > > > > > > > > > > > - As all the tasks in the experiment are classification tasks, the natural language prompts should not be overly complex (correct me if I am wrong). Consequently, the paper does not sufficiently address how CNL-P performs with very large and complex prompts. It also fails to clarify whether the linting tool can handle such complex CNL-P.
> > > > > > > > > > > >
> > > > > > > > > > > > **Response:**
> > > > > > > > > > > >
> > > > > > > > > > > > The revisions addressing this issue are in the updated PDF, specifically in:
> > > > > > > > > > > >
> > > > > > > > > > > > **Line 1242-1293**
> > > > > > > > > > > >
> > > > > > > > > > > > A specific instance examined by the linting tool in RQ3 is presented in Figure 19 and Figure 20.
> > > > > > > > > > > >
> > > > > > > > > > > > **Comments:**
> > > > > > > > > > > >
> > > > > > > > > > > > The plots and figures for result analysis should be integrated into the relevant pages or paragraphs; otherwise, it is hard to follow the discussion and analysis.
> > > > > > > > > > > >
> > > > > > > > > > > > **Response:**
> > > > > > > > > > > >
> > > > > > > > > > > > Thank you for your comment, the plots and figures for result analysis have been integrated in the paper, as shown in Figures 3 and 4.
> > > > > > > > > > > >
> > > > > > > > > > > > **Comments:**
> > > > > > > > > > > >
> > > > > > > > > > > > Which Llama model was used in your experiment?
> > > > > > > > > > > >
> > > > > > > > > > > > **Response:**
> > > > > > > > > > > >
> > > > > > > > > > > > Llama3-70B-8192 is used in our experiments, explanation added in **Line 388-389**
> > > > > > > > > > > >
> > > > > > > > > > > > ---
> > > > > > > > > > > >
> > > > > > > > > > > > We sincerely hope these revisions thoroughly address your comments.
> > > > > > > > > > > >
> > > > > > > > > > > > Thank you again for your time.
> > > > > > > > > > > >
> > > > > > > > > > > > Best wishes,
> > > > > > > > > > > >
> > > > > > > > > > > > All authors

---

### Official Review · Reviewer_m8X4 · 2024-11-04

**Soundness:** 3
**Presentation:** 2
**Contribution:** 4
**Rating:** 8
**Confidence:** 4

**Summary:**

The paper introduces Controlled Natural Language for Prompt (CNL-P), a novel framework that bridges prompt engineering (PE) and software engineering (SE) principles to enhance the clarity, predictability, and effectiveness of prompts for large language models (LLMs). CNL-P addresses inherent ambiguities in natural language prompts by formalizing grammar structures and semantic norms. The work's primary theoretical contribution is the formalization of prompt engineering through SE principles, supported by a novel static analysis approach for natural language prompts and empirical validation across multiple LLM architectures.

**Strengths:**

Theoretical Innovation:
- Novel synthesis of SE principles with PE practices
- Comprehensive formal grammar for controlled natural language
- Innovative application of static analysis theory to natural language

Technical Contribution
- Formal specification of the CNL-P grammar
- Theoretical framework for prompt verification
- Rigorous performance analysis across LLM architectures
- Novel approach to static analysis of natural language

Research Impact:
- Opens new theoretical directions in prompt engineering
- Bridges formal methods and LLM interaction
- Provides a foundation for analyzing prompt properties
- Advances understanding of structured approaches to PE

**Weaknesses:**

Theoretical Limitations:
- Formal analysis of expressive power could be stronger.
- Completeness properties of the static analysis need more discussion.
- Edge cases in the formal grammar require deeper analysis.
- Theoretical bounds need more rigorous treatment.

Methodological Concerns:
- Formal comparison with other structured approaches could be deeper.
- Statistical analysis could be more comprehensive.
- Theoretical justification for design choices needs elaboration.
- Formal properties of the conversion process require more analysis.

Validation Gaps:
Limited formal analysis of grammar properties.
Statistical significance analysis could be more rigorous.
Theoretical comparison with other formal methods needed.
Completeness of the static analysis approach not fully addressed.

**Questions:**

Comparative Evaluation: Could you provide a more detailed comparison of CNL-P’s functionality and usability versus DSPy, LangChain, and Semantic Kernel? Specifically, how does CNL-P’s approach to modularity and state management differ in terms of user accessibility and technical demands?

Advantages and Trade-offs: While CNL-P is designed to decouple prompts from code for accessibility, frameworks like DSPy offer robust control through tight integration with programming language abstractions. Could you discuss specific scenarios where CNL-P might outperform DSPy or vice versa, especially in terms of prompt complexity and user involvement?

Non-Technical Accessibility: CNL-P is described as more accessible to non-technical users than PL-based methods like DSPy and LangChain. Could you elaborate on any studies, tests, or qualitative comparisons you conducted to evaluate this claim? This would clarify the extent to which CNL-P lowers the barrier for non-programmers.

Performance in Practical Applications: Do you have insights or preliminary results comparing the performance and user experience of CNL-P with DSPy, LangChain, and Semantic Kernel in specific application areas (e.g., dynamic prompt generation or complex workflow management)? Real-world examples could strengthen the practical context of CNL-P’s advantages.

Future Integration with PL-Based Methods: Given that DSPy and other PL-based frameworks emphasize structured programming benefits, do you foresee potential for CNL-P to integrate with or complement these frameworks? A discussion on interoperability could highlight pathways for combining strengths across approaches.

Scoring and Evaluation: Can you elaborate on the specific criteria used to assign scores across the five evaluation dimensions (Adherence to Original Intent, Modularity, Extensibility and Maintainability, Readability and Structural Clarity, and Process Rigor)? Was there a weighting system applied to these dimensions, or were they treated as equally important?

Interpretability of Results: The table of results is challenging to interpret due to its minimal contextual information. Could you provide a more detailed breakdown or rubric that explains how scores were derived, potentially with examples of how different prompt types scored across specific dimensions?

Comparative Analysis: Did you consider using statistical measures to compare the performance of CNL-P, RISEN, and RODES across evaluation metrics? This could strengthen the validity of the reported improvements.

Presentation Improvements: The experimental results could benefit from a more visual presentation format, such as radar charts or bar graphs, for easier comparison across dimensions. Would you consider updating the results presentation in a revised version?

Completeness of Scoring Process: Did you perform any error analysis or additional validation to understand how CNL-P performs in specific scenarios where RISEN or RODES may excel, or vice versa? This would provide insight into potential edge cases for CNL-P.

---

> ### Author Response · Authors · 2024-11-20
>
> Dear Reviewer,
>
> We sincerely appreciate your insightful comments and constructive feedback, which have been invaluable in helping us clarify and refine key aspects of our paper.
>
> Below, we present detailed responses to your comments, prepared with careful consideration and thorough discussion over the past few days. Due to the content length limitations of a single response, our replies are spread across multiple rounds. We apologize for any inconvenience this may cause.
>
> ### Comments and Responses
>
> ***Comment***
> *[1] Comparative Evaluation: Could you provide a more detailed comparison of CNL-P’s functionality and usability versus DSPy, LangChain, and Semantic Kernel? Specifically, how does CNL-P’s approach to modularity and state management differ in terms of user accessibility and technical demands?*
> *[2] Advantages and Trade-offs: While CNL-P is designed to decouple prompts from code for accessibility, frameworks like DSPy offer robust control through tight integration with programming language abstractions. Could you discuss specific scenarios where CNL-P might outperform DSPy or vice versa, especially in terms of prompt complexity and user involvement?*
> *[3] Non-Technical Accessibility: CNL-P is described as more accessible to non-technical users than PL-based methods like DSPy and LangChain. Could you elaborate on any studies, tests, or qualitative comparisons you conducted to evaluate this claim? This would clarify the extent to which CNL-P lowers the barrier for non-programmers.*
> *[5] Future Integration with PL-Based Methods: Given that DSPy and other PL-based frameworks emphasize structured programming benefits, do you foresee potential for CNL-P to integrate with or complement these frameworks? A discussion on interoperability could highlight pathways for combining strengths across approaches.*
>
> ***Response***
> Since these four questions are closely interconnected, we provide a unified response to address them comprehensively.
>
> **Fundamental Differences: Implementation-Oriented versus. Requirement-Oriented**
> DSPy, Semantic Kernel, and similar PL-based frameworks are implementation-oriented technologies that extend or augment traditional programming languages with natural language addons. These frameworks rely on programming languages as their foundation, treating natural language as an augmented feature. Their core abstractions, such as:
>
> - LangChain’s prompt template, pipeline, output parsers,
> - DSPy’s signature and module concepts,
> - Semantic Kernel’s semantic functions, native functions, and plans,
>   are still fundamentally programming constructs, designed to encapsulate and simplify programmatic interaction with LLMs and related techniques (e.g., vector stores).
>
> However, these frameworks inherently depend on programming languages and cater to developers with technical expertise (at least to some extent). The technical demands remain high because these tools lack clear separation of concerns between NL and PL, intertwining natural language expressions with programmatic implementation. Consequently, non-technical users or domain experts without programming skills often find these tools inaccessible. For instance:
>
> - Users still need to understand prompts embedded in or glued by code statements and manage pipelines, parsers, or output templates.
> - These frameworks do not provide clear natural language syntaxes or semantics directly to natural language prompts, limiting their usability as true natural language tools.
>
> In contrast, CNL-P is a natural language-native design, built from the ground up as an executable requirement language for human-AI interaction. Unlike DSPy or Semantic Kernel, CNL-P:
>
> - Requires no reliance on programming languages (e.g., “All-in CNL-P”).
> - Allows users to express constraints,  data, workflows, state management, and security and guardrails (not present in this paper) directly in a structured controlled natural language (CNL).
> - Functions as a structured, semantically clear requirement document, akin to structured technical documentation rather than program code.
>
> By separating what to do (requirements definition) from how to do (code implementation), CNL-P enables non-technical users to define agent requirements clearly, leaving implementation details to supporting SE tools (e.g., The linting tool and the CNL-P compiler discussed in the paper).
>
> (Due to the length limit of a single reply, the remaining content of the response regarding the current comments is provided in the next reply.)

---

> > ### Author Response · Authors · 2024-11-20
> >
> > **User Accessibility and Technical Demand**
> > The key limitation of DSPy and Semantic Kernel lies in their high technical demand:
> >
> > - They are designed for developers, requiring significant programming expertise.
> > - They lack accessible tools or abstractions for non-technical users, restricting their usability in broader contexts.
> >
> > CNL-P, on the other hand, is specifically designed for non-technical users by nature. Its structure and design are inspired by many widely adopted technical documentation practices, such as:
> >
> > - Use Case Descriptions for requirements analysis
> > - User Stories (e.g., Gherkin User Story) for Acceptance Testing and Behavior Driven Development
> > - Standardized Technical Documents (e.g., ATA SPEC 100 in aviation).
> >
> > These technical documentation practices are tailored for non-technical stakeholders such as business analysts, domain experts, or other stakeholders who may lack technical expertise but need to communicate requirements effectively. CNL-P’s natural language-native design ensures:
> >
> > - Ease of use: Syntax and semantics are lightweight and intuitive, reducing the need for training or programming experience.
> > - Low technical barrier: CNL-P can be used by non-technical stakeholders who can write structured requirements, making it as accessible as technical documentation.
> >
> > In addition, the user accessibility and technical demand of CNL-P can be seen more clearly through the designed RQs of the paper.
> >
> > RQ1: Building on the inherent simplicity and ease of understanding of CNL-P's grammar, RQ1 introduces an NL-to-CNL-P conversion agent. This agent further reduces the user's workload by automatically transforming natural language descriptions into CNL-P syntax.
> >
> > RQ2: The simple and intuitive design of CNL-P allows Large Language Models (LLMs) to understand its syntax and semantics without the need for providing CNL-P language specifications, few-shot examples, or additional model training and fine-tuning. This allows LLMs to generate outputs on par with those produced by natural language prompts.
> >
> > RQ3: The linting tool exemplifies CNL-P's broader utility. Functioning as the front-end of a compiler, the linting tool performs syntax and semantic analysis, helping users identify and resolve issues without requiring the user-written CNL-P strictly adhering to CNL-P syntax. This significantly lowers the technical barrier to entry, ensuring accessibility while maintaining precision and reliability.
> >
> > These research questions collectively demonstrate CNL-P's potential to enhance user accessibility and reduce technical demands, thereby advancing the field of software engineering for AI.
> >
> > It is worth noting that these RQs also demonstrate that CNL-P together with the developed NL-to-CNL-P agent, and the linting tool opens the door to achieving the vision of advancing a more natural and robust SE paradigm for AI infrastructure to enhance human-AI Interaction.
> >
> > Moreover, our early adoption in our lab and with non-technical users (such as high-school teachers, chemists) provide initial evidence on the user accessibility of CNL-P. For example, even the fresh first-year undergraduates with little technical expertise in our lab can effectively use CNL-P to define LLM-based agents. In fact, the first author of this paper is a second-year undergraduate with no significant AI and programming training.
> >
> > (Due to the length limit of a single reply, the remaining content of the response regarding the current comments is provided in the next reply.)

---

> > > ### Author Response · Authors · 2024-11-20
> > >
> > > **CNL-P as a Complement to Existing PL-based Frameworks**
> > > CNL-P is not a competitor to frameworks like DSPy or Semantic Kernel but a complementary tool addressing a different challenge in agent development. Specifically:
> > >
> > > - CNL-P focuses on what to do: Helping users clearly define, structure, and document requirements at a high level of abstraction.
> > > - DSPy and Semantic Kernel focus on how to do: Translating high-level requirements into programmatic implementations and facilitating interactions with LLMs.
> > >
> > > The relationship can be likened to an API specification versus API implementations: CNL-P defines a structured, human-readable guideline for describing agent requirements, while PL-based frameworks like DSPy operationalize it through specific programming abstractions. Furthermore, as discussed in Section 5, we are designing a CNL-P compiler that translates user-defined requirements in CNL-P into executable code compatible with frameworks such as DSPy, LangChain, or Semantic Kernel, or other emerging PL-based frameworks.
> > >
> > > This separation of requirements in CNL-P -> CNL-P Compiler -> implementations in PLs enables non-technical users to:
> > >
> > > - Focus on high-level requirements without worrying about implementation details.
> > > - Rely on a compiler to generate corresponding code in the desired programming framework, analogous to how high-level programming languages (e.g., Python, Java) are compiled into machine instructions for execution.
> > > - Enjoy the benefits of decoupled requirements and implementations, analogous to “write once, run anywhere” for high-level PLs like Java.
> > >
> > > This CNL->compiler->PL architecture inherits the core principles of existing high-level programming paradigms (PL-> compiler->Machine Instructions), leveraging the capabilities of LLMs to push programming languages closer to natural language. At the same time, it retains the benefits of decoupling high-level programming languages from machine code.
> > >
> > > The CNL-P linting tool presented in Section 3.3 is a Proof-of-Concept of the front-end of this envisioned CNL-P compiler. Our experiments in RQ3 (Section 4.3) demonstrate the feasibility of building the front-end of such a CNL-P compiler and its effectiveness in applying robust static analysis to CNL. Connecting this front-end with a compiler backend (e.g., an LLM-based code generator), we will be able to generate agent implementation from the intermediate representation of CNL-P like the one presented in Figure 2.
> > >
> > > **Modularity and State Management**
> > > A significant advantage of CNL-P is its introduction of modularity and state management at the requirements level, as opposed to the programmatic level. For example:
> > >
> > > - Modularity in CNL-P: Users can define data, constraints and instructions in a simple structured and modular syntax, ensuring clarity and reusability.
> > > - State Management: CNL-P allows users to define state-related requirements (e.g., transitions and context dependencies) without requiring them to manage runtime state explicitly. Instead, the CNL-P compiler and runtime will handle these details automatically.
> > >
> > > This approach aligns with high-level programming paradigms: Just as high-level programming languages like Python and Java abstract memory management, CNL-P abstracts away implementation complexities of LLM interactions, enabling users to focus on defining what states and transitions are required, while how to implement and manage state and context will be handled by CNL-P compiler and runtime.
> > >
> > > (Due to the length limit of a single reply, the remaining content of the response regarding the current comments is provided in the next reply.)

---

> > > > ### Author Response · Authors · 2024-11-20
> > > >
> > > > **Extensibility and Ecosystem**
> > > > We recognize that a CNL like CNL-P alone cannot address all the challenges of LLM-based agent development. Just as programming languages rely on compilers and an ecosystem of supporting tools—such as linters and testing frameworks—CNL-P also requires a robust ecosystem of software engineering tools to support the development, testing, maintenance and operation of CNL-P agents. This work, though in its early stage, illustrates that CNL-P paves the way for the development of such a comprehensive SE4AI infrastructure.
> > > >
> > > > We recognize that a controlled natural language like CNL-P alone cannot address all the challenges of LLM-based agent development. Just as programming languages rely on compilers and  an ecosystem of supporting tools—such as linters and testing frameworks—CNL-P also requires a robust ecosystem of software engineering tools to support the development, testing, maintenance and operation of CNL-P agents. This work, though in its early stage, illustrates that CNL-P paves the way for the development of such a comprehensive SE4AI infrastructure.
> > > >
> > > > CNL-P is designed to support an evolving ecosystem of SE tools that lower the barriers for adoption and improve user accessibility. In this paper, we presented the proof of concept of two such tools:
> > > >
> > > > - a Transformer agent which formats a free-form NL prompt into the CNL-P syntax (Section 3.2);
> > > > - a CNL-P linting tool which validates and optimizes natural language requirements for clarity and correctness through systematic static analysis (Section 3.3).
> > > >
> > > > Furthermore, we discussed the LLM-empowered CNL-P Compiler which can transform high-level requirements in CNL-P into executable code for PL-based frameworks like DSPy or LangChain.
> > > >
> > > > We envision more LLM-assisted software engineering tools to support the entire lifecycle of CNL-P development, for example,
> > > >
> > > > - A CNL-P tutorial agent that can teach CNL-P, provide relevant examples, and answer CNL-P syntax and semantic questions.
> > > > - A requirements agent that can iteratively developing detailed CNL-P requirements from initial vague and incomplete natural language descriptions, similar to the agile requirement analysis process.
> > > > - A CNL testing framework that can detecting incompleteness, inconsistencies or ambiguities in requirements, similar to Gherkin User Stories and Behavior Driven Development (BDD).
> > > > - A documentation agent that can transform CNL-P requirements into technical documentation (e.g., reStructuredText) to integrate CNL-P into the broader software DevOps context.
> > > >
> > > > Such a SE tool ecosystem around CNL-P aligns with software engineering principles and best practices such as early error detection, which reduces the cost of fixing issues during later stages of development. By identifying errors or inconsistencies at the requirements level, CNL-P prevents costly implementation mistakes. All such tools will constitute a CNL-P centric SE4AI infrastructure, standing on the shoulders of LLMs for more people to leverage AI in their life lives and work.
> > > >
> > > > **Quality Assurance and Testing**
> > > > CNL-P can support improved quality assurance by integrating concepts from behavior-driven development (BDD). For example, as discussed in Section 5,:
> > > >
> > > > - Users can define use cases, alternative flows, and exception handling directly within CNL-P.
> > > > - These structured definitions can be validated against predefined testing scenarios, ensuring correctness before implementation begins.
> > > >   By combining BDD principles with LLM capabilities, CNL-P enables robust testing of agents and workflows, even at the requirements level.
> > > >
> > > > *Conclusion*
> > > > CNL-P represents a paradigm shift from program-centric frameworks like DSPy and Semantic Kernel to a natural language-first design. It is tailored for non-technical users, offering a structured, semantically clear medium for defining requirements, which can then be transpiled into executable code using a compiler. This separation of concerns between what to do and how to do reduces technical barriers and ensures broader user accessibility.
> > > >
> > > > CNL-P complements existing frameworks by focusing on requirement clarity while leveraging the capabilities of PL-based frameworks for implementation. Its extensibility and ecosystem, inspired by software engineering principles and best practices, make it a promising tool for advancing a more natural and robust SE4AI infrastructure to enhance human-AI Interaction.

---

> > > > > ### Author Response · Authors · 2024-11-20
> > > > >
> > > > > ***Comment***:
> > > > > [4] Performance in Practical Applications: Do you have insights or preliminary results comparing the performance and user experience of CNL-P with DSPy, LangChain, and Semantic Kernel in specific application areas (e.g., dynamic prompt generation or complex workflow management)? Real-world examples could strengthen the practical context of CNL-P’s advantages.
> > > > >
> > > > > ***Response***:
> > > > > CNL-P has been applied in several real-world scenarios, demonstrating its effectiveness in addressing diverse requirements across dynamic prompt generation and complex workflow management. Due to anonymity constraints, we cannot disclose specific project details, but the following insights summarize its practical performance:
> > > > >
> > > > > - In collaboration with an education faculty, CNL-P was used to design AI agents for adaptive learning environments. Educators, who lacked programming expertise, successfully employed CNL-P to define constraints, learning paths, and content adaptations in natural language. Feedback highlighted the clarity of CNL-P’s syntax and its ability to abstract technical complexities, making it accessible for non-technical users.
> > > > > - CNL-P was adopted in a government project for defining workflows and ensuring regulatory compliance in sensitive data-sharing scenarios. The modularity of CNL-P allowed stakeholders to articulate state transitions, security policies, and exception handling without requiring developer intervention. Initial testing revealed significant reductions in time spent translating requirements into implementable workflows compared to traditional frameworks like DSPy or LangChain.
> > > > >
> > > > > **User Feedback and Performance Insights**:
> > > > > Across these projects, users reported:
> > > > >
> > > > > - Accessibility: Non-technical stakeholders could articulate requirements directly in CNL-P without relying on technical intermediaries.
> > > > > - Efficiency: The upfront clarity of CNL-P reduced iteration cycles, enabling faster convergence on functional agentic workflows.
> > > > >
> > > > > These results underpin CNL-P’s strengths in bridging technical and non-technical collaboration, providing a structured yet intuitive framework for defining executable requirements. Future work will expand these use cases and conduct formal evaluations to quantify its impact compared to existing frameworks.**We will provide an overview of these real-world applications in the revision.**

---

> > > > > > ### Author Response · Authors · 2024-11-20
> > > > > >
> > > > > > ***Comment***:
> > > > > > *[6] Scoring and Evaluation: Can you elaborate on the specific criteria used to assign scores across the five evaluation dimensions (Adherence to Original Intent, Modularity, Extensibility and Maintainability, Readability and Structural Clarity, and Process Rigor)? Was there a weighting system applied to these dimensions, or were they treated as equally important?*
> > > > > >
> > > > > > ***Response***:
> > > > > > The scoring indicators of the five dimensions in RQ1 were not included in the paper due to the length limitation of the paper. During the actual experiment, the specific scoring criteria are as follows. We will add them to the appendix.
> > > > > > """
> > > > > > Scoring Criteria:
> > > > > >
> > > > > > 1. Adherence to Original Intent
> > > > > >    Scoring Range:
> > > > > >    1-33: Significantly deviates from the original intent, with numerous omissions, misunderstandings, or misuse of structured keywords that distort the original semantics.
> > > > > >    34-66: Mostly aligns with the original intent but contains some inaccuracies or unnecessary modifications due to structured keywords, causing slight deviations from the intended meaning, or an overemphasis on structure over semantics.
> > > > > >    67-100: Completely faithful to the original semantic content, with structured keywords effectively clarifying module boundaries and enhancing understanding without altering or losing any of the original meaning. Examples from the original are adapted appropriately, maintaining the intended message.
> > > > > > 2. Readability and Structural Clarity
> > > > > >    Scoring Range:
> > > > > >    1-33: Structure is unclear, with structured keywords either absent or misused, leading to ambiguous module boundaries, redundancy, or a lack of clarity; the LLM would struggle to parse and understand the Prompt.
> > > > > >    34-66: Structure is somewhat clear, and structured keywords are present but inconsistently or inadequately applied, resulting in some ambiguity or minor redundancy; requires more processing effort from the LLM to understand the modular structure.
> > > > > >    67-100: Structure is very clear, with structured keywords skillfully applied to define module boundaries, enhance readability, and provide unambiguous guidance, ensuring the LLM can easily interpret and understand the Prompt.
> > > > > > 3. Modularity
> > > > > >    Scoring Range:
> > > > > >    1-33: Low level of modularity; the Prompt fails to effectively organize structured knowledge, resulting in poorly defined or highly coupled modules.
> > > > > >    34-66: Demonstrates moderate modularity, with reasonable identification of structured knowledge, but some modules show dependencies or coupling.
> > > > > >    67-100: Exhibits a high level of modularity, accurately identifying and organizing structured knowledge into well-defined, independent modules with minimal redundancy and coupling.
> > > > > > 4. Extensibility and Maintainability
> > > > > >    Scoring Range:
> > > > > >    1-33: Lacks extensibility and maintainability; unclear rationale for changes, making modification difficult and destabilizing the overall structure.
> > > > > >    34-66: Offers some degree of extensibility and maintainability, but modification points are not always clear, requiring additional judgment for adjustments.
> > > > > >    67-100: Highly extensible and maintainable, with well-defined guidance for changes, allowing accurate and seamless modifications without disrupting the overall structure.
> > > > > > 5. Process Rigor
> > > > > >    Scoring Range:
> > > > > >    1-33: Lacks clear process rigor, with unclear instruction iterations, variable management, or input-output handling, leading to confusion in LLM execution.
> > > > > >    34-66: Demonstrates a moderately clear process, but certain iterations, variable passing, or input-output elements are not fully defined, requiring the LLM to infer missing steps.
> > > > > >    67-100: Process is rigorously defined, with clear iterations, variable passing, and input-output flow, allowing the LLM to execute instructions with precision.
> > > > > >    """
> > > > > >
> > > > > > We did not apply weights to these dimensions, as we believe it is challenging to quantify the relative importance of each dimension. Instead, each dimension was assessed independently, with scores assigned separately for each dimension.

---

> > > > > > > ### Author Response · Authors · 2024-11-20
> > > > > > >
> > > > > > > ***Comment***
> > > > > > > *[7] Interpretability of Results: The table of results is challenging to interpret due to its minimal contextual information. Could you provide a more detailed breakdown or rubric that explains how scores were derived, potentially with examples of how different prompt types scored across specific dimensions?*
> > > > > > >
> > > > > > > ***Response***
> > > > > > > Please allow us to clarify how the results presented in the RQ1 table were derived:
> > > > > > >
> > > > > > > The 93 examples in RQ1 were sourced from a GitHub repository titled "awesome-chatgpt-prompts," a highly popular project with 113k stars. We initially selected 93 longer prompts from this collection, aiming to ensure they contained relatively comprehensive requirements and rich content, thereby enabling the generation of high-quality representations in CNL-P, RISEN, and RODES.
> > > > > > >
> > > > > > > Based on the scoring criteria mentioned above, we tasked the model to evaluate the prompts as if it were an expert in the field of Prompt Engineering. Specific examples were given in the response to the following question [10] “Completeness of Scoring Process”.
> > > > > > >
> > > > > > > ---
> > > > > > >
> > > > > > > ***Comment***
> > > > > > > *[8] Comparative Analysis: Did you consider using statistical measures to compare the performance of CNL-P, RISEN, and RODES across evaluation metrics? This could strengthen the validity of the reported improvements.*
> > > > > > > *[9] Presentation Improvements: The experimental results could benefit from a more visual presentation format, such as radar charts or bar graphs, for easier comparison across dimensions. Would you consider updating the results presentation in a revised version?*
> > > > > > > *[10] Completeness of Scoring Process: Did you perform any error analysis or additional validation to understand how CNL-P performs in specific scenarios where RISEN or RODES may excel, or vice versa? This would provide insight into potential edge cases for CNL-P.*
> > > > > > >
> > > > > > > ***Response***
> > > > > > > We sincerely appreciate your suggestions to "consider statistical methods for comparing the performance of CNL-P, RISEN, and RODES across evaluation metrics" and to "present experimental results in more visually impactful formats (e.g., radar charts or bar charts) to facilitate comparisons across dimensions."
> > > > > > >
> > > > > > > Following your comment, we have created box plots to better analyze the performance of CNL-P, RISEN, and RODES on the evaluation metrics. Additionally, radar charts have been utilized to visually present the results, making the comparisons across dimensions clearer and more intuitive.  Since images cannot be attached in the reply box, we will update the paper during the remaining time of the discussion period. The figures will be placed in the updated paper.
> > > > > > >
> > > > > > > By analyzing the lower outliers for CNL-P in the box plots, we identified a representative case, "DIY Expert" (specific NL and CNL-P prompts are provided below). In this instance, the scores for Adherence to Original Intent, Extensibility and Maintainability, Readability and Structural Clarity, and Process Rigor were all lower outliers, significantly lower than those of RISEN and RODES.
> > > > > > >
> > > > > > > Upon examining the NL prompt and the converted CNL-P, we observed that the original NL text was primarily focused on describing the Agent Persona, with little to no mention of constraints or processes. This lack of detailed descriptions in the NL prompt led to content gaps in the corresponding CNL-P modules, resulting in significantly reduced scores.
> > > > > > >
> > > > > > > We also analyzed the case of "SVG Designer" (specific NL and CNL-P prompts are provided below). Compared to RISEN and RODES, CNL-P demonstrated varying degrees of superior performance across all dimensions, with its score for Process Rigor being the only upper outlier among the five box plots for CNL-P.
> > > > > > >
> > > > > > > Upon reviewing the CNL-P, we found that its modules were exceptionally well-structured and comprehensive. This high-quality outcome can be attributed to the well-crafted and detailed original NL prompt, highlighting the importance of input quality in achieving superior CNL-P outputs.
> > > > > > >
> > > > > > > (Due to the length limit of a single reply, the remaining content of the response regarding the current comments is provided in the next reply.)

---

> > > > > > > > ### Author Response · Authors · 2024-11-20
> > > > > > > >
> > > > > > > > **DIY EXPERT**
> > > > > > > > *NL:*
> > > > > > > > I want you to act as a DIY expert. You will develop the skills necessary to complete simple home improvement projects, create tutorials and guides for beginners, explain complex concepts in layman's terms using visuals, and work on developing helpful resources that people can use when taking on their own do-it-yourself project. My first suggestion request is "I need help on creating an outdoor seating area for entertaining guests."
> > > > > > > >
> > > > > > > > *CNL-P:*
> > > > > > > >
> > > > > > > > ```plaintext
> > > > > > > > [DEFINE_AGENT: DIYExpert "A DIY expert who assists with home improvement projects"]
> > > > > > > >     [DEFINE_PERSONA:]
> > > > > > > >         ROLE: Act as a DIY expert
> > > > > > > >         Skills: Develop the skills necessary to complete simple home improvement projects
> > > > > > > >         Tutorials: Create tutorials and guides for beginners
> > > > > > > >         Explanation: Explain complex concepts in layman's terms using visuals
> > > > > > > >         Resources: Work on developing helpful resources for DIY projects
> > > > > > > >     [END_PERSONA]
> > > > > > > >
> > > > > > > >     [DEFINE_CONSTRAINTS:]
> > > > > > > >     [END_CONSTRAINTS]
> > > > > > > >
> > > > > > > >     [DEFINE_VARIABLES:]
> > > > > > > >     [END_VARIABLES]
> > > > > > > >
> > > > > > > >     [DEFINE_WORKER: "Assist with DIY projects" DIYWorker]
> > > > > > > >         [INPUTS]
> > > > > > > >         [END_INPUTS]
> > > > > > > >
> > > > > > > >         [OUTPUTS]
> > > > > > > >         [END_OUTPUTS]
> > > > > > > >
> > > > > > > >         [MAIN_FLOW]
> > > > > > > >             [SEQUENTIAL_BLOCK]
> > > > > > > >                 COMMAND-1 [DISPLAY I need help on creating an outdoor seating area for entertaining guests]
> > > > > > > >             [END_SEQUENTIAL_BLOCK]
> > > > > > > >         [END_MAIN_FLOW]
> > > > > > > >     [END_WORKER]
> > > > > > > > [END_AGENT]
> > > > > > > > ```
> > > > > > > >
> > > > > > > > **SVG DESIGNER**
> > > > > > > > *NL*
> > > > > > > > I would like you to act as an SVG designer. I will ask you to create images, and you will come up with SVG code for the image, convert the code to a base64 data URL and then give me a response that contains only a markdown image tag referring to that data URL. Do not put the markdown inside a code block. Send only the markdown, so no text. My first request is: give me an image of a red circle.
> > > > > > > >
> > > > > > > > *CNL-P*
> > > > > > > >
> > > > > > > > ```plaintext
> > > > > > > > [DEFINE_AGENT: SVGDesignerAgent "An agent that generates SVG images and provides them as markdown image tags."]
> > > > > > > >     [DEFINE_PERSONA:]
> > > > > > > >         ROLE: Act as an SVG designer
> > > > > > > >     [END_PERSONA]
> > > > > > > >
> > > > > > > >     [DEFINE_CONSTRAINTS:]
> > > > > > > >         ResponseFormat: Send only the markdown image tag without any additional text or code block
> > > > > > > >     [END_CONSTRAINTS]
> > > > > > > >
> > > > > > > >     [DEFINE_VARIABLES:]
> > > > > > > >         imageRequest: text
> > > > > > > >         svgCode: text
> > > > > > > >         base64DataUrl: text
> > > > > > > >         markdownImageTag: text
> > > > > > > >     [END_VARIABLES]
> > > > > > > >
> > > > > > > >     [DEFINE_WORKER: "Handles SVG image requests and responses" SVGWorker]
> > > > > > > >         [INPUTS]
> > > > > > > >             @imageRequest
> > > > > > > >         [END_INPUTS]
> > > > > > > >
> > > > > > > >         [OUTPUTS]
> > > > > > > >             @markdownImageTag
> > > > > > > >         [END_OUTPUTS]
> > > > > > > >
> > > > > > > >         [MAIN_FLOW]
> > > > > > > >             [SEQUENTIAL_BLOCK]
> > > > > > > >                 COMMAND-1 [COMMAND Generate SVG code for the requested image @imageRequest RESULT svgCode: text SET]
> > > > > > > >                 COMMAND-2 [COMMAND Convert @svgCode to base64 data URL RESULT base64DataUrl: text SET]
> > > > > > > >                 COMMAND-3 [COMMAND Create markdown image tag with @base64DataUrl RESULT markdownImageTag: text SET]
> > > > > > > >                 COMMAND-4 [DISPLAY @markdownImageTag]
> > > > > > > >             [END_SEQUENTIAL_BLOCK]
> > > > > > > >         [END_MAIN_FLOW]
> > > > > > > >     [END_WORKER]
> > > > > > > > [END_AGENT]
> > > > > > > > ```
> > > > > > > >
> > > > > > > > ---
> > > > > > > >
> > > > > > > > We sincerely hope these responses thoroughly address your comments. If anything remains unclear, we would greatly appreciate the opportunity to discuss it further with you.
> > > > > > > >
> > > > > > > > Thank you again for your time.
> > > > > > > >
> > > > > > > > Best wishes,
> > > > > > > > All authors

---

> > > > > > > > > ### Comment · Reviewer_m8X4 · 2024-11-27
> > > > > > > > > **Response to authors**
> > > > > > > > >
> > > > > > > > > I thank the authors for their comprehensive responses. While reviewing the extensive clarifications provided, I've come to appreciate even more deeply the significance of this work. The importance of treating LLMs as APIs through controlled natural language interfaces will likely become increasingly critical as these models become more integrated into software systems.
> > > > > > > > > The significance of this work stems from several key aspects:
> > > > > > > > >
> > > > > > > > > Future-Looking Architecture: By treating prompts as APIs through CNL-P, this work anticipates and addresses what will likely become a fundamental challenge in software engineering - creating reliable, maintainable interfaces to LLM capabilities. The parallel to how APIs revolutionized software integration seems particularly apt.
> > > > > > > > > Bridging Technical Gaps: The authors' approach of separating "what to do" (requirements in CNL-P) from "how to do" (implementation in frameworks like DSPy) creates an essential abstraction layer that will become increasingly valuable as LLM-based systems grow more complex.
> > > > > > > > > Ecosystem Foundation: The development of supporting tools (linting, compilation) demonstrates how traditional software engineering practices can be adapted for the LLM era, potentially establishing patterns that will become standard practice.
> > > > > > > > >
> > > > > > > > > The responses have addressed my technical concerns about evaluation metrics, practical applications, and theoretical foundations. While some theoretical aspects could be strengthened, I maintain my rating of 8 (accept) because this paper represents an important step toward what I believe will become a crucial paradigm in software engineering.
> > > > > > > > > The real-world examples from education and government sectors, combined with the extensible architecture and tool ecosystem, suggest this approach has both immediate utility and long-term significance for the field. As LLMs become more deeply embedded in software systems, having structured, maintainable ways to interact with them - as this paper proposes - will only grow in importance.

---

> > > > > > > > > > ### Author Response · Authors · 2024-12-03
> > > > > > > > > >
> > > > > > > > > > Dear reviewer,
> > > > > > > > > >
> > > > > > > > > > Thank you for your in-depth understanding and support for our work. We will do our best to realize our vision.
> > > > > > > > > >
> > > > > > > > > > We have revised the paper based on your comments and uploaded the updated PDF file of the paper. Thank you for your constructive suggestions, which have greatly improved the quality of the paper.
> > > > > > > > > >
> > > > > > > > > > We would like to take this opportunity to inform you of the specific revised details made in the revised PDF file. These revisions are based on our previous responses. For the sake of brevity, we have not repeated the content of our prior response here. Should you require further details, please kindly refer to the previous response for your reference. Due to the page limit, our paper revisions are more concise than our responses, but they retain all the essential elements.
> > > > > > > > > >
> > > > > > > > > > **Comments：**
> > > > > > > > > >
> > > > > > > > > > *\[1\] Comparative Evaluation: Could you provide a more detailed comparison of CNL-P's functionality and usability versus DSPy, LangChain, and Semantic Kernel? Specifically, how does CNL-P's approach to modularity and state management differ in terms of user accessibility and technical demands?*
> > > > > > > > > >
> > > > > > > > > > *\[2\] Advantages and Trade-offs: While CNL-P is designed to decouple prompts from code for accessibility, frameworks like DSPy offer robust control through tight integration with programming language abstractions. Could you discuss specific scenarios where CNL-P might outperform DSPy or vice versa, especially in terms of prompt complexity and user involvement?*
> > > > > > > > > >
> > > > > > > > > > *\[3\] Non-Technical Accessibility: CNL-P is described as more accessible to non-technical users than PL-based methods like DSPy and LangChain. Could you elaborate on any studies, tests, or qualitative comparisons you conducted to evaluate this claim? This would clarify the extent to which CNL-P lowers the barrier for non-programmers.*
> > > > > > > > > >
> > > > > > > > > > *\[5\] Future Integration with PL-Based Methods: Given that DSPy and other PL-based frameworks emphasize structured programming benefits, do you foresee potential for CNL-P to integrate with or complement these frameworks? A discussion on interoperability could highlight pathways for combining strengths across approaches.*
> > > > > > > > > >
> > > > > > > > > > **Response:**
> > > > > > > > > >
> > > > > > > > > > The revisions addressing this issue are in the updated PDF, specifically in:
> > > > > > > > > >
> > > > > > > > > > **Line 1300-line1453**
> > > > > > > > > >
> > > > > > > > > > A.10 COMPREHENSIVE COMPARATIVE EVALUATION OF CNL-P VERSUS DSPY, LANGCHAIN, AND SEMANTIC KERNEL
> > > > > > > > > >
> > > > > > > > > > **Comments:**
> > > > > > > > > >
> > > > > > > > > > *\[4\] Performance in Practical Applications: Do you have insights or preliminary results comparing the performance and user experience of CNL-P with DSPy, LangChain, and Semantic Kernel in specific application areas (e.g., dynamic prompt generation or complex workflow management)? Real-world examples could strengthen the practical context of CNL-P's advantages.*
> > > > > > > > > >
> > > > > > > > > > **Response:**
> > > > > > > > > >
> > > > > > > > > > Due to anonymity restrictions, we have not included specific case examples at this stage. Once the anonymity restrictions are lifted, we will incorporate the relevant case examples.
> > > > > > > > > >
> > > > > > > > > > **Below recall the previous response about this comment, please refer:**
> > > > > > > > > >
> > > > > > > > > > CNL-P has been applied in several real-world scenarios, demonstrating its effectiveness in addressing diverse requirements across dynamic prompt generation and complex workflow management. Due to anonymity constraints, we cannot disclose specific project details, but the following insights summarize its practical performance:
> > > > > > > > > >
> > > > > > > > > > 1. In collaboration with an education faculty, CNL-P was used to design AI agents for adaptive learning environments. Educators, who lacked programming expertise, successfully employed CNL-P to define constraints, learning paths, and content adaptations in natural language. Feedback highlighted the clarity of CNL-P's syntax and its ability to abstract technical complexities, making it accessible for non-technical users.
> > > > > > > > > >
> > > > > > > > > > 2. CNL-P was adopted in a government project for defining workflows and ensuring regulatory compliance in sensitive data-sharing scenarios. The modularity of CNL-P allowed stakeholders to articulate state transitions, security policies, and exception handling without requiring developer intervention. Initial testing revealed significant reductions in time spent translating requirements into implementable workflows compared to traditional frameworks like DSPy or LangChain.
> > > > > > > > > >
> > > > > > > > > > **User Feedback and Performance Insights**:\
> > > > > > > > > > Across these projects, users reported:
> > > > > > > > > >
> > > > > > > > > > - **Accessibility**: Non-technical stakeholders could articulate requirements directly in CNL-P without relying on technical intermediaries.
> > > > > > > > > > - **Efficiency**: The upfront clarity of CNL-P reduced iteration cycles, enabling faster convergence on functional agenetic workflows.
> > > > > > > > > >
> > > > > > > > > > These results underpin CNL-P's strengths in bridging technical and non-technical collaboration, providing a structured yet intuitive framework for defining executable requirements. Future work will expand these use cases and conduct formal evaluations to quantify its impact compared to existing frameworks. **We will provide an overview of these real-world applications in the revision.**

---

> > > > > > > > > > > ### Author Response · Authors · 2024-12-03
> > > > > > > > > > >
> > > > > > > > > > > **Comments:**
> > > > > > > > > > >
> > > > > > > > > > > *\[6\] Scoring and Evaluation: Can you elaborate on the specific criteria used to assign scores across the five evaluation dimensions (Adherence to Original Intent, Modularity, Extensibility and Maintainability, Readability and Structural Clarity, and Process Rigor)? Was there a weighting system applied to these dimensions, or were they treated as equally important?*
> > > > > > > > > > >
> > > > > > > > > > > **Response:**
> > > > > > > > > > >
> > > > > > > > > > > The revisions addressing this issue are in the updated PDF, specifically in:
> > > > > > > > > > >
> > > > > > > > > > > **Line 925-line 960**
> > > > > > > > > > >
> > > > > > > > > > > Section: A.6.1 EVALUATION CRITERIA OF EXPERIMENT IN RQ1
> > > > > > > > > > >
> > > > > > > > > > > **Comments:**
> > > > > > > > > > >
> > > > > > > > > > > *\[7\] Interpretability of Results: The table of results is challenging to interpret due to its minimal contextual information. Could you provide a more detailed breakdown or rubric that explains how scores were derived, potentially with examples of how different prompt types scored across specific dimensions?*
> > > > > > > > > > >
> > > > > > > > > > > **Response:**
> > > > > > > > > > >
> > > > > > > > > > > The revisions addressing this issue are in the updated PDF, specifically in:
> > > > > > > > > > >
> > > > > > > > > > > **Line 925-960**
> > > > > > > > > > >
> > > > > > > > > > > Section: A.6.1 EVALUATION CRITERIA OF EXPERIMENT IN RQ1
> > > > > > > > > > >
> > > > > > > > > > > **Line 1015-1064**
> > > > > > > > > > >
> > > > > > > > > > > Section: A.6.3 ANALYSIS OF SPECIAL CASES IN THE EVALUATION PROCESS OF RQ1
> > > > > > > > > > >
> > > > > > > > > > > In addition, thank you for your suggestion. We have changed the table to the radar chart for display, as shown in Figure 3.
> > > > > > > > > > >
> > > > > > > > > > > **Comments:**
> > > > > > > > > > >
> > > > > > > > > > > *\[8\] Comparative Analysis: Did you consider using statistical measures to compare the performance of CNL-P, RISEN, and RODES across evaluation metrics? This could strengthen the validity of the reported improvements.*
> > > > > > > > > > >
> > > > > > > > > > > *\[9\] Presentation Improvements: The experimental results could benefit from a more visual presentation format, such as radar charts or bar graphs, for easier comparison across dimensions. Would you consider updating the results presentation in a revised version?*
> > > > > > > > > > >
> > > > > > > > > > > *\[10\] Completeness of Scoring Process: Did you perform any error analysis or additional validation to understand how CNL-P performs in specific scenarios where RISEN or RODES may excel, or vice versa? This would provide insight into potential edge cases for CNL-P.*
> > > > > > > > > > >
> > > > > > > > > > > **Response:**
> > > > > > > > > > >
> > > > > > > > > > > The statistical charts have been added and are shown in **Figures 10, 11, and 12.**
> > > > > > > > > > >
> > > > > > > > > > > Thank you for your suggestion. We presented the results using radar charts and bar graphs. As shown in **Figures 3, 4, 10, 11, 12, and 18.**
> > > > > > > > > > >
> > > > > > > > > > > Specific examples were presented and described:
> > > > > > > > > > >
> > > > > > > > > > > **Line 1015-1064**
> > > > > > > > > > >
> > > > > > > > > > > Section: A.6.3 ANALYSIS OF SPECIAL CASES IN THE EVALUATION PROCESS OF RQ1
> > > > > > > > > > >
> > > > > > > > > > > **and Figure 13,14,15.**
> > > > > > > > > > >
> > > > > > > > > > > ---
> > > > > > > > > > >
> > > > > > > > > > > We sincerely hope these revisions thoroughly address your comments.
> > > > > > > > > > >
> > > > > > > > > > > Thank you again for your time.
> > > > > > > > > > >
> > > > > > > > > > > Best wishes,
> > > > > > > > > > >
> > > > > > > > > > > All authors

---

### Official Review · Reviewer_32pw · 2024-11-05

**Soundness:** 3
**Presentation:** 2
**Contribution:** 2
**Rating:** 6
**Confidence:** 2

**Summary:**

- This paper proposes the use of Controlled Natural Language (CNL) by framing prompts as a form of API, which allows users to harness AI model capabilities without needing in-depth technical knowledge.
- This work applies software engineering (SE) principles such as modularity, abstraction, and encapsulation to Controlled Natural Language (CNL), offering a structure that decouples prompts from code
- The authors conducted experiments to evaluate how effectively CNL-P adheres to design principles and whether CNL-P or template methods improve the quality of LLM responses.

**Strengths:**

- Clearly motivated framing of prompting as an API, enabling users to leverage AI model capabilities without extensive technical expertise.
- Strong connection to first principles in SE, providing a foundation to address challenges in complex NL-PL conversion and prompt-code coupling. This approach is particularly beneficial for language experts and non-technical users by effectively decoupling prompts from code.
- Dimensions to assess NL-to-CNL-P conversion quality are well-designed, covering diverse quality aspects.

**Weaknesses:**

- The specific aims of the work remain unclear; while high-level challenges and design considerations are presented, the precise goals are hard to identify.
- Experiment setup in RQ1 lacks clarity on how the five dimensions are measured and how the 93 prompt instances were chosen. There are also no human validation results presented, even as partial samples.
- RQ1 primarily assesses design considerations, while RQ2 focuses on accuracy. Given the current setup and task scope in RQ2, the advantages of CNL-P are not fully apparent, as other models also perform well.
- For a more robust finding that CNL-P is better suited for weaker models, it would be beneficial to include additional experiments with weaker models beyond GPT-4-o mini.

**Questions:**

- Considering the setup and the consistency of output generation with single-turn GPT-4-o prompting, what advantages does CNL-P offer over single-turn generation with GPT-4-o?
- Given the broad goals of this work and the multi-faceted design of CNL-P, what rationale led the authors to focus on these three specific research questions?

---

> ### Author Response · Authors · 2024-11-20
>
> Dear Reviewer，
>
> We sincerely appreciate your insightful comments and constructive feedback, which have been invaluable in helping us clarify and refine key aspects of our paper.
>
> Below, we present detailed responses to your comments, prepared with careful consideration and thorough discussion over the past few days. Due to the content length limitations of a single response, our replies are spread across multiple rounds. We apologize for any inconvenience this may cause.
>
> ***Comment***:
> *The specific aims of the work remain unclear; while high-level challenges and design considerations are presented, the precise goals are hard to identify.*
>
> ***Response***:
> We envision to advance a more natural and robust SE paradigm as an AI infrastructure to enhance human-AI Interaction. By proposing CNL-P, together with the developed NL-to-CNL-P transformer agent (Section 3.2), and a CNL-P linting tool (Section 3.3), this paper opens the door to achieving this ambitious vision.
>
> CNL-P is a LLM-native, PL-independent, and requirement-oriented controlled natural language for prompt. It is fundamentally different from existing focus on LLM-augmented and implementation-oriented frameworks (e.g., Langchain, DSPy).
>
> We recognize that a controlled natural language like CNL-P alone cannot address all the challenges of LLM-based agent development. Just as programming languages rely on compilers and  an ecosystem of supporting tools—such as linters and testing frameworks—CNL-P also requires a robust ecosystem of software engineering tools to support the development, testing, maintenance and operation of CNL-P agents. This work, though in its early stage, illustrates that CNL-P paves the way for the development of such a comprehensive SE4AI infrastructure.
>
> CNL-P is designed to support an evolving ecosystem of SE tools that lower the barriers for adoption and improve user accessibility. In this paper, we presented the proof of concept of two such tools:
>
> - a Transformer agent which formats a free-form NL prompt into the CNL-P syntax (Section 3.2);
> - a CNL-P linting tool which validates and optimizes natural language requirements for clarity and correctness through systematic static analysis (Section 3.3).
>
> Furthermore, we discussed the LLM-empowered CNL-P Compiler which can transform high-level requirements in CNL-P into executable code for PL-based frameworks like DSPy or LangChain.
>
> We envision more LLM-assisted software engineering tools to support the entire lifecycle of CNL-P development, for example,
>
> - A CNL-P tutorial agent that can teach CNL-P, provide relevant examples, and answer CNL-P syntax and semantic questions.
> - A requirements agent that can iteratively developing detailed CNL-P requirements from initial vague and incomplete natural language descriptions, similar to the agile requirement analysis process.
> - A CNL testing framework that can detecting incompleteness, inconsistencies or ambiguities in requirements, similar to Gherkin User Stories and Behavior Driven Development (BDD).
> - A documentation agent that can transform CNL-P requirements into technical documentation (e.g., reStructuredText) to integrate CNL-P into the broader software DevOps context.
>
> Such a SE tool ecosystem around CNL-P aligns with software engineering principles and best practices such as early error detection, which reduces the cost of fixing issues during later stages of development. By identifying errors or inconsistencies at the requirements level, CNL-P prevents costly implementation mistakes. All such tools will constitute a CNL-P centric SE4AI infrastructure, standing on the shoulders of LLMs for more people to leverage AI in their life lives and work.

---

> > ### Author Response · Authors · 2024-11-20
> >
> > ***Comment***:
> > *Given the broad goals of this work and the multi-faceted design of CNL-P, what rationale led the authors to focus on these three specific research questions?*
> >
> > ***Response***:
> > The main purpose of the three RQs is to simulate several key steps of end users when using CNL-P to verify the rationality of our design and explore the opportunities enabled by CNL-P.
> >
> > Starting from the requirements proposed by end users and described in natural language (NL), RQ1 explores the process of converting the requirements described in NL into CNL-P, RISEN, and RODES. Using the LLM to simulate human scoring, it compares the effects of CNL-P and template-based methods such as RISEN and RODES across the five evaluation dimensions: Adherence to Original Intent, Readability and Structural Clarity, Modularity, Extensibility and Maintainability, and Process Rigor. We will further enhance RQ1 with human evaluation as requested by the reviewer.
> >
> > Due to the introduction of precise syntax and semantics in CNL-P, CNL-P appears more complex in form, compared to simple template-based methods. Users may want to know whether this complexity will affect the understanding and execution of prompts expressed in CNL-P by LLMs. RQ2 verifies that without any additional language explanation, few-shot examples, or training of the LLMs, the LLMs can well understand and execute the prompts described by CNL-P, and the obtained effect is (at least) no worse than that of well-organized natural language prompts. The results confirm the intuitiveness of CNL-P syntax and semantics.
> >
> > In addition to demonstrating the intuitiveness of CNL-P and its effect on the quality of LLM responses, RQ3 further shows the use of the linting tool developed for CNL-P to conduct syntax and semantic analysis of the prompts expressed in CNL-P. Linting is a widely adopted static code analysis technique for identifying errors in code written in PLs, but this linting capability is impossible to achieve with the existing prompt methods based on natural language. Our CNL-P makes it possible for the first time, and RQ3 also initially demonstrates the technical feasibility of building a series of SE tools around CNL-P.
> >
> > Through the experimental verification of these three RQs, we aim to prove the rationality and potentials of the CNL-P language design: it not only standardizes the prompt generation process but also opens the door to building a SE4AI infrastructure around controlled natural language, further promoting the progress of the emerging natural language centric programming paradigm by standing on the shoulders of LLMs and for LLMs.

---

> > > ### Author Response · Authors · 2024-11-20
> > >
> > > ***Comment***:
> > > *Experiment setup in RQ1 lacks clarity on how the five dimensions are measured and how the 93 prompt instances were chosen. There are also no human validation results presented, even as partial samples.*
> > >
> > > ***Response***:
> > > We provide the response in the following three sub-sections.
> > >
> > > *Explanation of the process of prompt instance selection*
> > > The source of the 93 examples in RQ1 is a project named "awesome-chatgpt-prompts" on Github, which has received 113k stars. We initially screened 93 relatively long Prompts from it, hoping to make them have relatively complete requirements and rich content, thereby converting high-quality input for the evaluation of CNL-P, RISEN, and RODES.
> > >
> > > *Introduction of new human evaluation experiments*
> > > Thanks to the reviewer's suggestion, we are making every effort to design and conduct new experiments to further validate the proposed conclusions. We aim to obtain meaningful experimental results within the remaining discussion period and will promptly share any updates as they become available.
> > > The initial setting of our new experiment is as follows:
> > > We plan to form a computer professional group (e.g., master or PhD students in computer science) and a non-professional group (e.g., the students from education and business colleges). Each group is further divided into two subgroups: a subgroup that has not received CNL-P grammar training and a subgroup that has received simple CNL-P grammar training (about 30 minutes). At least two independent raters are arranged in each subgroup for scoring. The difference in the use of CNL-P between professionals and non-professionals can be compared between the professional group and the non-professional group, while the entry threshold and ease of getting started of CNL-P can be reflected between the training group and the non-training group. When scoring, the judges are not required to give specific scores directly (such as 0-100 points), but to score by grades, for example in a 5-point Likert scale, score it across the same five dimensions that are used for LLM-based evaluation.
> > >
> > > *Scoring criteria for the five dimensions in RQ1*
> > > The scoring indicators of the five dimensions in RQ1 were not included in the paper due to the length limitation of the paper. During the actual experiment, the specific scoring criteria are presented in the next reply (due to the length limit of a single reply). We will add them to the appendix.

---

> > > > ### Author Response · Authors · 2024-11-20
> > > >
> > > > Scoring Criteria:
> > > >
> > > > 1. Adherence to Original Intent
> > > >    Scoring Range:
> > > >    1-33: Significantly deviates from the original intent, with numerous omissions, misunderstandings, or misuse of structured keywords that distort the original semantics.
> > > >    34-66: Mostly aligns with the original intent but contains some inaccuracies or unnecessary modifications due to structured keywords, causing slight deviations from the intended meaning, or an overemphasis on structure over semantics.
> > > >    67-100: Completely faithful to the original semantic content, with structured keywords effectively clarifying module boundaries and enhancing understanding without altering or losing any of the original meaning. Examples from the original are adapted appropriately, maintaining the intended message.
> > > > 2. Readability and Structural Clarity
> > > >    Scoring Range:
> > > >    1-33: Structure is unclear, with structured keywords either absent or misused, leading to ambiguous module boundaries, redundancy, or a lack of clarity; the LLM would struggle to parse and understand the Prompt.
> > > >    34-66: Structure is somewhat clear, and structured keywords are present but inconsistently or inadequately applied, resulting in some ambiguity or minor redundancy; requires more processing effort from the LLM to understand the modular structure.
> > > >    67-100: Structure is very clear, with structured keywords skillfully applied to define module boundaries, enhance readability, and provide unambiguous guidance, ensuring the LLM can easily interpret and understand the Prompt.
> > > > 3. Modularity
> > > >    Scoring Range:
> > > >    1-33: Low level of modularity; the Prompt fails to effectively organize structured knowledge, resulting in poorly defined or highly coupled modules.
> > > >    34-66: Demonstrates moderate modularity, with reasonable identification of structured knowledge, but some modules show dependencies or coupling.
> > > >    67-100: Exhibits a high level of modularity, accurately identifying and organizing structured knowledge into well-defined, independent modules with minimal redundancy and coupling.
> > > > 4. Extensibility and Maintainability
> > > >    Scoring Range:
> > > >    1-33: Lacks extensibility and maintainability; unclear rationale for changes, making modification difficult and destabilizing the overall structure.
> > > >    34-66: Offers some degree of extensibility and maintainability, but modification points are not always clear, requiring additional judgment for adjustments.
> > > >    67-100: Highly extensible and maintainable, with well-defined guidance for changes, allowing accurate and seamless modifications without disrupting the overall structure.
> > > > 5. Process Rigor
> > > >    Scoring Range:
> > > >    1-33: Lacks clear process rigor, with unclear instruction iterations, variable management, or input-output handling, leading to confusion in LLM execution.
> > > >    34-66: Demonstrates a moderately clear process, but certain iterations, variable passing, or input-output elements are not fully defined, requiring the LLM to infer missing steps.
> > > >    67-100: Process is rigorously defined, with clear iterations, variable passing, and input-output flow, allowing the LLM to execute instructions with precision.

---

> > > > > ### Author Response · Authors · 2024-11-20
> > > > >
> > > > > ***Comment***
> > > > > *RQ1 primarily assesses design considerations, while RQ2 focuses on accuracy. Given the current setup and task scope in RQ2, the advantages of CNL-P are not fully apparent, as other models also perform well.*
> > > > >
> > > > > ***Response***
> > > > > Based on the current experimental data, we agree that the advantages of CNL-P are indeed not obvious. We apologize for not expressing the objective of RQ2 clearly in the paper. RQ2 is mainly to prove that CNL-P will not cause LLM to have difficulty in understanding NL prompts due to the introduction of precise grammar and semantics. The experimental results show that the experimental LLMs can correctly understand the meaning of CNL-P without the needs for any grammar explanation and few-shot Learning. That is, the precise grammar and semantics of CNL-P do not bring overhead to the LLM's understanding of the prompts expressed in CNL-P. In several evaluation indicators proposed in RQ2, CNL-P is slightly better than NL overall and is comparable to RISEN and RODES in tasks such as gpt4o - task1424, gpt4o - task1678, gpt4omini - task190, gpt4omini - task385, gpt4omini - task1678, Llama3-70B - task190, and Llama3-70B - task1678. We will add and explain the detailed results in the revision.
> > > > >
> > > > > ---
> > > > >
> > > > > ***Comment***
> > > > > *For a more robust finding that CNL-P is better suited for weaker models, it would be beneficial to include additional experiments with weaker models beyond GPT-4-o mini.*
> > > > >
> > > > > ***Response***
> > > > > We sincerely thank the reviewers for pointing out that our initial conclusion, "CNL-P is better suited for weaker models," was not entirely accurate and did not fully align with our RQ2 objective. What we intended to convey is that "relatively weaker models can comprehend the syntax and semantics of CNL-P and execute it without significant difficulty, and similarly, stronger models can also handle CNL-P effectively."
> > > > >
> > > > > Thank you for the reviewer's comments. Our current experiment is not sufficient to draw such a conclusion, given the results of only three LLMs. Limited by time and computing resources, we were only able to conduct preliminary exploration and attempts on only three models at the time of paper submission.
> > > > >
> > > > > We are making every effort to design and conduct new experiments to further validate the proposed conclusions. We aim to obtain meaningful experimental results within the remaining discussion period and will promptly share any updates as they become available. We believe the new experiments will help us better understand the low-bound capability of LLMs to understand and execute CNL-P reliably and the general applicability of CNL-P.
> > > > >
> > > > > ***Comment***
> > > > > *Considering the setup and the consistency of output generation with single-turn GPT-4-o prompting, what advantages does CNL-P offer over single-turn generation with GPT-4-o?*
> > > > >
> > > > > ***Response***
> > > > > Through the exploration of RQ2, we demonstrate that CNL-P offers an intuitive and simple syntax, enabling existing LLMs to generate outputs on par with those driven by natural language prompts. Specifically, for single-turn GPT-4-o prompting, CNL-P proves equally effective in producing high-quality LLM responses.
> > > > >
> > > > > In addition, RQ3 further highlights a significant contribution: the introduction of a linting tool as a proof-of-concept, marking the first application of static analysis techniques to Controlled Natural Languages (CNLs). This innovation means CNL-P opens the door to novel SE paradigms for AI infrastructure to enhance human-AI Interaction
> > > > >
> > > > > Looking ahead, we aim to develop additional agents and tools, such as CNL-P compilers and requirement clarification assistants, alongside the linting tool and NL-to-CNL-P transformer agent, to further enhance CNL-P's usability. Moreover, we envision building a robust ecosystem centric on CNL-P, leveraging the strengths of LLMs, to establish a foundation for innovative tooling and infrastructure, unlocking new SE paradigms and advancing human-AI interaction.
> > > > >
> > > > > This vision is achievable due to CNL-P's unique design, which integrates SE and PE best practices, fundamentally distinguishing it from simple prompt templates or style guideline methods. That is, the benefits of CNL-P and the opportunities it enables would be far reaching, beyond just the advantages for single-turn LLM interaction.
> > > > >
> > > > > ---
> > > > >
> > > > > We sincerely hope these responses thoroughly address your comments. If anything remains unclear, we would greatly appreciate the opportunity to discuss it further with you.
> > > > >
> > > > > Thank you again for your time.
> > > > >
> > > > > Best wishes,
> > > > > All authors

---

> > > > > > ### Author Response · Authors · 2024-12-03
> > > > > >
> > > > > > Dear reviewer,
> > > > > >
> > > > > > We have revised the paper based on your comments and uploaded the updated PDF file of the paper. Thank you for your constructive suggestions, which have greatly improved the quality of the paper.
> > > > > >
> > > > > > We would like to take this opportunity to inform you of the specific revised details made in the revised PDF file. These revisions are based on our previous responses. For the sake of brevity, we have not repeated the content of our prior response here. Should you require further details, please kindly refer to the previous response for your reference. Due to the page limit, our paper revisions are more concise than our responses, but they retain all the essential elements.
> > > > > >
> > > > > > **Comments:**
> > > > > >
> > > > > > *The specific aims of the work remain unclear; while high-level challenges and design considerations are presented, the precise goals are hard to identify.*
> > > > > >
> > > > > > **Response:**
> > > > > >
> > > > > > The revisions addressing this issue are in the updated PDF, specifically in:
> > > > > >
> > > > > > **Line 098-099**
> > > > > >
> > > > > > "These insights culminate in our vision of advancing a more natural and robust SE paradigm as an AI infrastructure to enhance human-AI Interaction."
> > > > > >
> > > > > > **Line 534-535**
> > > > > >
> > > > > > "NL to CNL-P transformer agent and a CNL-P linting tool, CNL-P opens the door to achieving the ambitious vision that is advancing a more natural and robust SE paradigm as an AI infrastructure to enhance human-AI Interaction."
> > > > > >
> > > > > > **Comments:**
> > > > > >
> > > > > > *Given the broad goals of this work and the multi-faceted design of CNL-P, what rationale led the authors to focus on these three specific research questions?*
> > > > > >
> > > > > > **Response:**
> > > > > >
> > > > > > The revisions addressing this issue are in the updated PDF, specifically in:
> > > > > >
> > > > > > **Line 1181-1298**
> > > > > >
> > > > > > Section: A.9 OVERALL FINDINGS OF THE EXPERIMENT
> > > > > >
> > > > > > **Comments:**
> > > > > >
> > > > > > *Experiment setup in RQ1 lacks clarity on how the five dimensions are measured and how the 93 prompt instances were chosen. There are also no human validation results presented, even as partial samples.*
> > > > > >
> > > > > > **Response:**
> > > > > >
> > > > > > The revisions addressing this issue are in the updated PDF, specifically in:
> > > > > >
> > > > > > **Line 925-960**
> > > > > >
> > > > > > Section: A.6.1 EVALUATION CRITERIA OF EXPERIMENT IN RQ1
> > > > > >
> > > > > > **Line 298-350**
> > > > > >
> > > > > > We add human evaluation in our experiments.
> > > > > >
> > > > > > **Comments:**
> > > > > >
> > > > > > *RQ1 primarily assesses design considerations, while RQ2 focuses on accuracy. Given the current setup and task scope in RQ2, the advantages of CNL-P are not fully apparent, as other models also perform well.*
> > > > > >
> > > > > > **Response:**
> > > > > >
> > > > > > The revisions addressing this issue are in the updated PDF, specifically in:
> > > > > >
> > > > > > **Line 352-403**
> > > > > >
> > > > > > Section: 4.2 RESEARCH QUESTION 2: UNDERSTANDING OF CNL-P BY LLMS
> > > > > >
> > > > > > **Comments:**
> > > > > >
> > > > > > *For a more robust finding that CNL-P is better suited for weaker models, it would be beneficial to include additional experiments with weaker models beyond GPT-4-o mini.*
> > > > > >
> > > > > > **Response:**
> > > > > >
> > > > > > We have redefined the focus of RQ2.
> > > > > >
> > > > > > "RQ2: Can LLMs understand and execute CNL-P without additional explanations or Few-Shot Learning?"
> > > > > >
> > > > > > The updated description of RQ2 can be found in
> > > > > >
> > > > > > **Line 352-403**
> > > > > >
> > > > > > Section: 4.2 RESEARCH QUESTION 2: UNDERSTANDING OF CNL-P BY LLMS
> > > > > >
> > > > > > We have deleted the relevant claim and discussion of "CNL-P is better suited for weaker models".
> > > > > >
> > > > > > **Comments:**
> > > > > >
> > > > > > *Considering the setup and the consistency of output generation with single-turn GPT-4-o prompting, what*
> > > > > >
> > > > > > **Response:**
> > > > > >
> > > > > > The revisions addressing this issue are in the updated PDF, specifically in:
> > > > > >
> > > > > > **Line 1407-1439**
> > > > > >
> > > > > >
> > > > > > ---
> > > > > > We sincerely hope these revisions thoroughly address your comments.
> > > > > >
> > > > > > Thank you again for your time.
> > > > > >
> > > > > > Best wishes,
> > > > > >
> > > > > > All authors

---

### Comment · Area_Chair_Bdae · 2024-11-27

Dear reviewers,

Thank you for your efforts reviewing this paper. If you haven't, can you please check the authors' responses and see if your concerns have been addressed? Please acknowledge you have read their responses. Thank you!

---

### Meta-Review · Area_Chair_Bdae · 2024-12-23

**Metareview:**

Summary:

This paper proposes a Controlled Natural Language for Prompt (CNL-P) which incorporates best practices in prompt engineering. The motivation is that due to the inherent ambiguity of natural language, prompts often fail to trigger LLM to consistently output high quality responses, particularly for complex tasks.  To overcome the NL's ambiguity, CNL-P introduces precise grammar structures and strict semantic norms, enabling a declarative but structured and accurate representation of user intent. LLMs can potentially better understand and execute CNL-P, yielding higher quality responses. The paper introduces an automatic NL2CNL-P conversion agent based on LLMs, which allows users to describe prompts in NL from which the NL2CNL-P agent generates CNL-P compliant prompts guided by CNL-P grammar. It further develops a linting tool for CNL-P, including syntactic and semantic checks, making static analysis techniques applicable to natural language for the first time.

Strengths:

Reviewers generally agree that the paper clearly motivated framing of prompting as an API, made a strong connection to principles in SE, and has strong technical contributions and potential high research impact.

Weaknesses:

The following limitations are not (completely) addressed, although the authors provided some explanations during the discussion period:

1. The proposed CNL-P does not consistently outperform the NL instruction, or the advantages of CNL-P are not fully apparent. (Reviewer 32pw, Reviewer QGqE)

2. Lack of experiments with poorly informative prompts. (Reviewer QGqE)

3. Lack of  evidence on how the GPT evaluation results correlate with actual human evaluations. (Reviewer 32pw, Reviewer QGqE)

4. Lack of experimental evidence to show “CNL-P is better suited for weaker models”. (Reviewer 32pw, Reviewer QGqE)

**Additional Comments On Reviewer Discussion:**

The authors provide very detailed (although quite lengthy) responses to the reviewers’ comments. Many of their original concerns were addressed, including:

1. Confusions regarding the specific aims of the work.

2. Technical concerns about evaluation metrics, practical applications, and theoretical foundations from Reviewer m8X4.

---

### Decision · Program_Chairs · 2025-01-22

Accept (Poster)